# Quantifying the single scattering albedo for the January 2017 Chile wildfires from simulations of the OMI absorbing aerosol index

Jiyunting Sun[1,2], J. Pepijn Veefkind[1,2], Peter van Velthoven[1], Pieternel F. Levelt[1,2]

[1]Royal Netherlands Meteorological Institute, De Bilt, 3731 GA, the Netherlands
[2]Department of Geoscience and Remote Sensing (GRS), Civil Engineering and Geosciences, Delft University of Technology, Delft, 2628 CD, the Netherlands

*Correspondence to*: Jiyunting Sun (jiyunting.sun@knmi.nl)

**Abstract.** The absorbing aerosol index (AAI) is a qualitative parameter directly calculated from satellite measured reflectance. Its sensitivity to absorbing aerosols in combination with a long-term data record since the late 1970's makes it an
important parameter for climate research. In this study, we attempt to quantify the aerosol absorption by retrieving the single scattering albedo ($\omega_0$) at 550 nm from the satellite measured AAI in near-UV channel. In the first part of this study, AAI sensitivity studies are presented exclusively for biomass burning aerosols. Later on, this study employs a radiative transfer model (DISAMAR) to simulate the AAI measured by the Ozone Monitoring Instrument (OMI) in order to derive $\omega_0$ at 550 nm. Inputs for the radiative transfer calculations are satellite measurement geometry and surface conditions from OMI,
aerosol optical thickness ($\tau$) from the MODerate-resolution Imaging Spectroradiometer (MODIS), and aerosol micro-physical parameters from the AErosol RObotic NETwork (AERONET), respectively. This approach is applied to the Chile wildfires for the period from 26 to 30 January 2017, when the OMI observed AAI of this event reached its peak. The Cloud and Aerosol Lidar with Orthogonal Polarization (CALIOP) overpasses failed to capture the complete evolution of the smoke plume over the research region, therefore the aerosol profile is parameterized. The simulated plume ascends to an altitude of
4.5-4.9 km, which is in good agreement with available CALIOP backscatter coefficient measurements. Due to the data may contain the pixels outside the plume, an outlier detection criterion is applied. The results show that the AAI simulated by DISAMAR is consistent with satellite observations. The correlation coefficients fall into the range between 0.85 and 0.95. The retrieved mean $\omega_0$ at 550 nm for the entire plume over the research period from 26-30 January 2017 varies from 0.81 to 0.87, whereas the nearest AERONET station reported values in the range from 0.89 to 0.92. The difference in geolocation of
the AERONET site and the plume, the assumption of homogeneous and static plume properties, the lack of the aerosol profile information, and the uncertainties in the inputs for radiative transfer calculation are primarily responsible for this discrepancy.

## 1 Introduction

Biomass burning aerosols are generated from combustion of carbon-containing fuels, either by natural or anthropogenic
processes (Bond et al., 2004; IPCC, 2014). They are of great concern from the climate perspective (Kaufman and Boucher, 2002; IPCC, 2007; Koch and Del Genio, 2010; Huang et al., 2013; IPCC, 2014). The reported radiative forcing of black carbon (BC) produced by fossil fuel and biofuel is around 0.4 Wm$^{-2}$ (0.05 – 0.80 Wm$^{-2}$) (Ramanathan and Carmichael, 2008; Bond et al., 2013; Huang et al., 2013), but this estimate is highly uncertain. Accurate measurements of the aerosol single scattering albedo ($\omega_0$) on a global scale can reduce the uncertainty in aerosol radiative forcing assessments (Hu et al., 2007).
$\omega_0$ is defined as the ratio of the aerosol scattering over the extinction. Currently $\omega_0$ is mainly measured by ground-based instruments (Dubovik et al., 1998; Eck et al., 2003; Petters et al., 2003; Kassianov et al., 2005; Corr et al., 2009; Yin et al., 2015). Satellite sensors, such as the POLarization and Directionality of the Earth's Reflectances (POLDER), can retrieve $\omega_0$ from a combination of multi-angular, multi-spectral observations of the polarized radiation. By measuring the anisotropy of the reflected radiance for each pixel, POLDER is expected to determine the reflected solar flux more accurately (Leroy et al.,

1997). Unfortunately, there is no continuous temporal coverage of $\omega_0$ because the first two POLDER missions ended

prematurely due to technical problems, and the third POLDER mission only covered the period 2004-2014. Instead, satellite

derived $\omega_0$ is usually retrieved simultaneously with the aerosol optical thickness ($\tau$) based on the pre-defined aerosol

properties, such as the near-UV aerosol product (OMAERUV) of the Ozone Monitoring Instrument (OMI) (Torres et al.,

2005; Torres et al., 2007). But this aerosol absorption over near-UV is highly sensitive to the assumption on aerosol layer

height. Satheesh et al. (2009) therefore used the $\tau$ from MODerate-resolution Imaging Spectroradiometer (MODIS), which is

independent of aerosol layer height, to constrain the OMAERUV retrieval. Their validation showed that compared with

operational OMAERUV algorithm, the retrieved aerosol height by the hybrid method is in a better agreement with air-borne

measurements, implying a potential improvement in aerosol absorption retrieval. This OMI-MODIS joint retrieval was also

evaluated by Gassó and Torres (2016). They found that under less absorbing conditions, the hybrid method is sensitive to the

variations in the input $\tau$, which is used to select the retrieved pair of aerosol layer and $\omega_0$.

Herman et al. (1997) first defined the near Ultra-Violet (UV) absorbing aerosol index (AAI), which provides an alternative

methodology to retrieve $\omega_0$ from satellite observations. The near-UV AAI, usually derived from the spectral range between

340 and 390 nm, is a qualitative measure of absorbing aerosols that was first provided by the Total Ozone Mapping

Spectrometer (TOMS) on-board Nimbus-7 in 1979. Since then several instruments have contributed to the AAI data record,

that now spans nearly four decades. This long data record is an important motivation for us to derive quantitative aerosol

absorption information in the near-UV channel.

The most important advantage of the satellite retrieved AAI is that it does not depend on a-prior aerosol types, which are

major uncertainties in aerosol parameter retrievals, such as $\tau$. Ginoux et al. (2004) suggested that comparing model

simulations with AAI from TOMS allows a better control of discrepancies because the only error source is from models.

Further advantages of AAI are the low reflectivity of the Earth's surface and the absence of significant molecular absorption

over the near-UV range. Using this band can ensure the aerosol absorption is one of the major contributors to the total signal.

Moreover, the near-UV AAI is by definition highly sensitive to aerosol absorption. Previous studies have proven the

potential of the near-UV AAI from TOMS in absorbing aerosol properties retrieval. Torres et al. (1998) provided the

theoretical basis of an inversion method to derive $\tau$ and $\omega_0$ from backscattered radiation. This method was validated by

ground-based observations during the Southern African Regional Science Initiative (SAFARI) 2000 measurement campaign.

The agreement of $\tau$ and $\omega_0$ reaches $\pm 30\%$ and $\pm 0.03$, respectively (Torres et al., 2005). Hu et al. (2007) retrieved global

columnar $\omega_0$ based on the AAI from TOMS with an average uncertainty of 15%.

This study is inspired by previous research to quantify the aerosol absorption from AAI. We use the near-UV AAI provided

by OMI on-board Aura, the successor of TOMS, to derive the aerosol properties of the central Chile (Pichilemu 34.39°S,

72.00°W and Constitución 35.33°S, 72.42°W) wildfires in January 2017. The series of fires were triggered by a combination

of long-term drought and high temperature, and were regarded as the worst wildfire season in the national history (The

Guardian, 2017). The fires led to evacuations of the affected areas and massive losses of the local forestry industry (pine and

eucalyptus forests) (NASA.gov, 2017). The smoke plume was transported away from the source regions towards the tropical

area in the Pacific Ocean by north-westward winds (Fig.1). In this study, we quantitatively retrieve the $\omega_0$ of this smoke by

simulating the near-UV AAI of OMI with the radiative transfer model Determining Instrument Specifications and Analysing

Methods for Atmospheric Retrieval (DISAMAR). The aerosol inputs of DISAMAR includes the $\tau$ retrieved from MODIS

on-board the NASA EOS Aqua satellite, and information on aerosol micro-physical parameters provided by AERONET. In

the next section, we provide a brief introduction on the near-UV AAI and its sensitivity to various parameters. The

methodology is described in section 3. In section 4, retrieved results and uncertainty analysis of Chile 2017 wildfires are

discussed, followed by main conclusions in section 5.

## 2 AAI sensitivity studies based on DISAMAR

In this section, we first introduce the near-UV AAI. In the sensitivity analysis, we show that the AAI depends not only on aerosol parameters, but also on the surface conditions and the observation geometry. The sensitivity analysis in this study is only designed for biomass burning aerosols.

### 2.1 Near-UV AAI definition

The concept of the near-UV AAI was first conceived to detect UV-absorbing aerosols from the spectral contrast provided by TOMS observations, known as the residue method (Herman et al., 1997). The basic idea of the residue method is that in a pure Rayleigh atmosphere, the reflectance (or equivalently the radiance ($I_\lambda$)) decreases strongly with the wavelength. The presence of absorbing aerosols will reduce this spectral dependence of $I_\lambda$. The change in this wavelength dependence is summarized as the AAI, which is calculated from the $I_\lambda$ at the wavelength pair $\lambda_1$ and $\lambda_2$ ($\lambda_1 < \lambda_2$):

$$\text{AAI} = -100 \left( log_{10} \left(\frac{I_{\lambda 1}}{I_{\lambda 2}}\right)^{obs} - log_{10} \left(\frac{I_{\lambda 1}}{I_{\lambda 2}}\right)^{Ray} \right), \tag{1}$$

The *obs* and *Ray* denote the $I_\lambda$ is measured by satellite and calculated using a Rayleigh atmosphere, respectively. The longer wavelength $\lambda_2$ is treated as reference wavelength where the surface albedo ($a_s$) is determined by fitting the observed radiance, i.e. $I_{\lambda 2}^{Ray}(a_s) = I_{\lambda 2}^{obs}$. This is done using an atmosphere containing only molecular scattering bounded by a Lambertian surface. The spectral dependence of the surface albedo is neglected thus $I_{\lambda 1}^{Ray}$ is calculated using the same value for $a_s$. Defining $\Delta I_{\lambda 1} = I_{\lambda 1}^{Ray} - I_{\lambda 1}^{obs}$ , Eq.(1) can be rewritten as:

$$\text{AAI} = 100 log_{10} \left(\frac{\Delta I_{\lambda 1}}{I_{\lambda 1}^{obs}} + 1\right) \tag{2}$$

It is advantageous to use Eq.(2) because the AAI can be simply interpreted as the ratio between the simulated and observed radiance at $\lambda_1$.

### 2.2 Near-UV AAI sensitivity studies

In this section, we present results from sensitivity studies performed by the radiative transfer model DISAMAR. DISAMAR can perform simulations of the forward $I_\lambda$ spectrum in a wide spectral coverage (270 nm to 2.4 μm) and model scattering and absorption by gases, aerosols and clouds, as well as reflection by the surface (De Haan, 2011). It uses either the Doubling-Adding method or the Layer Based Orders of Scattering (LABOS) for the radiative transfer calculations. In this study the latter one is employed, because it is less computationally intensive ( De Haan et al., 1987; De Haan, 2011). DISAMAR allows to apply several aerosol scattering approximations. Here we assume Mie scattering aerosols. The parameters to describe Mie particles and their corresponding values are listed in Table 1. Considering the Chile wildfires plumes, which were dominated by biomass burning aerosols, these sensitivity studies are specifically performed for parameterized smoke aerosols, with only fine mode particles and weak linear wavelength dependence of the complex refractive index ($n_r$ and $n_i$). The default values refer to observations of the daily average on January 27 of the AERONET station Santiago_Beauchef (33.46°S, 70.66°W). We obtain the size distribution function and complex refractive index at 440, 675, 880 and 1018 nm from AERONET, and apply a linear interpolation / extrapolation to derive the complex refractive index over the spectrum from 340 to 675 nm, with spectral resolutions of 2 nm. Then DISAMAR uses above information to calculate the aerosol phase function P(Θ) and $\omega_0$ over the full spectrum (340 to 675 nm). The corresponding P(Θ) at 354 nm for default case ($r_g$ = 0.15 μm, $n_r$ = 1.5 and $n_i$ = 0.06) is presented in Fig.2. P(Θ) for other cases are provided in the Appendix A (Fig.A1, A2 and A3). DISAMAR requires τ to be defined at reference wavelength 550 nm. Surface parameters include a spectrally flat $a_s$ and the surface pressure $P_s$. The aerosol profile is parameterized as a single layer box shape, with its bottom

at $z_{aer}-\Delta z/2$ and top at $z_{aer}+\Delta z/2$, where $z_{aer}$ and $\Delta z$ are the geometric central height and the geometric thickness of the aerosol layer, respectively. The whole sensitivity analysis is performed for cloud-free conditions. The wavelength pair of OMI (354 and 388 nm) is applied to compute the AAI. To make different sensitivities studies comparable, the AAI calculated in this section is normalized by the maximum value in each sensitivity study. Note that sensitivity study always use the default settings listed in Table 1 unless different values are explicitly mentioned.

Aerosol optical properties are determined by micro-physics, such as the real and imaginary part of the complex refractive index ($n_r$ and $n_i$), and the particle size ($r_g$). Fig.3 shows how is the variation of the AAI, $\Delta I_{\lambda1}$, $I_{\lambda1}^{obs}$ as well as of the optical properties ($\omega_0$ and the asymmetry factor g) associated with the complex refractive and the particle size. The asymmetry factor g is the averaged cosine of the scattering angle $\Theta$, weighted by P($\Theta$). Fig. 3 shows that the effect of the complex refractive index is dual. As shown in Fig.3 (a), an increase in the real part of refractive index $n_r$ directly enhances the magnitude of $I_{\lambda1}^{obs}$, whereas $\Delta I_{\lambda1}$ reduces. This results in low values of the AAI, which correspond to a high $\omega_0$ (Fig.3 (b)). Under the condition that measurement angle is $\Theta=150°$, the declining g implies that more light is scattered in the line-of-sight of the detector, thus the higher $I_{\lambda1}^{obs}$. Conversely, the imaginary part of refractive index $n_i$, which is directly associated with $\omega_0$, has an opposite influence, see Fig.3 (c) and (d). The particle size distribution has a more complicated influence on the AAI. As shown in Fig.3 (e), the AAI first decreases and then increases, when $r_g$ is varying from 0.1 to 0.4 μm. The AAI primarily follows the behaviour of $\Delta I_{\lambda1}$, whereas $\omega_0$ is continuously decreasing and g is continuously increasing.

In addition to the micro-physics, the concentration and vertical distribution of aerosols also have a strong influence on the wavelength dependence of the radiance $\Delta I_{\lambda1}$. As shown in Fig.4 (a), the AAI is positively correlated with $\tau$. The AAI is highly sensitive to the aerosol vertical distribution (Herman et al., 1997; Torres et al., 1998; de Graaf et al., 2005). As the aerosol layer ascends (Fig.4 (b)), more molecular scattering beneath the aerosol layer is shielded, which reduces $I_{\lambda1}^{obs}$ while increases $\Delta I_{\lambda1}$. The relation between the AAI and $z_{aer}$ is almost linear. Fig.4 (c) shows that at the same altitude, the AAI slightly increases with the geometrical thickness of the aerosol layer. The reason could be that a larger $\Delta z$ indicates the coming sunlight has a higher possibility to be absorbed by aerosols, slightly enhancing the aerosol absorption. Although the sensitivity exists, the impact is only up to 5%, which is negligible for practical purposes.

The calculated AAI does not only depend on the aerosols themselves, but also on ambient parameters such as surface and clouds. Although the near-UV AAI is capable to distinguish absorbing and non-absorbing agents (Herman et al., 1997) and even to retrieve aerosol information over clouds (Torres et al., 2012), the uncertainty triggered by clouds is relatively high and therefore cloudy conditions are excluded in this study. Surface conditions are parameterized by $P_s$ and $a_s$. It can be seen in Fig.5 (a) that a decrease in $P_s$, or equivalently an elevated terrain height, leads to less Rayleigh scattering shielded between the surface and the aerosol layer. As a result, the AAI decreases significantly due to smaller $\Delta I_{\lambda1}$, in agreement with a previous study (de Graaf et al., 2005; Colarco et al., 2017). According to de Graaf et al. (2005), increasing $a_s$ has two counteracting effects. On the one hand, it increases the amount of directly reflected radiation at the top of the atmosphere, namely a larger $I_{\lambda1}^{obs}$, on the other hand it enhances the role of absorption by the aerosol layer rather than the surface, namely a larger $\Delta I_{\lambda1}$. Which effect of $a_s$ is dominant depends on $P_s$ (Fig.5 (b)). When the aerosol layer is relative to the sea level ($P_s$ = 1013 hPa), the first effect dominates. However, a brighter surface compensates the loss of molecular scattering shielded by the aerosols when the terrain height rises ($P_s$ = 813 hPa), which makes the absorbing layer more detectable.

The AAI also depends on the Sun-satellite geometry. Here we provide the AAI as a function of the measurement geometry for the default case with the relative azimuth angle $\Delta\varphi$ = 180°. As presented in Fig.6 (a), the AAI becomes very sensitive to the geometries for zenith angles larger than 60°, which confirms previous research (Herman et al., 1997; Torres et al., 1998; de Graaf et al., 2005). This is mainly due to the significant growth of P($\Theta$) when $\Theta$ becomes smaller (Fig.2). Thus, it is suggested that the OMI measurement with $\theta_0$ larger than this value should be removed due to large variations in the AAI. To analyse the radiance behaviour as previously, we plotted the $I_{\lambda1}^{obs}$ and $\Delta I_{\lambda1}$ as a function of $\Theta$ along the cross section,

respectively (Fig.6 (b)). It is noted that $I_{\lambda1}^{obs}$ increases when $\Theta$ is larger than $90^{\circ}$, whereas the P($\Theta$) decreases at this range (Fig.2). The reason could be that the Rayleigh scattering has an increasing contribution to the radiance at those measurement angles (backscattering).

## 3 Methodology and datasets

In this section, we first present the datasets used and their pre-processing, followed by the strategy to retrieve the aerosol $\omega_0$
while constraining the simulated near UV AAI with the observed one.

### 3.1 Datasets

### 3.1.1 OMI and GOME-2 absorbing aerosol index

The TOMS near-UV AAI retrieval has been proven a robust algorithm and applied to successive sensors, such as OMI on-board Aura and GOME-2 on-board MetOp-A/B. GOME-2 has higher spectral resolution (0.2-0.4 nm) than TOMS, but the
spatial resolution is rather coarse (80×40 km$^2$). In this study, GOME-2 measured AAI at wavelength pair 340 and 380 nm (http://archive.eumetsat.int) is only used as an independent dataset to assess the potential bias of the OMI measurements. OMI combines advantages of both TOMS and GOME-2. It covers wavelengths from 264 to 504 nm with a spectral resolution of approximately 0.5 nm and has a much higher spatial resolution than GOME-2 of 13×24 km$^2$ (Levelt et al., 2006). Note the GOME-2 and OMI have different equator crossing time (9:30 LT descending node for GOME-2 and 13:45
LT ascending node for OMI) that may affect the inter-comparison of the two satellite measurements.
Since OMI was launched in 2004, the AAI retrieved from this instrument has been widely used in various applications. Kaskaoutis et al. (2010) employed the OMI measured AAI for regional research of the aerosol temporal and spatial distribution in Greece. Torres et al. (2012) utilized the advantage of near-UV AAI to detect aerosols over clouds. The OMI observed AAI was even used to evaluate the impact of surface dust loading on human health (Deroubaix et al., 2013).
Buchar et al. (2015) validated the NASA MERRA aerosol reanalysis with the AAI retrieved from OMI.
In this study, the OMI level 2 product OMAERO (https://disc.gsfc.nasa.gov) is used to provide AAI retrieved at the wavelength pair of 354 and 388 nm, and the corresponding viewing geometry and the surface condition when the measurements took place. The samples are included in the radiative transfer simulation only if $\theta_0$ is smaller than 60°, and if satellite pixels are not contaminated by sun-glint, clouds, row anomalies of the instrument, etc. The simulation is only
applied to pixels inside the biomass burning plume, which is defined as AAI values larger than 1 for both OMI and GOME-2.

### 3.1.2 MODIS and OMI aerosol optical thickness

MODIS on-board Aqua/Terra is a sensor that was specifically designed for atmosphere and climate research. The combination of two satellites ensures daily global coverage. The spatial resolution ranges from 250 m to 1 km and it has 36
spectral bands in the wavelength range between 400 nm and 14.4 μm (Remer et al., 2005). MODIS employs separated algorithms for aerosol retrieval over oceans and land (Tanré et al., 1997; Kaufman and Tanré, 1998; Hsu et al., 2004; Remer et al., 2005). Currently the τ provided by MODIS is one of the most reliable datasets (Lee et al., 2009), with an estimated uncertainty of only 3-5% over ocean and 5-15% over land (Remer st al., 2005). Besides, the MODIS retrieved τ is free from the uncertainty triggered by assumed aerosol profile (Satheesh et al., 2009). As mentioned before, DISAMAR requires τ at
550 nm. This study uses cloud-filtered τ at 550 nm from the Collection 6 level 2 product MYD04 as the input for radiative transfer calculation (https://ladsweb.modaps.eosdis.nasa.gov).
In addition, the τ measured by OMI and AERONET are compared with MODIS. The OMAERO τ retrieval uses multi-spectral fitting techniques. The retrieved τ is reported in good accordance with AERONET and is highly correlated with

MODIS (Torres et al., 2007), with a correlation of 0.66 over land and 0.79 over the oceans (Curier et al., 2008), although it suffers from cloud contamination due to the relatively coarse spatial resolution of OMI. Due to the wavelength difference, the $\tau$ measured by OMI at 442 nm has to be transferred to 550 nm using the Ångström exponent (ÅE) 440 – 675 nm taken from AERONET near the time when OMI flies over the selected site. The AERONET dataset used in this study is introduced in the next section.

### 3.1.3 AERONET aerosol properties

AERONET is an aerosol monitoring network of ground-based sun photometers. With standardized instruments, calibration, processing and distribution, AERONET provides a long-term global database for aerosol research and air-borne and space-borne measurement validation. The system takes two basic measurements. The $\tau$ and ÅE are retrieved from the direct solar irradiance measurements; the $r_g$, $P(\Theta)$ (Nakajima et al., 1983; Nakajima et al., 1996), $\omega_0$ (Dubovik et al., 1998), $n_r$ and $n_i$ (Dubovik and King, 2000) are derived from multiple-angular measurements of sky radiance.

The AERONET site nearest to the fire sources of 2017 Chile wildfires is the Santiago_Beauchef (33.46°S, 70.66°W) (https://aeronet.gsfc.nasa.gov). The dataset in use is version 2 level 1.5 product. To minimize the influence of temporal difference, the parameters of AERONET measured closest to the time of the OMI overpass of the site are used to simulate the optical properties of Mie scattering aerosols in DISAMAR. Note that the AERONET level 1.5 dataset is not quality-assured. In addition, the location of this AERONET site is in downtown of Santiago City and close to major roads, where the presence of scattering urban aerosols may bias the measurements of the plume.

The AERONET retrieved $\tau$ and $\omega_0$ are used to evaluate the MODIS $\tau$ and retrieved $\omega_0$, respectively. The AERONET measured $\tau$ is transferred to 550 nm using the ÅE in range 440 – 675 nm while the $\omega_0$ at 550 nm is linearly interpolated by values between 440 and 675 nm.

The AERONET inversion product needs to be processed into the inputs required by DISAMAR. Firstly, a conversion from the volume size distribution $V(r_v, \sigma_v)$ provided by AERONET to the number size distribution $N(r_g, \sigma_g)$ used in DISAMAR is required:

$$\text{N}(r_g, \sigma_g) = \text{V}(r_v, \sigma_v) \frac{3}{4\pi r_g^3} e^{-4.5\sigma_n^2} , \tag{4}$$

The following relation between the geometric and volumetric mean radii ($r_g$ and $r_v$) and standard deviations ($\sigma_g$ and $\sigma_v$) is assumed:

$$r_g = r_v e^{-3\sigma_g^2} , \tag{5}$$

$$\sigma_g = \sigma_v , \tag{6}$$

The fine and coarse mode particle size are derived by finding the two peaks of the log-normal distribution function provided by AERONET. The complex refractive index is assumed the same for both modes. Since bi-modal aerosol is not applicable in DISAMAR yet, we first calculate optical properties of two modes individually, then we externally combine the optical properties of two modes into a bi-modal aerosol with a fraction:

$$w_f = \frac{N_f(r_{g,f}, \sigma_{g,f})}{N_f(r_{g,f}, \sigma_{g,f}) + N_c(r_{g,c}, \sigma_{g,c})} , \tag{7}$$

$$w_c = 1 - w_f , \tag{8}$$

Then the weights for calculating the total $\omega_0$ of the mixed aerosol are:

$$w_{\sigma,f} = \frac{w_f \sigma_f}{w_f \sigma_f + w_c \sigma_c} , \tag{9}$$

$w_{\sigma,c} = 1 - w_{\sigma,f}$ ,         (10)

Where the $\sigma_f$ and $\sigma_c$ are the extinction cross section of the fine and coarse aerosols. The expansion coefficients of the mixed aerosol is weighed by the $\omega_0$ of the fine and coarse aerosols ($\omega_{0,f}$ and $\omega_{0,c}$), respectively:

$$w_{\omega_0,f} = \frac{w_f \sigma_f \omega_{0,f}}{w_f \sigma_f \omega_{0,f} + w_c \sigma_c \omega_{0,c}},$$         (11)

$$w_{\omega_0,c} = 1 - w_{\omega_0,f},$$         (12)

The AERONET instrument at this site only cover the visible and infrared band (440 nm to 1018 nm) for sky radiance measurements, i.e. no aerosol inversion products at UV band. Due to the absence of observations, assumptions have to be made on the spectral dependence of aerosol properties to obtain their values in the near-UV band. The properties of biomass burning aerosol depend on the type of fuel, the procedure producing the smoke, the age of the smoke, and also the atmospheric conditions (Reid et al., 2005). Using measurements to constrain the input aerosol refractive index can reduce the

uncertainties due to a-priori knowledge. Our treatment on the complex refractive index is as following: (1) take the complex refractive index at visible band (440 to 675 nm) from AERONET measurements; (2) linearly extrapolate the complex refractive index to near-UV band. The real part $n_r$ for radiative transfer calculation is obtained in this step. A slight wavelength dependence of $n_r$ is found from the measurements (Fig.9 (a)); (3) for the imaginary part $n_i$, we multiply it (for the entire wavelength from UV to visible) with a scaling factor as we set it as a free parameter. By varying the value of the

scaling factor, both the magnitude and the wavelength dependence of $n_i$ can change to meet the requirement of retrieval (Fig.9 (b)).

### 3.1.4 CALIOP backscattering coefficient

The CALIOP on-board CALIPSO, which was launched in 2006, provides high-resolution profiles of aerosols and clouds. It has three channels with one measuring the backscattering intensity at 1064 nm and the rest measuring orthogonally polarized

components at 532 nm backscattering intensity (Winker and Omar, 2006). Due to the limited spatial coverage, CALIOP did not observe the Chile wildfires plume for all the cases where the OMI observation available. We only use the total attenuated backscatter at 532 nm from level 1B Version 4.10 Standard data to evaluate the parameterized aerosol profiles (https://eosweb.larc.nasa.gov/project/calipso).

### 3.2 Methodology

In this study, we employ the radiative transfer model DISAMAR to simulate the near-UV AAI from OMI and to derive the $\omega_0$ for a specific case, i.e. the Chile wildfires in January 2017. We select the period from 26 to 30 January 2017 (28 January is excluded due to lack of data) when the AAI value reached its peak.

The forward simulation consists of two major steps. First, DISAMR calculates the Mie aerosol optical properties with aerosol micro-physical information taken from AERONET measurements ($r_g$, $n_r$ and $n_i$). As mentioned in Section 3.1.3, we

set the spectral-dependent imaginary part $n_i$ as a free parameter to vary $\omega_0$. Then, DISAMAR operates radiative transfer calculation with the calculated aerosol optical properties for the corresponding aerosol and environmental conditions of OMI.

It is noted that the observed aerosol vertical distribution is limited for the Chile wildfires. Previous research suggested AAI cannot be quantitatively used without $\tau$ or $z_{aer}$ information (Gassó and Torres, 2016). Instead, we implement the same

parameterization as in the sensitivity study to obtain the aerosol layer height. Since the AAI dependence on $\Delta z$ is minor (Fig.4 (c)), and to reduce the computational cost, $\Delta z$ is set a constant of 2 km based on the information from the CALIOP measurements of backscattering coefficient ($\beta$) at 532 nm (Fig.7). The $z_{aer}$, to which the AAI is highly sensitive, is treated as an unknown variable to be retrieved together with $\omega_0$.

Consequently, with various combinations of $z_{aer}$ and $\omega_0$, a lookup table (LUT) of the calculated AAI is constructed with
DISAMAR. It should be noted that for all pixels in the plume we assume the same aerosol microphysical properties as well
as the same aerosol layer height. Pixels outside the plume may have had significantly different properties and this will affect
the results. As shown in Fig.8, the distribution of OMI measurements is sparse in space, which implies that the dataset is
quite sensitive to geographical outliers that may cause the heterogeneous properties of the plume. Consequently, we apply a
data quality control procedure before retrieving $\omega_0$. First, we manually remove the pixels that are geographically isolated
from the main plume. Furthermore, we remove the potential outliers based on statistical tool. We filter the dataset using an
outlier detection based on the interquartile range (IQR) of the AAI difference between DISAMAR simulations and OMI
measurements. According to Tukey's fences (Tukey, 1977), an AAI difference falling outside range between Q1-1.5 IQR
and Q3+1.5 IQR may be regarded as an outlier and removed, where Q1 and Q3 are the first and third quartiles of the AAI
difference, and the IQR is the range between Q1 and Q3. Only the data passing the outlier detection criterion is used to
calculate the cost function (Eq.(13)):

$$\text{RMSE} = \sqrt{\frac{\sum_i^n \left(AAI_{DSM,i}^{qualified} - AAI_{OMI.i}\right)^2}{n}} \,, \tag{13}$$

Here $AAI_i$ indicates the AAI for $i$th satellite pixel of the selected OMI data; subscripts *DSM* and *OMI* indicate DISAMAR
simulation and OMI observation, respectively. The combination of $z_{aer}$ and $\omega_0$ that leads to the minimum residue is used to
simulate the AAI.
Finally, the simulated AAI is compared with OMI observations. We also employ the independent data from GOME-2 as a
reference to identify the potential bias of OMI. Similarly, the $\tau$ retrieved from OMI and AERONET serves as a reference to
that of MODIS. The estimated aerosol layer height and $\omega_0$ at 550 nm are evaluated with independent observations from
CALIOP and AERONET, respectively.

## 4 Results and discussion

By applying the methodology described in the previous section, we quantitatively retrieved the aerosol layer height and $\omega_0$ at
550 nm of the Chile 2017 wildfires by AAI simulation. The OMI measurements of the plume are displayed in Fig.8 (a) – (d).
The presented satellite pixels are with AAI value larger than 1, and are free of cloud contamination, sun-glint and row
anomaly of the instrument. Fortunately, the remaining data is still able to capture the main plume features. It can be clearly
seen that from 26 to 30 January, the plume produced by wildfires in the central Chile was transported by the south-easterly
300  trade wind from the continent towards the lower latitude region of the Pacific Ocean. The plume travelled over a distance of
3000 km during the period.
The vertical movement of the plume is given by CALIOP backscattering coefficient measurements ($\beta$) at 532 nm (Fig.7).
The CALIOP paths closest to the plume are marked by a black dashed line in Fig.7. It is noted that due to the spatial
coverage and the measuring time difference, CALIOP are not able to represent the entire plume detected by OMI. The
305  aerosol layer captured by CALIOP is distributed from 2 km to 6 km, with an average height at approximately 4-5 km. The
ascent of the plume was driven by the heat generated by the fires and sunlight absorption, as well as the atmospheric vertical
motions.
Fig.8 (e) – (h) show the AAI simulation selected by the data quality control mentioned in Section 3.2. The spatial
distribution of the simulated AAI shows similar patterns as the OMI observations. Some data points that are geographically
310  isolated from the plume, e.g. in case 26 and 30 January, differ strongly from what are observed inside the plume. Including
these outliers in the optimization could bias the retrieved aerosol properties. This can also be seen in Fig.8 (i) – (l), where the

points passing the data quality control described in Section 3.2 are highlighted in red color. By removing the outliers, the average spatial correlation coefficient reaches 0.90.

Table 2 lists the statistics of the qualified AAI data, in terms of the median, relative difference and RMSE. The median of measured AAI ranges from 2 to 4 during the research period. Except for 26 January, the median of simulated AAI in other cases is in good agreement with the measurements, with relative differences within ±6%. The RMSE is only acceptable reflects that part of the plume cannot be fit by the assumptions in the forward simulation. The majority of the simulated AAI for 26 January is negatively biased, which is reflected by the small slope without an intercept correction in Fig.8 (i). A systematic bias in the inputs might cause this result. In terms of $\omega_0$, both the AERONET measured and the AAI retrieved aerosol absorption become weaker with time (Table 2). Although the simulated and observed AAI are in good agreement, the difference in $\omega_0$ is significant. The mean of the retrieved $\omega_0$ at 550 nm for the whole period is 0.84, contrast to the AERONET measurements with a mean value of 0.90.

There are many sources contributing to this discrepancy in $\omega_0$. First of all, the nearest site Santiago_Beauchef is not exactly in the primary biomass burning regions as mentioned in section 3.1.3. The AERONET site is located in downtown, where reflective urban or industrial aerosols may have been mixed with the smoke and enhanced the $\omega_0$. This would also affect the spectral dependence of complex refractive index used to constrain the radiative transfer calculation. According to Table 2, the retrieved $n_i$ reveals that the difference between 354 and 388 nm is less than 5%. This small spectral dependence of $n_i$ is mainly determined by AERONET measurements in the visible band (dashed lines), whereas the effect of the scaling factor is minor in this case (Fig.9 (b)). We thus find a much weaker wavelength dependence than the value in Jethva and Torres (2011) study, where a 20% difference between the two UV wavelengths was applied to OMAERUV algorithm to achieve the result that 70% of the retrieved $\omega_0$ differ less than ±0.03 from the $\omega_0$ from the AERONET measurements. This 20% spectral dependence adopted in their work is associated with findings of Hoffer et al. (2006). They conducted in situ measurements on humic-like substances (HULIS) of Amazonia biomass burning aerosols and found that around 35% - 50% light absorption occurred at 300 nm, whereas only around 15% at 400 nm. In terms of the absorbing Ångström exponent (AÅE), a 20% increase at 354 nm with respect to the value at 388 nm is equivalent to an AÅE value between 2.5 and 3, depending on the aerosol models of OMAERUV. According to Kirchstetter et al. (2004), the AÅE of urban pollution is near unit and biomass burning aerosols ranges is approximately 2 between 300 nm to 1 μm. Bergstrom et al. (2007) also confirmed this conclusion from several field programs (SAFARI 2000, ACE Asia, PRIDE, TARFOX, INTEX-A). From the sensitivity study of Jethva and Torres (2011), a stronger spectral dependence of $n_i$ between 354 and 388 nm would allow simulations to reach a higher AAI while keeping $n_i$ at a relatively low level. In our study, this means to retrieve a higher $\omega_0$ at 550 nm. The presence of non-absorbing aerosols weakens the spectral dependence (particularly in the UV spectral range) and the linear extension would overestimate the aerosol absorption in the visible band. Furthermore, the AERONET inversion product is not error-free. The uncertainty of size distribution retrieval is minor for biomass burning aerosols (Dubovik et al., 2000), but under optically thick circumstances, even when retrievals are quality-assured (i.e. level 2 data), the reported accuracy of complex refractive index is 0.04 for $n_r$ and 30%-50% for $n_i$, respectively (Dubovik et al., 2002). It is also reported that AERONET tends to underestimate the absorption of biomass burning aerosols compared with in situ measurements (Dubovik et al., 2002; Reid et al., 2004). The uncertainty of $\omega_0$ is 0.03 under high aerosol loading ($\tau_{440} > 0.5$) and 0.05-0.07 under low aerosol loading (Dubovik et al., 2002; Holben et al., 2006). Last but not least, the spatial representation of the in situ instrument also concerns. Santese et al. (2007) showed that the selected AERONET aerosol parameters can be representative of a 300×300 km$^2$ southeast Italy area. For the Chile wildfires with the most remote pixel over 3000 km away from the continent, the measurements at AERONET cannot fully represent the plume detected by satellites.

Apart from AERONET itself, information from other datasets could also bias our estimate of aerosol absorption. Among all the inputs, the parameterization of a one-layer box-shape aerosol profile could be the largest error source. Although the influence of $\Delta z$ on the AAI is limited (Fig.4 (c)), the AAI calculation highly depends on $z_{aer}$ (Fig.4 (b)). As shown in Table 2,

the estimated plume altitude varies from 4.5 to 4.9 km. As the black solid line indicated in Fig.7, the retrieved $z_{aer}$ can approximately capture the measured geometric vertical location of the plume. The $z_{aer}$ on 26 January seems overestimated because of the temporal and spatial difference. Concretely, CALIOP sampled the plume near the sources and close to the surface, while the plume observed by OMI had been already elevated and transported to the open ocean. The lack of information on the real plume height makes it challenging to determine whether the plume height is responsible for the

systematic bias in Fig.8 (i). Except for 26 January, $z_{aer}$ is in good agreement with what CALIOP observed. Although the retrieved aerosol layer heights are convincing to some extent, one should keep in mind that CALIOP and OMI observations are not exactly co-located. Besides, the parameterized aerosol profile may fail to represent the spatial variation of the plume. Therefore, the uncertainty cannot be directly determined due to the lack of validation data.

Among the four days for which we retrieved $\omega_0$, the value for 27 January is significantly lower than others. For this day the

agreement with CALIOP is reasonable and also the CALIOP track is not far away from the OMI measurement. We therefore explore the effect of observational biases in AAI and $\tau$ on the retrieved $\omega_0$. We investigate the potential bias of these two datasets by plotting the histogram of the AAI measurement difference between GOME-2 and OMI (Fig.10 (a)), against the $\tau$ measurement difference between MODIS and OMI (Fig.10 (b), both are converted into 550 nm). It is clear that on 27 January, the AAI from OMI seems to be overestimated compared to GOME-2. Although the difference in wavelength pair

choice for AAI retrieval, measurement conditions, etc., could contribute to the AAI discrepancy between GOME-2 and OMI, exploring the difference between the two datasets is beyond the scope of this study. In aspect of input aerosol concentration, the $\tau$ from MODIS could be potentially underestimated. Fitting a higher AAI with a lower input $\tau$ leads to an overestimation in aerosol absorption. Here, we analytically quantify the impact of $\tau$ for this specific case by systematically enhancing the $\tau$ of MODIS with a constant variation ($\Delta\tau$) added to all pixels, with the AAI level and the aerosol layer height

remain unchanged. Fig.10 (c) presents how the AAI RMSE and the estimated $\omega_0$ respond to the enhanced $\tau$. It can be clearly seen that an increase in overall $\tau$ level by 0.07 raises $\omega_0$ to 0.84 and optimizes the AAI simulation to a RMSE less than 0.45. If we apply this $\tau$ adaption, the retrieved $\omega_0$ of 27 January becomes more consistent with the other days.

Apart from the observational errors in AERONET, OMI and MODIS data, the assumption that the plume features are homogeneous could also result in the discrepancy between AAI retrieved and AERONET measured $\omega_0$. In reality, the plume

altitude, the optical properties and even the chemical compositions could vary in space and time, while our simulations cannot take into account those effects.

## 5 Conclusions

Biomass burning is a major source of absorbing aerosols making a significant contribution to climate warming. Quantitatively characterizing the absorption by biomass burning aerosols is therefore important to reduce the uncertainty in

assessments of global radiative forcing. Facing the lack of long-term $\omega_0$ records, this study explores an approach to retrieve $\omega_0$ based on reflectivity in the near-UV channel measured by OMI. Although AAI is not a geophysical parameter and depends on many factors, its independence from pre-defined aerosol types, its high sensitivity to aerosol absorption as well as its long-term data record, makes it an attractive parameter to aerosol research.

We test the retrieval of $\omega_0$ for the wildfires happening in central Chile in January 2017. After filtering the data from outliers,

high spatial correlation coefficients (0.85 to 0.95) reach between the simulated and observed AAI. The retrieved aerosol layer heights indicate the plume was elevated to height of 4.5-4.9 km during the research period. These results are in agreement with CALIOP measurements. This average of the retrieved $\omega_0$ at 550 nm is approximately 0.84, which is 0.06 lower than that of AERONET retrieval. The sources for discrepancy includes: the location of the AERONET site that may bias the measured $\omega_0$ and the spectral dependence of complex refractive index; the simplified parameterization of the aerosol

profile; the insufficient spatial representativeness of a single AERONET site; the observational errors in the input aerosol

micro-physics, $\tau$, as well as AAI; and the assumption of homogeneous and static plume properties, which ignores the plume evolution over space and time. We quantitatively analyze the uncertainty of $\tau$ for a specific case (27 January) when the estimated aerosol layer height is in good agreement with the CALIOP measurements. An improvement in retrieved $\omega_0$ can be seen by adapting the magnitude of aerosol concentration.

This study proves the potential of utilizing OMI measured AAI to quantitatively characterize aerosol optical properties like $\omega_0$. Currently, it is challenging to retrieve and validate results without reliable aerosol profile information. In the future, the availability of daily global aerosol layer height data, e.g. the L2 aerosol layer height product of TROPOspheric Monitoring Instrument on-board Sentinel-5 Precursor (TROPOMI) that is underdevelopment (Sanders and de Haan, 2016), are expected to provide a stronger constraint on the forward calculation and to reduce the uncertainty in the retrieved aerosol properties. It

is also reliable to retrieve aerosol absorption for each individual pixel with constraint of the aerosol layer height product. The problem due to the poor spatial representativeness of in situ measurement can then be eased by comparing with the retrievals of nearby satellite pixels. Perhaps, more sophisticated assumptions on spectral-dependent aerosol absorption (e.g. steeper gradient of $n_i$ in UV than visible band) have to be made and evaluated by other observational aerosol properties in UV spectral range, e.g. AERONET measured $\tau$ in UV band, instead of only depending on measured refractive index in visible

band.

**Acknowledgements**

This work was performed in the framework of the KNMI Multi-Annual Strategic Research (MSO). The authors thank to NASA's GES-DISC, LAADS DAAC and ASDC for free online access of OMI, MODIS and CALIOP data. The authors also thank to the Centre for Climate Resilience Research (CR)2 at University of Chile (CONICYT/FONDAP/15110009)

providing the data of the Santiago_Beauchef AERONET station.

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

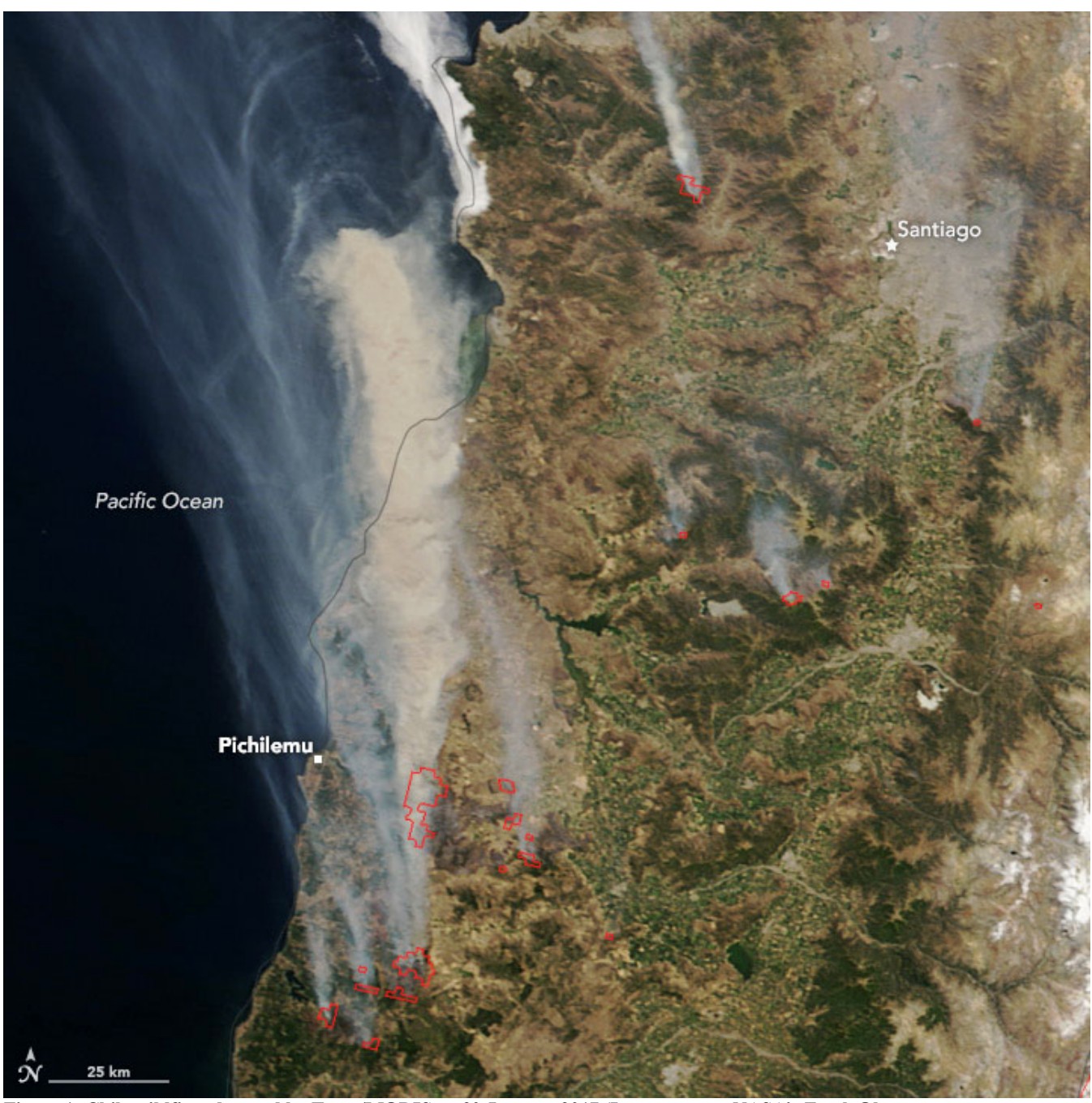


**Figure 1: Chile wildfires detected by Terra/MODIS on 20 January 2017 (Image source: NASA's Earth Observatory https://earthobservatory.nasa.gov/IOTD/view.php?id=89496).**




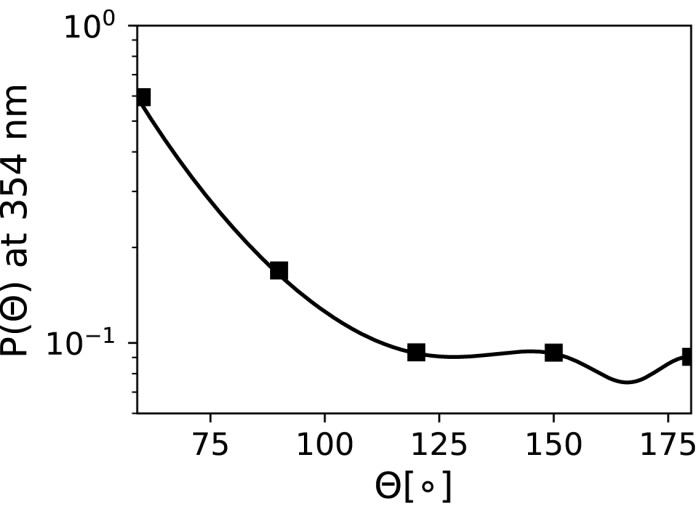

**Figure 2: Phase function p(Θ) at 354 nm of the parameterized Mie scattering aerosol of default case ($r_g$ = 0.15 μm, $n_r$ = 1.5 and $n_i$ = 0.06) in sensitivity analysis. The markers in the plot correspond to the value when Θ=60°, 90°, 120°, 150°, 180°.**





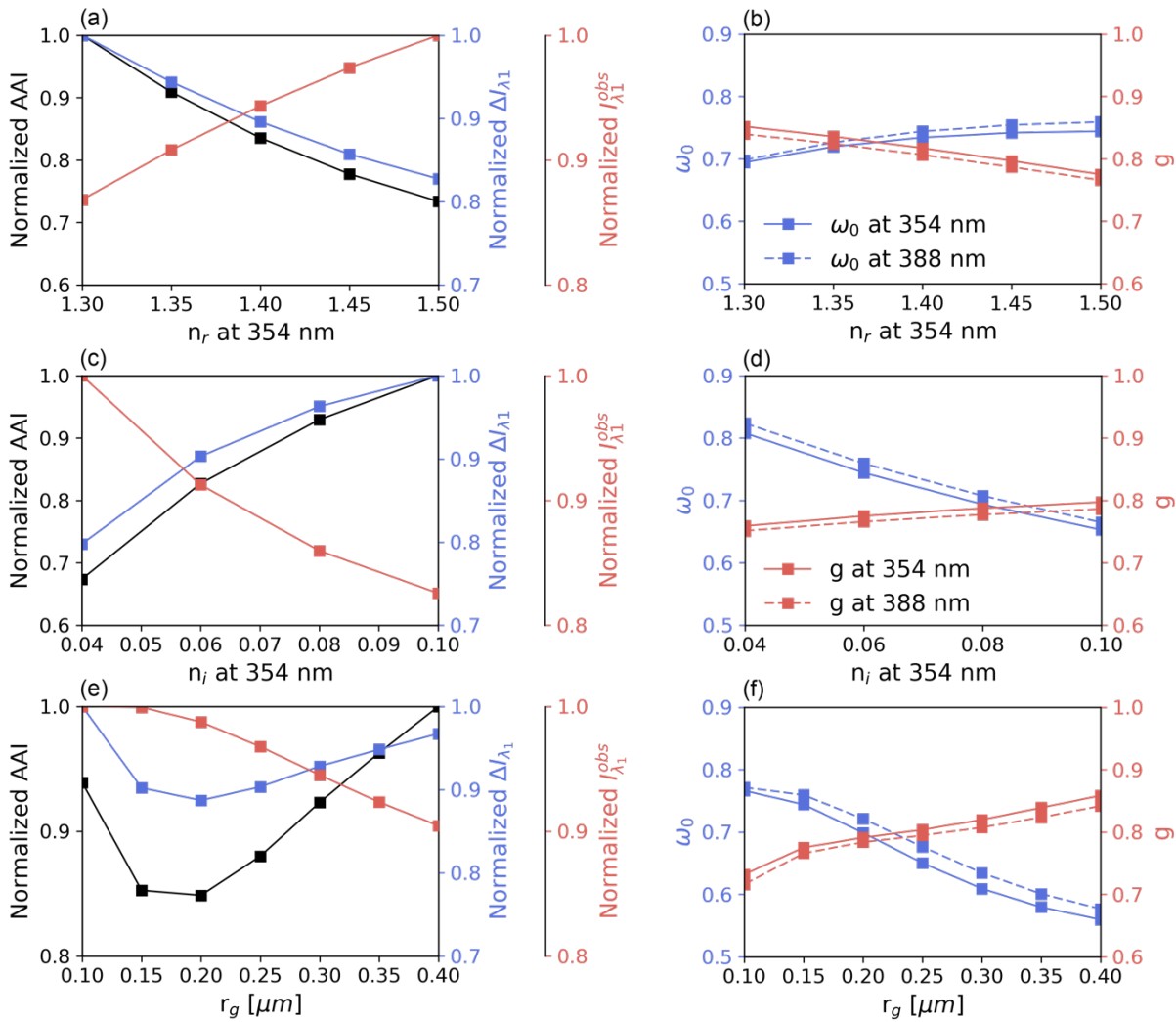

**Figure 3: AAI sensitivity to micro-physical parameters: $n_i$ (a, b), $n_r$ (c, d), and $r_g$ (e, f). The left panels (a, c and e) show the sensitivity of the normalized AAI (black), the normalized $\Delta I_{\lambda 1}$ (blue) and the normalized $I_{\lambda 1}^{obs}$ (red). The right panels (b, d and f) show $\omega_0$ (blue) and g (red) at wavelength 354 (solid line) and 388 (dashed line) nm, respectively.**


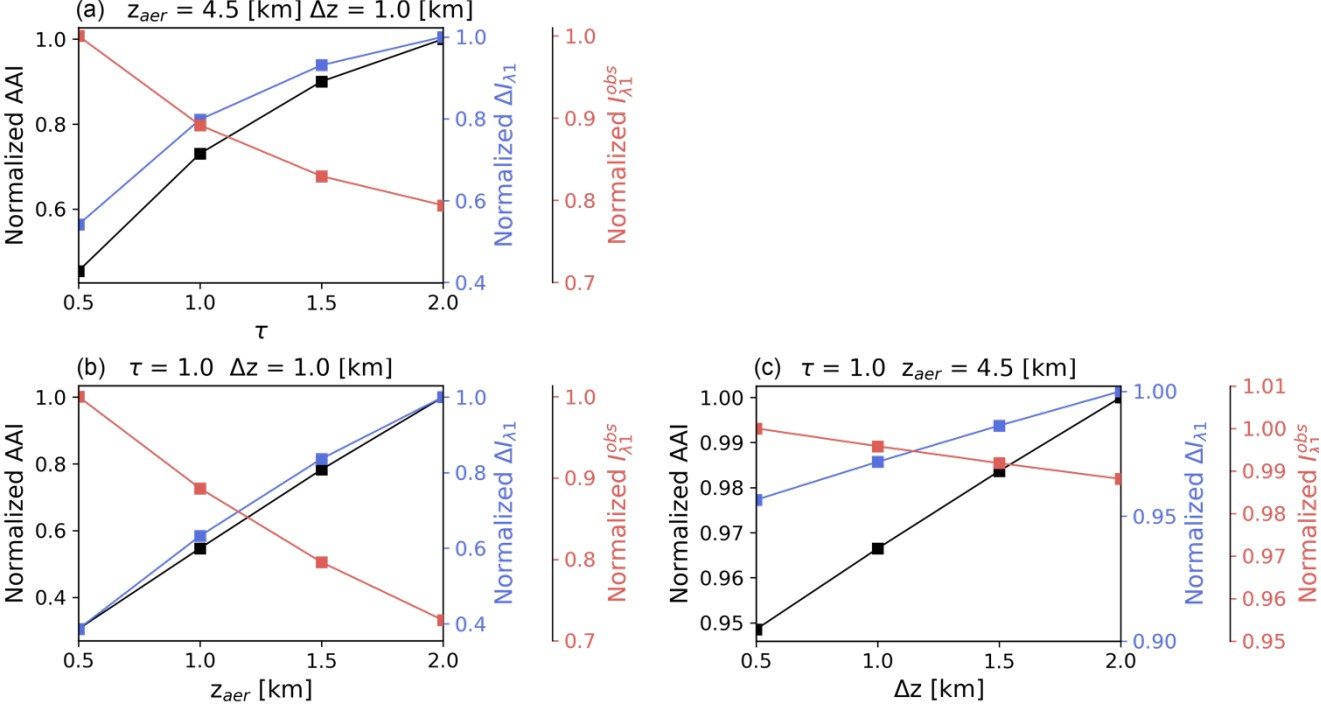

Figure 4: AAI sensitivity to macro-physical parameters: (a) $\tau$ at 550 nm, (b) $z_{aer}$ and (c) $\Delta z$.

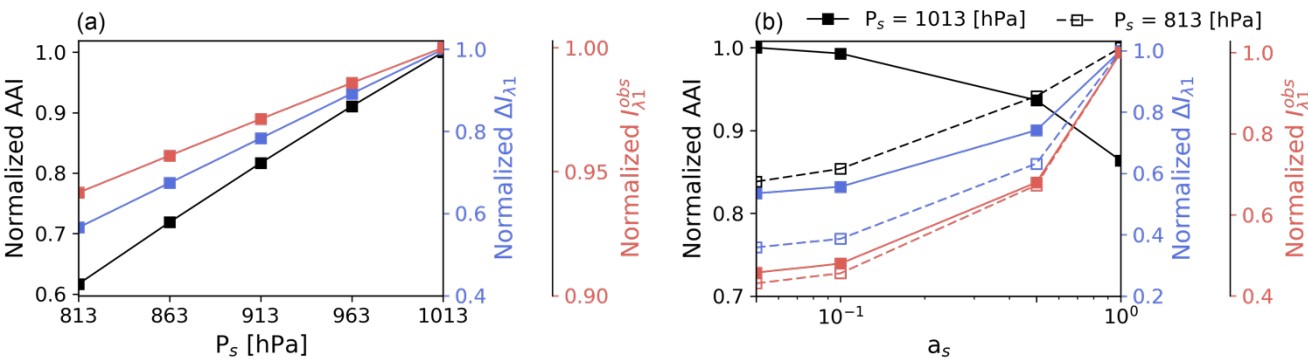

**Figure.5 AAI sensitivity to surface parameters: $P_s$(a) and $a_s$(b). The solid line and dashed line in (b) indicates terrain height at sea level ($P_s$ = 1013 hPa) and elevated terrain height ($P_s$ = 813 hPa), respectively.**






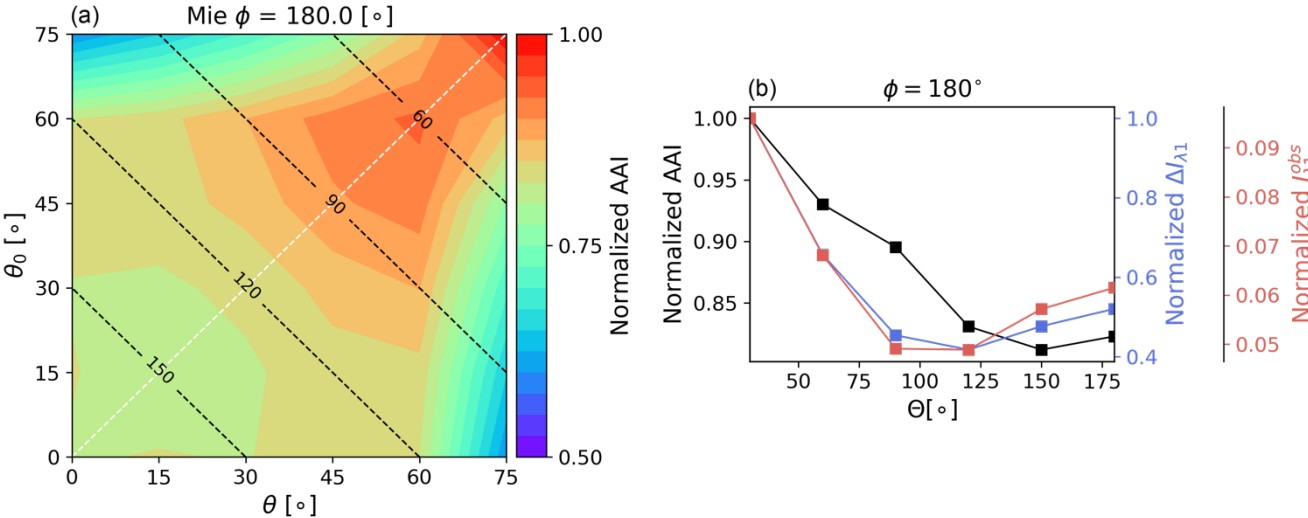


**Figure.6 AAI sensitivity to θ and θ₀ at φ=180°. The black dashed contour in (a) indicates the Θ=60°, 90°, 120°, 150°. The white dashed line in (a) indicates the cross sections, with its corresponding normalized AAI, $\Delta I_{\lambda 1}$ and $I_{\lambda 1}^{obs}$ in (b).**





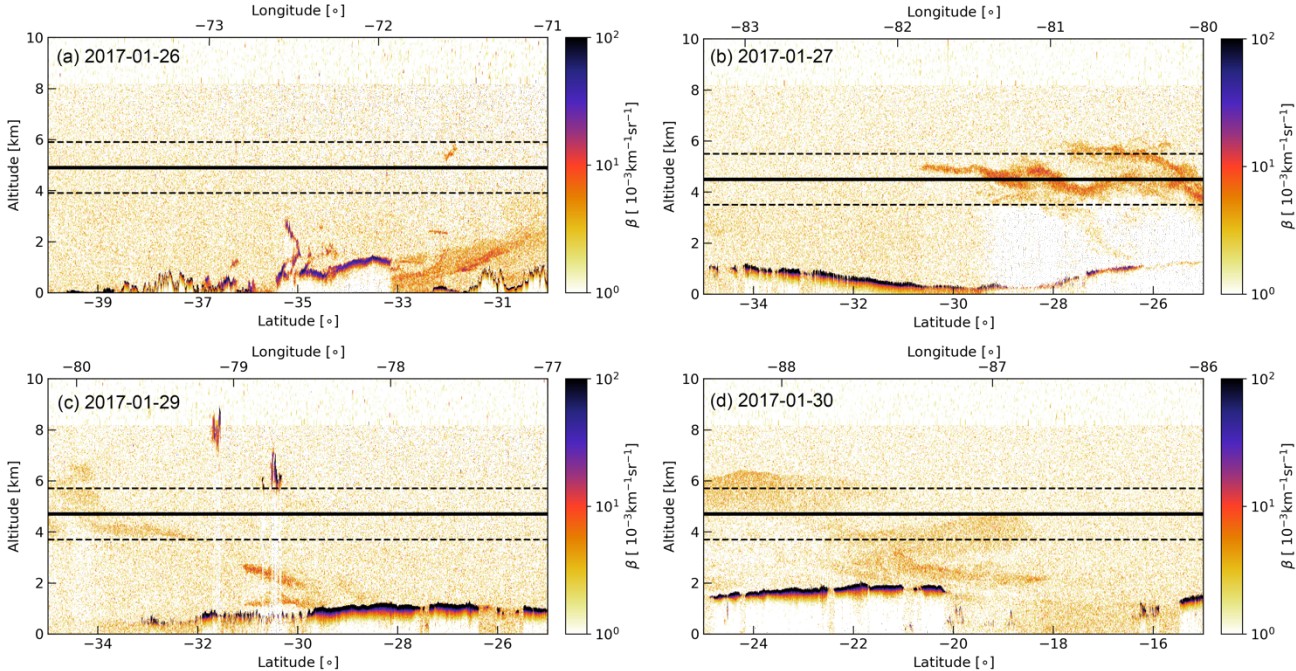

**Figure.7 CALIOP backscatter coefficient β at 532 nm. The solid and dashed line indicate the retrieved z$_{aer}$ and Δz, respectively. The red to black dots indicate clouds and the orange dots indicate aerosol layers, respectively.**



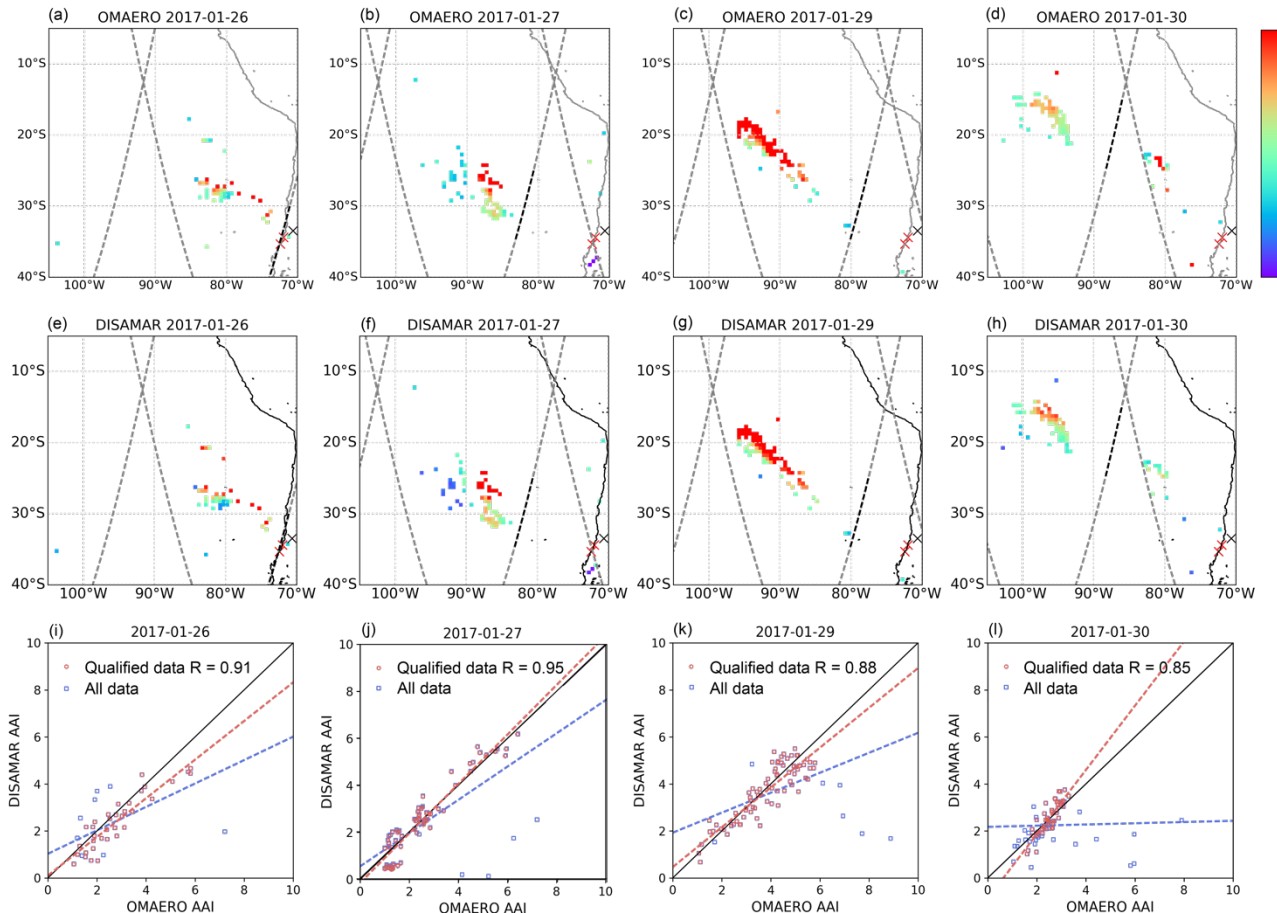

**Figure.8 AAI from OMI observations (a–d) and DISAMAR simulations (e–h) of the Chile wildfires on 26, 27, 29 and 30 January 2017. The black and red cross symbols are the AERONET station and the main fire sources (Pichilemu W34.39° S72.00° and Consititución S35.33°, W72.42°), respectively. The grey dashed line indicates the CALIOP paths in the region of interest, where the paths used to validate the plume height are marked by black dashed line. The scatter plots (i–l) present the OMI observations against DISAMAR simulations for only qualified data (red dot) and all data (blue dot), respectively.**

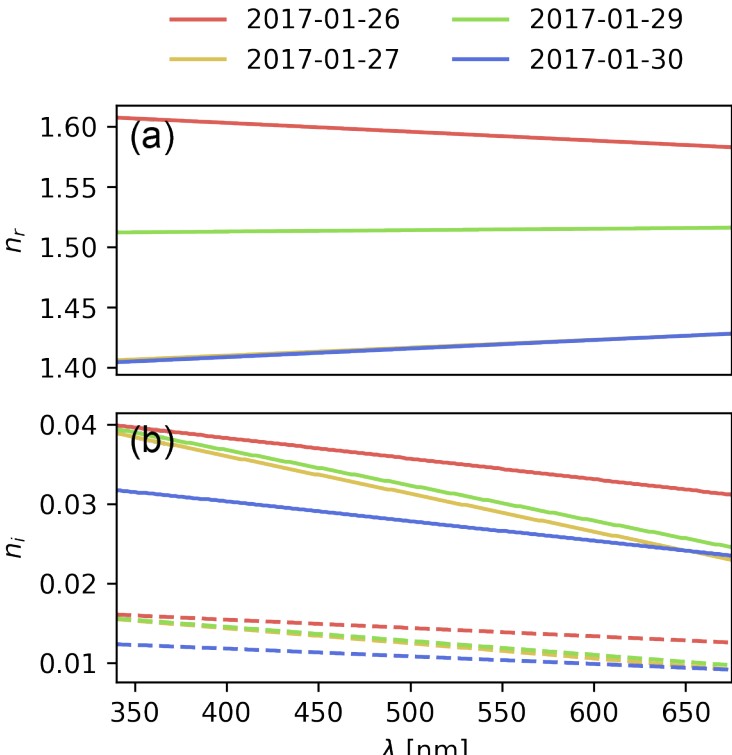

**Figure.9 Retrieved complex refractive index for each case: (a) $n_r$ and (b) $n_i$. The dashed line in lower panel is the wavelength dependent $n_i$ measured by AERONET.**

715

720

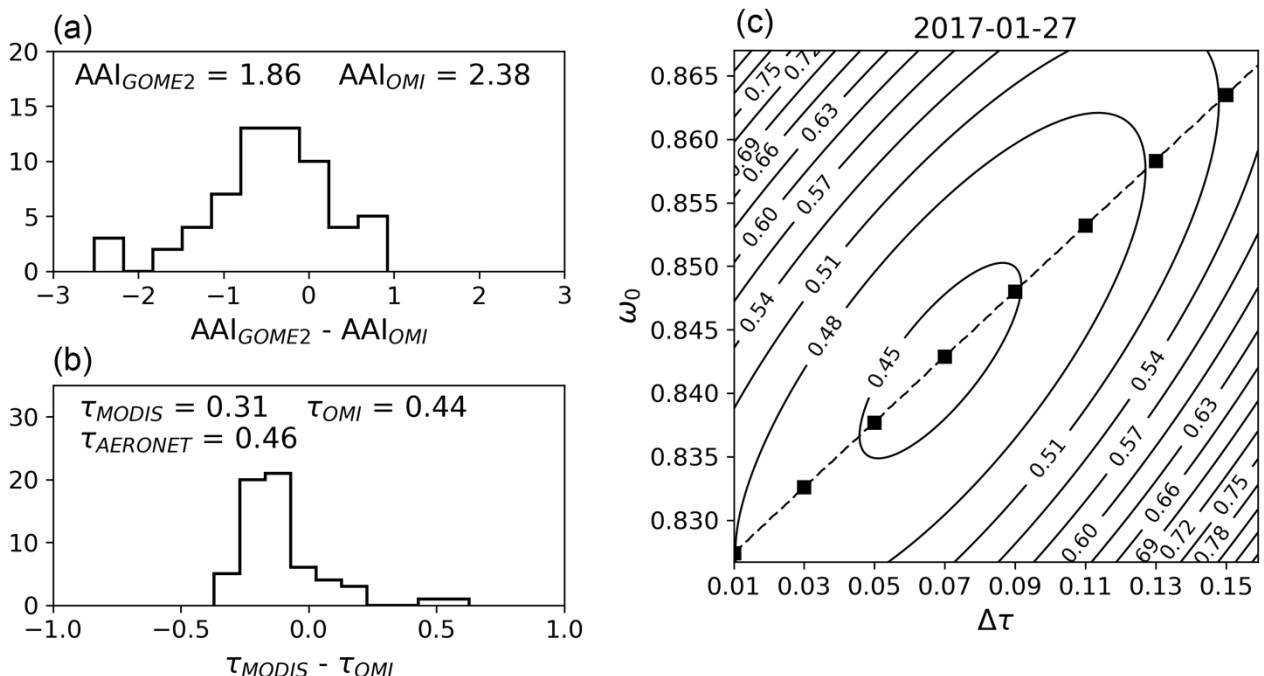

**Figure.10 Histogram of (a) the AAI difference between GOME-2 and OMI, against (b) the τ difference at 550 nm between MODIS and OMI for 27 January. Contour of (c) the AAI RMSE as a function of variation in τ and ω₀ for 27 January. The dashed line is the best estimation for each pair of Δτ and ω₀.**

**Table 1. Parameters used in sensitivity studies.**

| Parameters | Default value | Sensitivity range | Unit |
|---|---|---|---|
| Geometric mean radius ($r_g$) | 0.15 | 0.1, 0.15, 0.2, 0.25, 0.3, 0.35, 0.4 | μm |
| Geometric standard deviation ($\sigma_g$) | 1.5 | - | μm |
| Real refractive index ($n_r$) at 354 nm | 1.5 | 1.3, 1.35, 1.4, 1.45, 1.5 | - |
| Imaginary refractive index ($n_i$) at 354 nm | 0.06 | 0.04, 0.06, 0.08, 0.1 | - |
| Aerosol layer geometric central height ($z_{aer}$) | 4.5 | 2.5, 4.5, 6.5, 8.5 | km |
| Aerosol layer geometric thickness ($\Delta z$) | 1 | 0.5, 1, 1.5, 2 | km |
| Aerosol optical thickness ($\tau$) at 550 nm | 1 | 0.5, 1, 1.5, 2 | - |
| Surface albedo ($a_s$) | 0.05 | 0.05, 0.1, 0.5, 1.0 | - |
| Surface pressure ($P_s$) | 1013 | 1013, 963, 913, 863, 813 | hPa |
| Solar zenith angle ($\theta_0$) | 30 | 0, 15, 30, 45, 60, 75 | ° |
| Viewing zenith angle ($\theta$) | 0 | 0, 15, 30, 45, 60, 75 | ° |
| Relative azimuth angle ($\Delta\varphi = \varphi - \varphi_0 + 180°$) | 0 | 0, ±45, ±90, ±135, ±180 | ° |







**Table.2 Summary of retrieved results (after applying IQR outlier detection).**

| Date | | 2017-01-26 | 2017-01-27 | 2017-01-29 | 2017-01-30 |
|---|---|---|---|---|---|
| Number of pixels in the plume | | 44 | 70 | 82 | 75 |
| AAI | AAI median (OMAERO) | 2.52 | 2.38 | 4.05 | 2.61 |
| | AAI median (DISAMAR) | 2.17 | 2.48 | 3.81 | 2.49 |
| | Relative difference (%) | -13.88 | 4.20 | -5.93 | -4.60 |
| | RMSE | 0.67 | 0.51 | 0.60 | 0.41 |
| Aerosol profile parameters | $z_{aer}$ [km] | 4.9 | 4.5 | 4.7 | 4.7 |
| | $\Delta z$ [km] | 2 | | | |
| $n_i$ | $n_i$ at 354 nm | 0.0395 | 0.0382 | 0.0388 | 0.0314 |
| | $n_i$ at 388 nm | 0.0386 | 0.0366 | 0.0373 | 0.0306 |
| | $n_i$ difference between 354 and 388 nm | 2.33% | 4.37% | 4.02% | 2.61% |
| $\omega_0$ at 550 nm | $\omega_0$ (AERONET) | 0.89 | 0.89 | 0.92 | 0.91 |
| | $\omega_0$ (DISAMAR) | 0.83 | 0.81 | 0.87 | 0.85 |
| | Relative difference (%) | -6.74 | -8.99 | -5.43 | -6.59 |






**Appendix A**

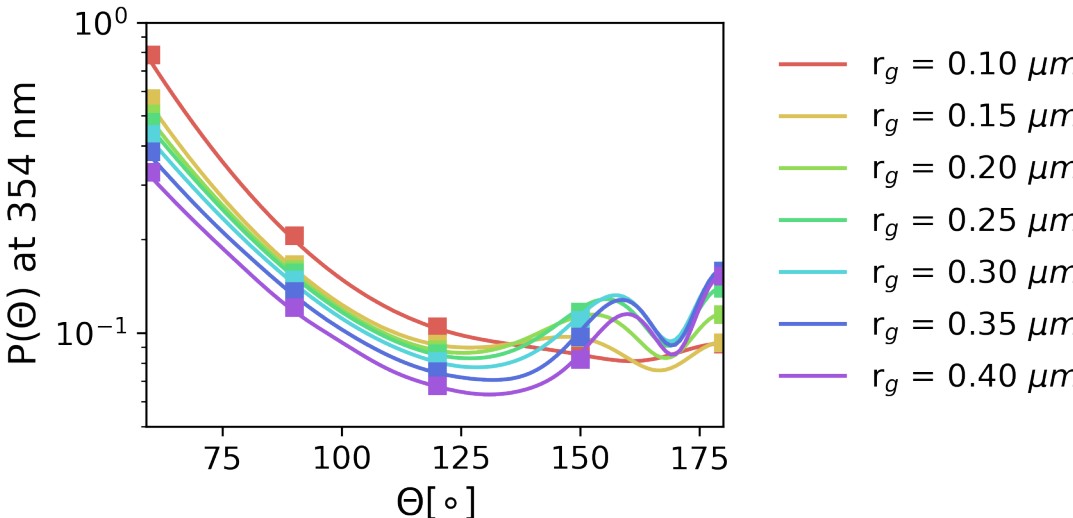

**Figure.A1: Phase function p(Θ) at 354 nm of the parameterized Mie scattering aerosol in sensitivity studies as a function of $r_g$ (with $n_r = 1.5$ and $n_i = 0.06$). The markers in the plot correspond to values when Θ=60°, 90°, 120°, 150°, 180°.**




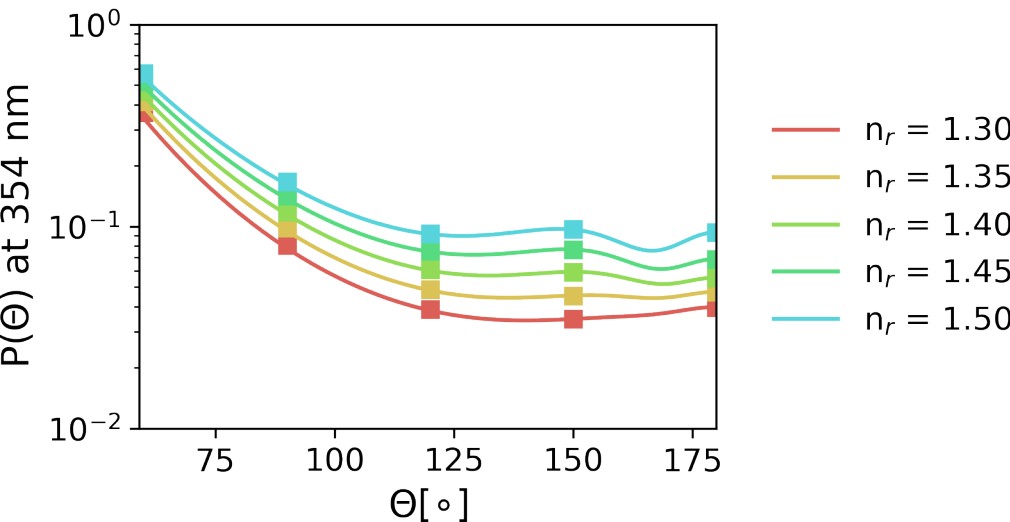

**Figure.A2: Phase function p(Θ) at 354 nm of the parameterized Mie scattering aerosol in sensitivity studies as a function of $n_r$ (with $r_g$ = 0.15 μm and $n_i$ = 0.06). The markers in the plot correspond to values when Θ=60°, 90°, 120°, 150°, 180°.**




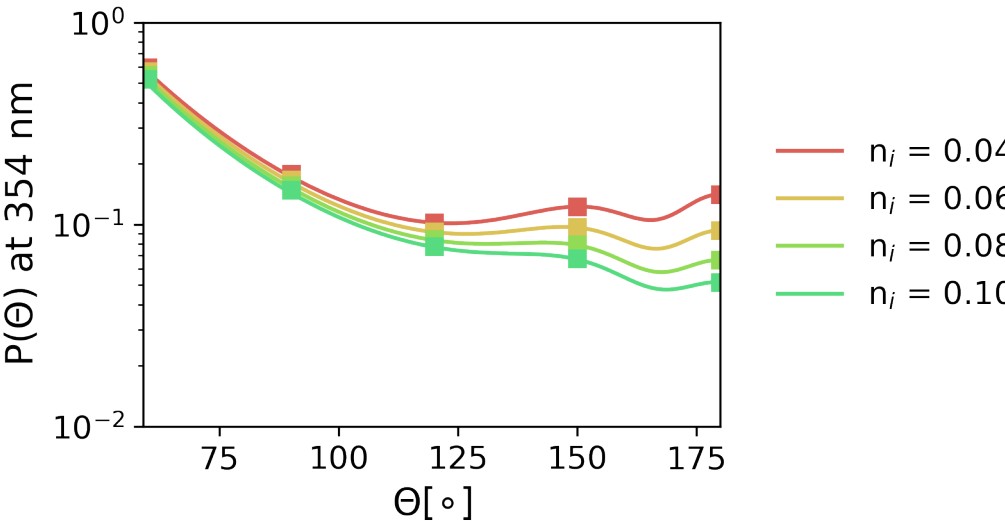

**Figure.A3: Phase function p(Θ) at 354 nm of the parameterized Mie scattering aerosol in sensitivity studies as a function of $n_i$**
**(with $r_g$ = 0.15 μm and $n_r$ = 1.5). The markers in the plot correspond to values when Θ=60°, 90°, 120°, 150°, 180°.**