# Peer review of "Quantifying the single scattering albedo for the January 2017 Chile wildfires from simulations of"

_Atmospheric Measurement Techniques, 2018_

## Referee Comment (RC1) · Anonymous Referee #1 · 5 Mar 2018

Review of the Sun et al paper.

Summary

This manuscript uses a case study to illustrate the derivation of the aerosol single scattering albedo using observations from the OMI sensor of large biomass burning event in south Chile in 2017. Although at first sight this manuscript proposes an interesting case study, it fails at many levels: it does not clearly explain what are the objectives to accomplish with this application and does not make clear what was achieved with it, the introduction has too many formal inaccuracies and poor explanations. Also there are missing explanations of the terms using in an equation, these terms are later used

to describe two figures rendering a significant part of the paper with not clear understanding. This study is not publication ready and my recommendation is to reject as it is. I do see value in this application though . The AAI is a parameter that deserves more attention and studies like these are excellent illustrations on how to use it. However, the manuscript fails to provide enough details and nuance. I would encourage a resubmission if more attention is given to the composition of the manuscript.

A few examples are provided below which by no means are the only ones that need to be corrected:

Abstract:

Line 20-21: a difference in SSA of 0.06 is very big as far as aerosol remote sensing and climate applications are concerned! It is not "slightly" smaller.

Also, missing what is the purpose of this study? And what are the conclusions that the reader should take out of this work?

Introduction:

Line 41: incorrect definition of SSA, it is not a ratio of radiation. This is too basic to be missed.

Line 43-44: No, POLDER does not measure the "aerosol polarized phase function". It measured polarized radiation that can be linked after modeling to the aerosol phase function.

Line 53-54: This is not entirely correct. Eck et al (2013) demonstrated that MODIS retrievals are impacted by the variation of SSA in smoke. (Eck, T. F., B. N. Holben, J. S. Reid, et al. 2013. "A seasonal trend of single scattering albedo in southern African biomass-burning particles: Implications for satellite products and estimates of emissions for the world's largest biomass-burning source." J. Geophys. Res. Atmos. 118 (12): 6414-6432 [10.1002/jgrd.50500] )

Line 60-61: not clear with what you mean that the AAI reduces the retrieval uncertainty. Uncertainty of what?

Line 66-67: this is a poorly phrased sentence and very confusing, the AAI in presence of aerosol is sensitive to aerosol height, SSA and concentration,... not SSA alone.

Overall introduction is too long and it does not make the case on what this study is important, what is unknown and what the objectives are.

Section 2.1

Line 98-99: incomplete/confusing sentence

Equation 2: what is the definition of deltaI(lambda)? , This equation is different than what other groups use as definition of AAI. Can you provide a reference where this equation is derived? It seems to me a minus sign is missing, also not clear where the delta is coming from? For example de Graaf et al., (2007) uses the standard definition of AAI. how does you equation related with the more commonly used equations? (de Graaf, M., P. Stammes, and E. A. A. Aben (2007), Analysis of reflectance spectra of UV-absorbing aerosol scenes measured by SCIAMACHY, J. Geophys. Res., 112, D02206, doi:10.1029/2006JD007249.)

Line 117: linear interpolation of what? What parameters are being interpolated? Please explain.

Line 123: Aeronet Phase function data is not reported at 354nm , where does this come from? As it is , this is not correct.

Figures 3 and 4 cannot be interpreted because deltaI has not been explained/defined.

No further reading of the manuscript since what I found until here warrants a rejection.

Technical comments:

Line 15: what max value was observed? Line 18 : what measurements/obs you are

referring to? Radiances? Line 18-20: not clear what you want to say in the sentence starting with "Due to the relatively..." Line 20-21: are these SSA values averages over the plume or specific pixels? Line 33: replace "bright surfaces" with "snow", I am assuming this is what you meant. Line 82-83 : what are the locations of Pichilemu and Constitution? Are those forests ? Cities? regions? Please provide more details of the geographical setting. Was there a drought?

Line 84: figure 1 has very poor contrast when printed, please correct.

---

## Referee Comment (RC2) · Anonymous Referee #2 · 13 Apr 2018

Manuscript amt-2018-40.

Quantifying the single scattering albedo for the January 2017 Chile wildfires from simulations of the OMI absorbing aerosol index, by Jiyunting Sun J. Pepijn Veefkind, Peter van Velthoven1, Pieternel F. Levelt.

Reviewer comments.

The manuscript presents technique to retrieve single scattering albedo (SSA) and parameterized aerosol vertical profile from OMI absorbing aerosol index (AAI).The approach first models AAI using external information on aerosol optical and microphysical parameters and then matches simulated and observed AAI by varying SSA and aerosol vertical profile.

The manuscript presents sensitivity of AAI to aerosol complex index of refraction, particle size, and vertical distribution along with surface reflection, surface elevation and OMI viewing geometry. The conditions for optimal OMI observation geometry are outlined.

The technique is applies to Chile wildfire biomass burning event on January 2017 utilizing MODIS aerosol optical depth (AOD), AERONET complex index of refraction and particle size. The result are compared to AERONET retrieved SAA and the closest CALIOP profile of aerosol backscattering.

I believe that manuscript is clearly in scope of AMT and can be published after the following comments will be answered.

1. On page 6 authors write: "the fine and coarse mode are derived separately from AERONET". This is not true. AERONET inversion algorithm primarily retrieves aerosol size distribution (ASD) in 22 discrete radius points. The separation in two modes is done after the inversion by finding inflection point in between two ASD peaks and then approximation each part of ASD by log-normal distribution. The parameters of these approximations are provided as ASD parameters for fine and coarse modes. The detailed description of this procedure can be find at https://aeronet.gsfc.nasa.gov/new_web/Documents/Inversion_products_V2.pdf. Therefore the averaging of SSA of two modes is not needed. The total SSA for the initially retrieved ASD (at 22 points) can be used instead. I am wondering how close averaged SSA to the total SSA is provided by AERONET.
In addition, SSA for fine and coarse modes are not advised to use because retrieval in implemented under assumption that complex index of refraction is the same for all the sizes.
2. On page 1 authors mention that retrieved SSA (0.84) is **slightly** lower that AERONET value. I encourage authors to replace or remove world slightly because the absolute difference 0.06 is significant in terms of radiative forcing estimation.

3. Page 2. "The foremost advantage of the AAI is its independence from assumptions on aerosol types, which significantly reduce the retrieval uncertainty". This statement is confusing because the simulation of AAI is still dependent on aerosol type.
4. Page 3. What wavelength interpolation is used for? Index of retraction or aerosol optical parameters?

---

## Editor Comment (EC1) · H. Jethva (Editor) · 25 May 2018

Dear Author,

Your manuscript # amt-2018-40 has received comments and suggestions from two anonymous reviews for which you have just submitted a response addressing their concerns. Thank you. Along the review process, I have read your paper and gave some thoughts on the research content. As I understood, the concept of the retrieving aerosol SSA in the near-UV region by constraining AOD quite resembles to the methods presented in Satheesh et al. (2008) and Gassó and Torres (2016). Both earlier studies, and the results of your own paper demonstrate that it is possible to retrieve

aerosol SSA and layer height from the color ratio information (UVAI) in the near-UV given the AOD as an a priori and a pre-defined aerosol model. It was a surprise that your paper completely misses to discuss and cite these two papers relevant to the present study.

In this inversion process, your study employs an aerosol model developed using the AERONET data. It was not clear in the Methodology section what kind of spectral dependence is assumed between the two near-UV wavelengths. A number of studies, using laboratory measurements and satellite data, have shown that the carbonaceous aerosols generated from the open field agricultural burning and wildfires exhibit a strong wavelength-dependent of absorption (imaginary part of the refractive index) in the near-UV region. In our earlier study (Jethva and Torres, 2011, ACT), we learned that the relative spectral dependence in the imaginary part of the refractive index needs to be increased to 20% with reference to no spectral dependence (equivalent to black carbon) in order to obtain a better agreement in AOD and SSA between OMI and AERONET. A strong wavelength dependence in aerosol absorption in the UV has been shown as a proxy for the amount of organics in the biomass burning smoke.

The color ratio (UVAI) information in the near-UV strongly varies with the Absorption Angstrom Exponent-a parameter that quantifies the spectral dependence of aerosol absorption. Since your paper lacks to provide this essential information, it is hard to interpret the SSA retrievals presented in the paper. Perhaps, a significant disagreement of ~0.06 in SSA between the OMI retrievals and that of AERONET reflects the issue described above that you should consider while revising your manuscript. Please verify the relative spectral dependence in absorption assumed in the aerosol model and discuss/mention in the revision.

Given below are the citations relevant to your manuscript.

Satheesh, S. K., Torres, O., Remer, L. A., Babu, S. S., Vinoj, V., Eck, T. F., Kleidman, R. G., and Holben, B. N.: Improved assessment of aerosol absorption using OMI-MODIS

joint retrieval, J. Geophys. Res., 114, D05209, doi:10.1029/2008JD011024, 2009.

Jethva, H. and Torres, O.: Satellite-based evidence of wavelength-dependent aerosol absorption in biomass burning smoke inferred from Ozone Monitoring Instrument, Atmos. Chem. Phys., 11, 10541-10551, https://doi.org/10.5194/acp-11-10541-2011, 2011.

Gassó, S. and Torres, O.: The role of cloud contamination, aerosol layer height and aerosol model in the assessment of the OMI near-UV retrievals over the ocean, Atmos. Meas. Tech., 9, 3031-3052, https://doi.org/10.5194/amt-9-3031-2016, 2016.

— Dr. Hiren Jethva Associate Editor (AMT) Research Scientist, USRA/GESTAR, NASA Goddard Space Flight Center Code 614 Greenbelt, MD, USA 20771 E-mail: hiren.t.jethva@nasa.gov http://science.gsfc.nasa.gov/sed/bio/hiren.t.jethva

---

## Author Comment (AC1) · 25 May 2018

**This document contains the responses to the first referee, followed by a version 3 manuscript (starts in page 8). The reviewers' comments and questions are in bold. For each comment / question, the authors' reply / answer is in black, and the corresponding modifications in the manuscript version 3 are marked in blue colour.**

**(1) Author's response to Anonymous Referee #1**

Thank you very much for your comments to our manuscript. The introduction has been trimmed. We also have revised the entire manuscript. The following are the responses to your specific comments.

**Line 20-21: a difference in SSA of 0.06 is very big as far as aerosol remote sensing and climate applications are**
**concerned! It is not "slightly" smaller.**

We admit that a difference 0.06 is not a small value for SSA and we have deleted this word accordingly. This statement has been rephrased into The retrieved mean $\omega_0$ at 550 nm for the entire plume over the period from 26-30 January 2017 varies from 0.81 to 0.87, whereas the nearest AERONET station reported values in the range from 0.89 to 0.92 (line 21 - 23).

But it is comparable with previous research. Hu et al. (2007) used TOMS AAI to retrieve SSA, their analytical uncertainty is 15%. For the typical AOD level in our case (0.3 ~ 0.7), the SSA uncertainty is 0.02 ~ 0.06 according to Hu et al. (2007) research, which matches with our results.

Furthermore, as we mentioned in the abstract and stressed throughout the manuscript, the AERONET site and the plume we defined are not collocated (the AERONET site is in the city centre as mention in line 200, which potentially
overestimates the SSA). This makes this kind of retrieving to be validate. Other concerns, such as the lack of aerosol layer height, uncertainty in MODIS AOD and AERONET itself, should also be considered.

**Also, missing what is the purpose of this study? And what are the conclusions that the reader should take out of this work?**

This application attempts to quantify the aerosol absorption by retrieving SSA from satellite measured AAI. The conclusion is satellite retrieved AAI is a useful parameter to constrain forward simulation and to derive SSA. Although currently we have a difference of 0.06 compared with AERONET, this discrepancy can be interpreted by the uncertainties in the inputs and AERONET itself as well as difference in measurement techniques (i.e. satellite vs ground-based measurements).

The purpose is presented in the last paragraph in Section 1 (line 59 - 61).

**Line 41: incorrect definition of SSA, it is not a ratio of radiation. This is too basic to be missed.**

Thank you for the correction on the SSA definition.

We have rephrased to $\omega_0$ is defined as the ratio of the aerosol scattering over the extinction (line 33).

**Line 43-44: No, POLDER does not measure the "aerosol polarized phase function". It measured polarized radiation that can be linked after modeling to the aerosol phase function.**

Thank you for the correction.

We have rephrased to More advanced sensors, such as the POLarization and Directionality of the Earth's Reflectances
(POLDER), can retrieve $\omega_0$ from a combination of multi-angular, multi-spectral observations of the polarized radiation (line 36-38).

**Line 53-54: This is not entirely correct. Eck et al (2013) demonstrated that MODIS retrievals are impacted by the variation of SSA in smoke. (Eck, T. F., B. N. Holben, J. S. Reid, et al. 2013. "A seasonal trend of single scattering albedo in southern African biomass-burning particles: Implications for satellite products and estimates of emissions for the world's largest biomass-burning source." J. Geophys. Res. Atmos. 118 (12): 6414-6432 [10.1002/jgrd.50500])**

It may be a misunderstanding. We are not saying aerosol absorption has no effect on AOD, we are just saying AOD is less sensitive to aerosol absorption. AOD could be large under either a very scattering case or a very absorbing case.

The reference as you mentioned here states the effect of retrieving AOD from a constant pre-assumed SSA, and this effect is presented as the systematic bias of retrieved MODIS AOD from AERONET AOD. This statement stresses more on the AOD retrieval bias is sensitive to SSA, rather than AOD itself.

Because the major revision in the Section 1 Introduction part, this sentence is no longer available.

**Line 60-61: not clear with what you mean that the AAI reduces the retrieval uncertainty. Uncertainty of what?**

Using AAI, instead of AOD to constrain the inversion of aerosol properties retrieval can reduce the uncertainty of retrieved aerosol parameters. For AAI, the uncertainties come from the measured reflectance. But for AOD, the uncertainties come from both the measured reflectance and pre-assumed aerosol types.

Because the major revision in the Section 1 Introduction part, this sentence is no longer available. The corresponding content is in line 48 – 49: The most important advantage of the satellite retrieved AAI is that it does not dependent on assumptions on aerosol types, while a-prior aerosol types are major uncertainties in aerosol parameter retrievals, such as $\tau$.

**Line 66-67: this is a poorly phrased sentence and very confusing, the AAI in presence of aerosol is sensitive to aerosol height, SSA and concentration,... not SSA alone.**

Sorry for the confusion. But we do not mean that AAI is only sensitive to SSA, just in terms of SSA, AAI is more sensitive than AOD.

Because the major revision in the Section 1 Introduction part, this sentence is no longer available. The corresponding content is in line 53: Moreover, the near-UV AAI is by definition highly sensitive to $\omega_0$.

**Line 98-99: incomplete/confusing sentence**

Sorry for the confusion.

We have rephrased to The basic idea of the residue method is that in a pure Rayleigh atmosphere, the reflectance (or equivalently the radiance ($I_\lambda$)) decreases strongly with the wavelength (line 79-80).

**Equation 2: what is the definition of deltaI(lambda)? , This equation is different than what other groups use as definition of AAI. Can you provide a reference where this equation is derived? It seems to me a minus sign is missing, also not clear where the delta is coming from? For example de Graaf et al., (2007) uses the standard definition of AAI. how does you equation related with the more commonly used equations? (de Graaf, M., P. Stammes, and E. A. A. Aben (2007), Analysis of reflectance spectra of UV- absorbing aerosol scenes measured by SCIAMACHY, J. Geophys. Res., 112, D02206, doi:10.1029/2006JD007249.)**

We have added the definition of $\Delta I_{\lambda 1}$ in line 88. We also have added that AAI calculation assumes a Rayleigh atmosphere at $\lambda_2$, $I_{\lambda 2}^{Ray}(a_s) = I_{\lambda 2}^{obs}$ (Herman et al., 1997) in line 87. The derivation from Eq.(1) to Eq.(2) is then not difficult (line 83). If it is still not clear to you, here is the derivation procedure:

According to the AAI definition:

$$AAI = -100\left(log_{10}\left(\frac{I_{\lambda 1}}{I_{\lambda 2}}\right)^{obs} - log_{10}\left(\frac{I_{\lambda 1}}{I_{\lambda 2}}\right)^{Ray}\right) \tag{1}$$

, which can be re-written into:

$$AAI = -100(log_{10}(I_{\lambda 1})^{obs} - log_{10}(I_{\lambda 2})^{obs} - log_{10}(I_{\lambda 1})^{Ray} + log_{10}(I_{\lambda 2})^{Ray}) \tag{2}$$

, the Rayleigh radiance is calculated by a surface albedo that satisfies $(I_{\lambda 2})^{obs} = (I_{\lambda 2})^{Ray}$, then Eq.(2) can be re-written into:

$$AAI = -100(log_{10}(I_{\lambda 1})^{obs} - log_{10}(I_{\lambda 1})^{Ray}) \tag{3}$$

, reformed into:

$$AAI = 100(log_{10}(I_{\lambda 1})^{Ray} - log_{10}(I_{\lambda 1})^{obs}) \tag{4}$$

$$AAI = 100 log_{10}\left(\frac{(I_{\lambda 1})^{Ray}}{(I_{\lambda 1})^{obs}}\right) \tag{5}$$

, here we define $(I_{\lambda 1})^{Ray} = (I_{\lambda 1})^{obs} + \Delta I_{\lambda 1}$, where $\Delta I_{\lambda 1}$ can be explained as the change of radiance spectral dependency between a Rayleigh atmosphere and an observed atmosphere. Under cloud-free condition, the presence of absorbing aerosols lead to a positive $\Delta I_{\lambda 1}$. The definition of $\Delta I_{\lambda 1}$ is also mentioned at line 105 in the manuscript. Then Eq.(5) can be re-written into:

$$AAI = 100 log_{10}\left(\frac{(I_{\lambda 1})^{obs} + \Delta I_{\lambda 1}}{(I_{\lambda 1})^{obs}}\right) \tag{6}$$

$$AAI = 100 log_{10}\left(\frac{\Delta I_{\lambda 1}}{(I_{\lambda 1})^{obs}} + 1\right) \tag{7}$$

, which is the Eq.(2) in the manuscript.

**Line 117: linear interpolation of what? What parameters are being interpolated? Please explain.**

Linear interpolation of complex refractive index over spectral range from 340 to 675 nm.

We have rephrased into part into We obtain the size distribution function and complex refractive index at 440, 675, 880 and 1018 nm from AERONET, and apply the linear interpolation / extrapolation to derive the complex refractive index over the spectrum from 340 to 675 nm, with spectral resolutions of 2 nm. Then DISAMAR uses above information to calculate the aerosol phase function P(Θ) and ω0 over the full spectrum (line 103-106).

**Line 123: Aeronet Phase function data is not reported at 354nm , where does this come from? As it is , this is not correct.**

The phase function at 354 nm is calculated by the radiative transfer model DISAMAR with AERONET constraints. We took size distribution function, and complex refractive index at 440, 675, 870 and 1018 nm from AERONET. We used linear extrapolation method to extend the spectrum refractive index to 340 nm. Then the radiative transfer model used that information to calculate the phase function and SSA over the full spectrum (those are intermediate outputs). That is how the phase function and SSA at 354 nm comes. With those intermediate outputs (that carry information on aerosol types), DISAMAR can execute forward simulation of AAI.

We have rephrased this part into We obtain the size distribution function and complex refractive index at 440, 675, 880 and 1018 nm from AERONET, and apply the linear interpolation / extrapolation to derive the complex refractive index over the spectrum from 340 to 675 nm, with spectral resolutions of 2 nm. Then DISAMAR uses above information to calculate the aerosol phase function $P(\Theta)$ and $\omega_0$ over the full spectrum (line 103-106).

**Figures 3 and 4 cannot be interpreted because deltaI has not been explained/defined. No further reading of the manuscript since what I found until here warrants a rejection.**

Please refer to the derivation of Eq (2) $(I_{\lambda 1})^{Ray} = (I_{\lambda 1})^{obs} + \Delta I_{\lambda 1}$ in previous or line 88 in the manuscript.

**Line 15: what max value was observed?**

To your question, the maximum AAI observed by OMI for Chile 2017 wildfires over all the pixels of the 4 days is 5.80. The maximum median value is 4.05 and obtained on 29 January 2017.

**Line 18: what measurements/obs you are referring to? Radiances?**

We are referring to the CALIOP backscatter coefficient measurements.

We have rephrased into The simulated plume ascends to an altitude of 4.5-4.9 km, which is in good agreement with available CALIOP backscatter coefficient measurements (line 18-19).

**Line 18-20: not clear what you want to say in the sentence starting with "Due to the relatively..."**

To your question, the OMI observation is sparsely distributed which may contains geographical outliers that may not be the plume even it has AAI value larger than 1. Therefore, we applied an additional data quality control procedure with interquartile range technique. That is, calculate the difference between simulated and observed AAI, and remove the pixels that have AAI difference outside the interquartile range. This is detailed described in line 250-260.

We have rephrased into Due to the heterogeneity of the data that may contain the pixels outside the plume, an outlier detection criterion has to be applied (line 19-20).

**Line 20-21: are these SSA values averages over the plume or specific pixels?**

To your question, the retrieved SSA is the mean value for the entire plume over the period from 26-30 January 2017.

We have rephrased to The retrieved mean $\omega_0$ at 550 nm for the entire plume over the period from 26-30 January 2017 varies from 0.81 to 0.87, whereas the nearest AERONET station reported values in the range from 0.89 to 0.92 (line 21-23).

**Line 33: replace "bright surfaces" with "snow", I am assuming this is what you meant.**

Thank you for correction.

Because the major revision in the Section 1 Introduction part, this sentence is no longer available.

**Line 82-83: what are the locations of Pichilemu and Constitution? Are those forests ? Cities? regions? Please provide more details of the geographical setting. Was there a drought?**

The location of Pichilemu and Consititución are two cities at the central of Chile as mentioned in line 62. The local forestry industry (pine and eucalyptus) contributed a large fraction of the fire source in line 64. There was a drought as mentioned in line 61. All this information was actually mentioned in the version 2 manuscript.

**Line 84: figure 1 has very poor contrast when printed, please correct.**
The original plot is from NASA. It has been adjusted now (line 515).

[revised manuscript text omitted]
$ in the UV band are linearly extrapolated using available data between 440 and 675 nm as mentioned in Section 2.2. Finally, the AERONET retrieved $\tau$ and $\omega_0$ is also linearly interpolated to 550 nm.

**3.1.4 CALIOP backscattering coefficient**

The CALIOP on-board CALIPSO, which was launched in 2006, provides high-resolution profiles of aerosols and clouds. It has three channels with one measuring the backscattering intensity at 1064 nm and the rest measuring orthogonally polarized components at 532 nm backscattering intensity (Winker and Omar, 2006). Due to the limited spatial coverage, CALIOP did not observe the Chile plume for all the cases for which we have OMI observations. We only use the total attenuated backscatter at 532 nm from level 1B Version 4.10 Standard data to evaluate the parameterized aerosol profiles (https://eosweb.larc.nasa.gov/project/calipso).

**3.2 Methodology**

In this study, we employ the radiative transfer model DISAMAR to simulate the near-UV AAI from OMI and to derive the $\omega_0$ for a specific case, i.e. the Chile wildfires in January 2017. We select the period from 26 to 30 January 2017 (28 January is excluded due to lack of data) when the AAI value reached its peak. The aerosol information consists of the cloud free column $\tau$ retrieved from MODIS, and the aerosol micro-physical parameters ($r_g$, $n_r$ and $n_i$) retrieved from AERONET. The real part of the refractive index $n_r$ in the UV band has a fixed value which is obtained by linearly extrapolating that from the AERONET observations at 440 to 675 nm assuming a small wavelength dependency of $n_r$. We set the imaginary part $n_i$ as a free parameter to vary $\omega_0$, with an initial guess value obtained by extrapolation from AERONET like $n_r$.

[revised manuscript text omitted]

We test the retrieval of $\omega_0$ for the wildfires happening in central Chile in January 2017. After filtering the data from outliers, the high spatial correlation coefficients (0.85 to 0.95) between the simulated and observed AAI proves its necessity and
effectiveness. The retrieved aerosol profiles indicate the plume was elevated to height of 4.5-4.9 km during the research period. These results are in agreement with CALIOP measurements. This average of the retrieved $\omega_0$ at 550 nm is approximately 0.84, which is 0.06 lower than that of AERONET retrieval. The retrieved $\omega_0$ is reasonable if one takes into account the typical uncertainty in the $\omega_0$ retrieved from AERONET ($\pm$0.03). The remaining discrepancy is probably caused by the location of the AERONET site; the assumption of homogeneous and static plume properties, which ignores the plume
evolution over space and time; the simplified parameterization of the aerosol profile; and the observational errors in the input aerosol micro-physics, $\tau$, as well as AAI. We quantitatively analyse the uncertainty of $\tau$ for a specific case (27 January) when the estimated aerosol profile is in good agreement with the CALIOP measurements.

This study proves the potential of utilizing OMI measured AAI to quantitatively characterize aerosol optical properties like $\omega_0$. Even though without direct observation of aerosol profiles, this parameter can be retrieved with quite confidence.

However, apart from the observational uncertainties, the current study is probably limited by the necessary assumptions of homogeneous and static plume properties to some extent, whose impact on retrieved $\omega_0$ is difficult to quantify. In the future planned work, climatological data is expected to describe the evolution of the plume properties in space and time.

[revised manuscript text omitted]

---

## Author Comment (AC2) · 25 May 2018

This document contains the responses to the second referee, followed by a version 3 manuscript (starts in page 8). The reviewers' comments and questions are in **bold**. For each comment / question, the authors' reply / answer is in black, and the corresponding modifications in the manuscript version 3 are marked in blue colour.

**5 (2) Author's response to the Anonymous Referee #2**

Thank you very much for your comments on our manuscripts. The following are the responses to your comments.

1. On page 6 authors write: "the fine and coarse mode are derived separately from AERONET". This is not true. AERONET inversion algorithm primarily retrieves aerosol size distribution (ASD) in 22 discrete radius

- 10 points. The separation in two modes is done after the inversion by finding inflection point in between two ASD peaks and then approximation each part of ASD by log-normal distribution. The parameters of these approximations are provided as ASD parameters for fine and coarse modes. The detailed description of this procedure can be find at https://aeronet.gsfc.nasa.gov/new web/Documents/Inversion products V2.pdf. Therefore the averaging of SSA of two modes is not needed. The total SSA for the initially retrieved ASD (at 22
- 15 points) can be used instead. I am wondering how close averaged SSA to the total SSA is provided by AERONET. In addition, SSA for fine and coarse modes are not advised to use because retrieval in implemented under assumption that complex index of refraction is the same for all the sizes. Sorry for the confusion. We are not indicating that the AERONET derives fine and coarse mode separately, but indicating that we obtained the two modes by finding the peaks in the size distribution function provided by
- 20 AERONET.

The radiative transfer model DISAMAR currently cannot directly apply bi-modal distribution function. Instead, we built two modes individually, corresponding to the two modes taken from AERONET. For each mode, we assigned the optical properties (refractive index) provided by AERONET. Then DISAMAR calculates the phase function and SSA for each mode. Finally, we combined two modes into one weighted by number density fraction and extinction cross

25 section.

> We have rephrased the sentence into The fine and coarse mode particle size are derived by finding the two peaks of the log-normal distribution function provided by AERONET. The complex refractive index is assumed the same for both modes. Since bi-modal aerosol is not applicable in DISAMAR yet, we first calculate optical properties of two modes individually, then we externally combine the optical properties of two modes into a bi-modal aerosol with a fraction (line 210-213).

30

**2. On page 1 authors mention that retrieved SSA (0.84) is slightly lower that AERONET value. I encourage authors to replace or remove word slightly because the absolute difference 0.06 is significant in terms of radiative forcing estimation.**

Thank you for your suggestion. We admit that 0.06 difference for SSA retrieval is not minor, and 'slightly' is not 35 properly used.

We have deleted 'slightly' accordingly, and rephrased into The retrieved mean  $\omega_0$  at 550 nm for the entire plume over the period from 26-30 January 2017 varies from 0.81 to 0.87, whereas the nearest AERONET station reported values in the range from 0.89 to 0.92 (line 21-23).

**3. Page 2. "The foremost advantage of the AAI is its independence from assumptions on aerosol types, which significantly reduce the retrieval uncertainty". This statement is confusing because the simulation of AAI is still dependent on aerosol type.**

Sorry for the confusion. Indeed, AAI forward simulation (like what we do in this study) needs aerosol information,

- 45 including aerosol loading, profile, micro-physics, etc. But AAI retrieval from satellite is independent of pre-assumed aerosol types. It is calculated directly from the measured radiance (Eq.(1)). Here we want to stress that the AAI retrieval is independent of aerosol information itself, thus there is fewer uncertainties in the retrieved AAI. We have rephrased this sentence into The most important advantage of the satellite retrieved AAI is that it does not dependent on assumptions on aerosol types, while a-prior aerosol types are major uncertainties in aerosol parameter
- 50 retrievals, such as  $\tau$  (line 48-49).

**4. Page 3. What wavelength interpolation is used for? Index of retraction or aerosol optical parameters?**

To your question, we used linear interpolation to interpolate the AERONET refractive index from 440 to 1080 nm, and used the same technique to extrapolate the spectrum to 340 nm. With wavelength dependent size distribution function

- 55 and refractive index, the radiative transfer model (DISAMAR) calculates the phase function and SSA for the full spectrum we specified (340 to 675 nm, with resolution of 2 nm) (those are intermediate outputs). With those intermediate outputs (that carry information on aerosol types), DISAMAR can execute the forward simulation of AAI. We have rephrased this part into We obtain the size distribution function and complex refractive index at 440, 675, 880 and 1018 nm from AERONET, and apply the linear interpolation / extrapolation to derive the complex refractive index
- 60 over the spectrum from 340 to 675 nm, with spectral resolutions of 2 nm (line 103-105). This is also mentioned in line 223-225 where we applied the same method for the case study: To constrain the spectral dependency of optical properties in the near-UV band, complex refractive index nr and ni in the UV band are linearly extrapolated using available data between 440 and 675 nm as mentioned in Section 2.2.
- In line 225-226 we mentioned the treatment for AERONET AOD and SSA (we use them for evaluating our results): 65 Finally, the AERONET retrieved  $\tau$  and  $\omega_0$  is also linearly interpolated to 550 nm.

70

[revised manuscript text omitted]
$  in the UV band are linearly extrapolated using available data between 440 and 675 nm as mentioned in Section 2.2. Finally, the AERONET retrieved  $\tau$  and  $\omega_0$  is also linearly interpolated to 550 nm.

**305 3.1.4 CALIOP backscattering coefficient**

The CALIOP on-board CALIPSO, which was launched in 2006, provides high-resolution profiles of aerosols and clouds. It has three channels with one measuring the backscattering intensity at 1064 nm and the rest measuring orthogonally polarized components at 532 nm backscattering intensity (Winker and Omar, 2006). Due to the limited spatial coverage, CALIOP did not observe the Chile plume for all the cases for which we have OMI observations. We only use the total attenuated

[revised manuscript text omitted]

430 This study proves the potential of utilizing OMI measured AAI to quantitatively characterize aerosol optical properties like  $\omega_0$ . Even though without direct observation of aerosol profiles, this parameter can be retrieved with quite confidence. However, apart from the observational uncertainties, the current study is probably limited by the necessary assumptions of homogeneous and static plume properties to some extent, whose impact on retrieved  $\omega_0$  is difficult to quantify. In the future planned work, climatological data is expected to describe the evolution of the plume properties in space and time.

[revised manuscript text omitted]

Nakajima, T., Tonna, G., Rao, R., Boi, P., Kaufman, Y. and Holben, B.: Use of sky brightness measurements from ground

- for remote sensing of particulate polydispersions, Appl. Opt., 35, 2672–2686, doi:10.1364/AO.35.002672, 1996.
   NASA.gov: NASA's Terra Catches Fires in Central Chile, [online] Available from: https://www.nasa.gov/image-feature/goddard/2017/nasas-terra-catches-fires-in-central-chile (Accessed 1 May 2017), 2017.
   Petters, J. L., Saxena, V. K., Slusser, J. R., Wenny, B. N. and Madronich, S.: Aerosol single scattering albedo retrieved from measurements of surface UV irradiance and a radiative transfer model, J. Geophys. Res., 108, 4288,
- 530 doi:10.1029/2002JD002360, 2003.

Ramanathan, V. and Carmichael, G.: Global and regional climate changes due to black carbon, Nat. Geosci., 1, 221–227, doi:10.1038/ngeo156, 2008.

Reid, J. S., Eck, T. F., Christopher, S. A., Koppmann, R., Dubovik, O., Eleuterio, D. P., Holben, B. N., Reid, E. A. and Zhang, J.: A review of biomass burning emissions part III: intensive optical properties of biomass burning particles, Atmos.

Chem. Phys. Discuss., doi:10.5194/acpd-4-5201-2004, 2004.
Remer, L.A., Kaufman; Y.J., Tanré; D., Mattoo, S., Chu, D.A., Martins, J.V., Li, R.R., Ichiku, C., Levy, R.C., Kleidman, R.G., Eck, T.K., Vermote, E. and Holben, B. N.: The MODIS Aerosol Algorithm, Products, and Validation, J. Atmos. Sci., 62, 947–973, 2005.

Remer, L.A., Tanré, D. and Kaufman, Y. J.: Algorithm for remote sensing of tropospheric aerosol from MODIS:Collection 5
 Product ID: MOD04/MYD04-C005., 1998.

D.C. Stein Zweers: TROPOMI ATBD of the UV aerosol index., 2016.

Tanr, D., J Kaufman, B. Y., Herman, M. and Mattoo, S.: Remote sensing of aerosol properties over oceans using the MODIS/EOS spectral radiances, J. Geophys. Res., 102, 16971–16988, doi:10.1029/96JD03437, 1997.

The Guardian: Chile battles devastating wildfires: 'We have never seen anything on this scale', [online] Available from:

545 https://www.theguardian.com/world/2017/jan/25/chile-fire-firefighting-internationalhelp?utm\_source=Climate+News+Network&utm\_campaign=afdf3cf10c-

EMAIL\_CAMPAIGN\_2017\_02\_03&utm\_medium=email&utm\_term=0\_1198ea8936-afdf3cf10c-38798061 (Accessed 25 Jan 2017), 2017.

Torres, O., Bhartia, P. K., Herman, J. R., Ahmad, Z. and Gleason, J.: Derivation of aerosol properties from satellite

550 measurements of backscattered ultraviolet radiation: Theoretical basis, J. Geophys. Res. Atmos., 103, 17099–17110, doi:10.1029/98JD00900, 1998.

Torres, O., Bhartia, P. K., Sinyuk, A., Welton, E. J. and Holben, B.: Total Ozone Mapping Spectrometer measurements of aerosol absorption from space: Comparison to SAFARI 2000 ground-based observations, J. Geophys. Res. D Atmos., 110,

D10S18, doi:10.1029/2004JD004611, 2005.

Torres, O., Tanskanen, A., Veihelmann, B., Ahn, C., Braak, R., Bhartia, P. K., Veefkind, P. and Levelt, P.: Aerosols and surface UV products from Ozone Monitoring Instrument observations : An overview, J. Geophys. Res., 112, D24S47, doi:10.1029/2007JD008809, 2007.
 Torres, O., Jethva, H. and Bhartia, P. K.: Retrieval of Aerosol Optical Depth above Clouds from OMI Observations:

Sensitivity Analysis and Case Studies, J. Atmos. Sci., 69, 1037–1053, doi:10.1175/JAS-D-11-0130.1, 2012.

560 Tukey, J.W.: Exploratory data analysis, Addison-Wesley Publishing Company, 1977. Winker, D. M., Vaughan, M. A., Omar, A., Hu, Y., Powell, K. A., Liu, Z.;Hunt, W.H.;Young, S. A.: Overview of the CALIPSO Mission and CALIOP Data Processing Algorithms, Technol. J. Atmos. Ocean., 26, 2310–2323, doi:10.1175/2009JTECHA1281.1, 2009.

Yin, B., Min, Q. and Joseph, E.: Retrievals and uncertainty analysis of aerosol single scattering albedo from MFRSR
measurements, J. Quant. Spectrosc. Radiat. Transf., 150, 95–106, doi:10.1016/j.jqsrt.2014.08.012, 2015.

570

575

580

---

## Author Response (AR1)

This document contains the responses to the 2 reviewers (response to the reviewer 1 starts in page 1 and response to the reviewer 2 starts in page 6), response to the editor (starts in page 8) followed by a marked-up version 3 manuscript (starts in page 12).

The reviewers' comments and questions are in **bold**. For each comment / question, the authors' reply / answer is 5 in black, and the corresponding modifications and line number is the manuscript version 3 are marked in blue colour.

**(1) Author's response to Anonymous Referee #1**

Thank you very much for your comments to our manuscript. The introduction has been trimmed. We also have revised 10 the entire manuscript. The following are the responses to your specific comments.

**Line 20-21: a difference in SSA of 0.06 is very big as far as aerosol remote sensing and climate applications are concerned! It is not "slightly" smaller.**

We admit that a difference 0.06 is not a small value for SSA and we have deleted this word accordingly. This statement 15 has been rephrased into The retrieved mean  $\omega 0$  at 550 nm for the entire plume over the period from 26-30 January 2017

varies from 0.81 to 0.87, whereas the nearest AERONET station reported values in the range from 0.89 to 0.92 (line 21 - 23).

But it is comparable with previous research. Hu et al. (2007) used TOMS AAI to retrieve SSA, their analytical uncertainty is 15%. For the typical AOD level in our case ( $0.3 \sim 0.7$ ), the SSA uncertainty is  $0.02 \sim 0.06$  according to

20 Hu et al. (2007) research, which matches with our results.

Furthermore, as we mentioned in the abstract and stressed throughout the manuscript, the AERONET site and the plume we defined are not collocated (the AERONET site is in the city centre as mention in line 200, which potentially overestimates the SSA). This makes this kind of retrieving to be validate. Other concerns, such as the lack of aerosol layer height, uncertainty in MODIS AOD and AERONET itself, should also be considered.

25

**Also, missing what is the purpose of this study? And what are the conclusions that the reader should take out of this work?**

This application attempts to quantify the aerosol absorption by retrieving SSA from satellite measured AAI. The conclusion is satellite retrieved AAI is a useful parameter to constrain forward simulation and to derive SSA. Although

30 currently we have a difference of 0.06 compared with AERONET, this discrepancy can be interpreted by the uncertainties in the inputs and AERONET itself as well as difference in measurement techniques (i.e. satellite vs ground-based measurements).

The purpose is presented in the last paragraph in Section 1 (line 59 - 61).

**35 Line 41: incorrect definition of SSA, it is not a ratio of radiation. This is too basic to be missed.**

Thank you for the correction on the SSA definition.

We have rephrased to  $\omega_0$  is defined as the ratio of the aerosol scattering over the extinction (line 33).

**Line 43-44: No, POLDER does not measure the "aerosol polarized phase function". It measured polarized 40 radiation that can be linked after modeling to the aerosol phase function.**

Thank you for the correction.

We have rephrased to More advanced sensors, such as the POLarization and Directionality of the Earth's Reflectances (POLDER), can retrieve  $\omega 0$  from a combination of multi-angular, multi-spectral observations of the polarized radiation (line 36-38).

45

Line 53-54: This is not entirely correct. Eck et al (2013) demonstrated that MODIS retrievals are impacted by the variation of SSA in smoke. (Eck, T. F., B. N. Holben, J. S. Reid, et al. 2013. "A seasonal trend of single scattering albedo in southern African biomass-burning particles: Implications for satellite products and estimates of emissions for the world's largest biomass-burning source." J. Geophys. Res. Atmos. 118 (12): 6414-6432 [10.1002/jarrd 50500])

**50 [10.1002/jgrd.50500])**

It may be a misunderstanding. We are not saying aerosol absorption has no effect on AOD, we are just saying AOD is less sensitive to aerosol absorption. AOD could be large under either a very scattering case or a very absorbing case. The reference as you mentioned here states the effect of retrieving AOD from a constant pre-assumed SSA, and this effect is presented as the systematic bias of retrieved MODIS AOD from AERONET AOD. This statement stresses more on the AOD

55 retrieval bias is sensitive to SSA, rather than AOD itself. Because the major revision in the Section 1 Introduction part, this sentence is no longer available.

**Line 60-61: not clear with what you mean that the AAI reduces the retrieval uncertainty. Uncertainty of what?**

Using AAI, instead of AOD to constrain the inversion of aerosol properties retrieval can reduce the uncertainty of 60 retrieved aerosol parameters. For AAI, the uncertainties come from the measured reflectance. But for AOD, the uncertainties come from both the measured reflectance and pre-assumed aerosol types.

Because the major revision in the Section 1 Introduction part, this sentence is no longer available. The corresponding content is in line 48 – 49: The most important advantage of the satellite retrieved AAI is that it does not dependent on assumptions on aerosol types, while a-prior aerosol types are major uncertainties in aerosol parameter retrievals, such as T.

65

**Line 66-67: this is a poorly phrased sentence and very confusing, the AAI in presence of aerosol is sensitive to aerosol height, SSA and concentration,... not SSA alone.**

Sorry for the confusion. But we do not mean that AAI is only sensitive to SSA, just in terms of SSA, AAI is more root sensitive than AOD.

Because the major revision in the Section 1 Introduction part, this sentence is no longer available. The corresponding content is in line 53: Moreover, the near-UV AAI is by definition highly sensitive to  $\omega_0$ .

**Line 98-99: incomplete/confusing sentence**

75 Sorry for the confusion.

We have rephrased to The basic idea of the residue method is that in a pure Rayleigh atmosphere, the reflectance (or equivalently the radiance  $(L_{\lambda})$ ) decreases strongly with the wavelength (line 79-80).

Equation 2: what is the definition of deltaI(lambda)?, This equation is different than what other groups use as definition of AAI. Can you provide a reference where this equation is derived? It seems to me a minus sign is missing, also not clear where the delta is coming from? For example de Graaf et al., (2007) uses the standard definition of AAI. how does you equation related with the more commonly used equations? (de Graaf, M., P.

**Stammes, and E. A. A. Aben (2007), Analysis of reflectance spectra of UV- absorbing aerosol scenes measured by SCIAMACHY, J. Geophys. Res., 112, D02206, doi:10.1029/2006JD007249.)**

85 We have added the definition of  $\Delta I_{\lambda 1}$  in line 88. We also have added that AAI calculation assumes a Rayleigh atmosphere at  $\lambda_2$ ,  $I_{\lambda 2}^{Ray}(a_s) = I_{\lambda 2}^{obs}$  (Herman et al., 1997) in line 87. The derivation from Eq.(1) to Eq.(2) is then not difficult (line 83). If it is still not clear to you, here is the derivation procedure:

According to the AAI definition:

$$AAI = -100 \left( log_{10} \left( \frac{l_{\lambda 1}}{l_{\lambda 2}} \right)^{obs} - log_{10} \left( \frac{l_{\lambda 1}}{l_{\lambda 2}} \right)^{Ray} \right)$$
(1)

90 , which can be re-written into:

$$AAI = -100(log_{10}(I_{\lambda 1})^{obs} - log_{10}(I_{\lambda 2})^{obs} - log_{10}(I_{\lambda 1})^{Ray} + log_{10}(I_{\lambda 2})^{Ray})$$
(2)

, the Rayleigh radiance is calculated by a surface albedo that satisfies  $(I_{\lambda 2})^{obs} = (I_{\lambda 2})^{Ray}$ , then Eq.(2) can be re-written into:

$$AAI = -100(log_{10}(I_{\lambda 1})^{obs} - log_{10}(I_{\lambda 1})^{Ray})$$
(3)

95 , reformed into:

$$AAI = 100(log_{10}(l_{\lambda 1})^{Ray} - log_{10}(l_{\lambda 1})^{obs})$$
(4)

$$AAI = 100 log_{10} \left( \frac{(\lambda_1)^2}{(l_{\lambda_1})^{obs}} \right)$$

$$\tag{5}$$

, here we define  $(I_{\lambda 1})^{Ray} = (I_{\lambda 1})^{obs} + \Delta I_{\lambda 1}$ , where  $\Delta I_{\lambda 1}$  can be explained as the change of radiance spectral dependency between a Rayleigh atmosphere and an observed atmosphere. Under cloud-free condition, the presence of absorbing 100 aerosols lead to a positive  $\Delta I_{\lambda 1}$ . The definition of  $\Delta I_{\lambda 1}$  is also mentioned at line 105 in the manuscript. Then Eq.(5) can be re-written into:

$$AAI = 100 log_{10} \left( \frac{(l_{\lambda1})^{obs} + \Delta l_{\lambda1}}{(l_{\lambda1})^{obs}} \right)$$

$$AAI = 100 log_{10} \left( \frac{\Delta l_{\lambda1}}{(l_{\lambda1})^{obs}} + 1 \right)$$
(6)
(7)

, which is the Eq.(2) in the manuscript.

105

**Line 117: linear interpolation of what? What parameters are being interpolated? Please explain.**

Linear interpolation of complex refractive index over spectral range from 340 to 675 nm.

We have rephrased into part into We obtain the size distribution function and complex refractive index at 440, 675, 880 and 1018 nm from AERONET, and apply the linear interpolation / extrapolation to derive the complex refractive index
over the spectrum from 340 to 675 nm, with spectral resolutions of 2 nm. Then DISAMAR uses above information to calculate the aerosol phase function P(Θ) and ω₀ over the full spectrum (line 103-106).

**Line 123: Aeronet Phase function data is not reported at 354nm , where does this come from? As it is , this is not correct.**

115 The phase function at 354 nm is calculated by the radiative transfer model DISAMAR with AERONET constraints. We took size distribution function, and complex refractive index at 440, 675, 870 and 1018 nm from AERONET. We used

linear extrapolation method to extend the spectrum refractive index to 340 nm. Then the radiative transfer model used that information to calculate the phase function and SSA over the full spectrum (those are intermediate outputs). That is how the phase function and SSA at 354 nm comes. With those intermediate outputs (that carry information on aerosol
 types), DISAMAR can execute forward simulation of AAI.

We have not have a this work into We obtain the size distribution f

We have rephrased this part into We obtain the size distribution function and complex refractive index at 440, 675, 880 and 1018 nm from AERONET, and apply the linear interpolation / extrapolation to derive the complex refractive index over the spectrum from 340 to 675 nm, with spectral resolutions of 2 nm. Then DISAMAR uses above information to calculate the aerosol phase function  $P(\Theta)$  and  $\omega_0$  over the full spectrum (line 103-106).

**125**

**Figures 3 and 4 cannot be interpreted because deltaI has not been explained/defined. No further reading of the manuscript since what I found until here warrants a rejection.**

Please refer to the derivation of Eq (2)  $(I_{\lambda 1})^{Ray} = (I_{\lambda 1})^{obs} + \Delta I_{\lambda 1}$  in previous or line 88 in the manuscript.

**130 Line 15: what max value was observed?**

To your question, the maximum AAI observed by OMI for Chile 2017 wildfires over all the pixels of the 4 days is 5.80. The maximum median value is 4.05 and obtained on 29 January 2017.

**Line 18: what measurements/obs you are referring to? Radiances?**

135 We are referring to the CALIOP backscatter coefficient measurements. We have rephrased into The simulated plume ascends to an altitude of 4.5-4.9 km, which is in good agreement with available CALIOP backscatter coefficient measurements (line 18-19).

**Line 18-20: not clear what you want to say in the sentence starting with "Due to the relatively..."**

140 To your question, the OMI observation is sparsely distributed which may contains geographical outliers that may not be the plume even it has AAI value larger than 1. Therefore, we applied an additional data quality control procedure with interquartile range technique. That is, calculate the difference between simulated and observed AAI, and remove the pixels that have AAI difference outside the interquartile range. This is detailed described in line 250-260.

We have rephrased into Due to the heterogeneity of the data that may contain the pixels outside the plume, an outlier detection criterion has to be applied (line 19-20).

**Line 20-21: are these SSA values averages over the plume or specific pixels?**

To your question, the retrieved SSA is the mean value for the entire plume over the period from 26-30 January 2017.

We have rephrased to The retrieved mean 600 at 550 nm for the entire plume over the period from 26-30 January 2017
varies from 0.81 to 0.87, whereas the nearest AERONET station reported values in the range from 0.89 to 0.92 (line 21-23).

**Line 33: replace "bright surfaces" with "snow", I am assuming this is what you meant.**

Thank you for correction.

155 Because the major revision in the Section 1 Introduction part, this sentence is no longer available.

Line 82-83: what are the locations of Pichilemu and Constitution? Are those forests ? Cities? regions? Please provide more details of the geographical setting. Was there a drought?

The location of Pichilemu and Constitución are two cities at the central of Chile as mentioned in line 62. The local
forestry industry (pine and eucalyptus) contributed a large fraction of the fire source in line 64. There was a drought as mentioned in line 61. All this information was actually mentioned in the version 2 manuscript.

**Line 84: figure 1 has very poor contrast when printed, please correct.** The original plot is from NASA. It has been adjusted now (line 515).

165

**(2) Author's response to the Anonymous Referee #2**

Thank you very much for your comments on our manuscripts. The following are the responses to your comments. 200

1. On page 6 authors write: "the fine and coarse mode are derived separately from AERONET". This is not true. AERONET inversion algorithm primarily retrieves aerosol size distribution (ASD) in 22 discrete radius points. The separation in two modes is done after the inversion by finding inflection point in between two ASD peaks and then approximation each part of ASD by log-normal distribution. The parameters of these

- 205 approximations are provided as ASD parameters for fine and coarse modes. The detailed description of this procedure can be find at https://aeronet.gsfc.nasa.gov/new\_web/Documents/Inversion\_products\_V2.pdf. Therefore the averaging of SSA of two modes is not needed. The total SSA for the initially retrieved ASD (at 22 points) can be used instead. I am wondering how close averaged SSA to the total SSA is provided by AERONET. In addition, SSA for fine and coarse modes are not advised to use because retrieval in
- 210 implemented under assumption that complex index of refraction is the same for all the sizes. Sorry for the confusion. We are not indicating that the AERONET derives fine and coarse mode separately, but indicating that we obtained the two modes by finding the peaks in the size distribution function provided by AERONET.
- The radiative transfer model DISAMAR currently cannot directly apply bi-modal distribution function. Instead, we 215 built two modes individually, corresponding to the two modes taken from AERONET. For each mode, we assigned the optical properties (refractive index) provided by AERONET. Then DISAMAR calculates the phase function and SSA for each mode. Finally, we combined two modes into one weighted by number density fraction and extinction cross section.

We have rephrased the sentence into The fine and coarse mode particle size are derived by finding the two peaks of the
 log-normal distribution function provided by AERONET. The complex refractive index is assumed the same for both modes. Since bi-modal aerosol is not applicable in DISAMAR yet, we first calculate optical properties of two modes individually, then we externally combine the optical properties of two modes into a bi-modal aerosol with a fraction (line 210-213).

225 2. On page 1 authors mention that retrieved SSA (0.84) is slightly lower that AERONET value. I encourage authors to replace or remove word slightly because the absolute difference 0.06 is significant in terms of radiative forcing estimation.

Thank you for your suggestion. We admit that 0.06 difference for SSA retrieval is not minor, and 'slightly' is not properly used.

230 We have deleted 'slightly' accordingly, and rephrased into The retrieved mean  $\omega_0$  at 550 nm for the entire plume over the period from 26-30 January 2017 varies from 0.81 to 0.87, whereas the nearest AERONET station reported values in the range from 0.89 to 0.92 (line 21-23).

**3. Page 2. "The foremost advantage of the AAI is its independence from assumptions on aerosol types, which 235 significantly reduce the retrieval uncertainty". This statement is confusing because the simulation of AAI is still**

**dependent on aerosol type.**

Sorry for the confusion. Indeed, AAI forward simulation (like what we do in this study) needs aerosol information, including aerosol loading, profile, micro-physics, etc. But AAI retrieval from satellite is independent of pre-assumed

aerosol types. It is calculated directly from the measured radiance (Eq.(1)). Here we want to stress that the AAI
 retrieval is independent of aerosol information itself, thus there is fewer uncertainties in the retrieved AAI.
 We have rephrased this sentence into The most important advantage of the satellite retrieved AAI is that it does not dependent on assumptions on aerosol types, while a-prior aerosol types are major uncertainties in aerosol parameter

- retrievals, such as τ (line 48-49).
  245 4. Page 3. What wavelength interpolation is used for? Index of retraction or aerosol optical parameters? To your question, we used linear interpolation to interpolate the AERONET refractive index from 440 to 1080 nm, and
- used the same technique to extrapolate the spectrum to 340 nm. With wavelength dependent size distribution function and refractive index, the radiative transfer model (DISAMAR) calculates the phase function and SSA for the full spectrum we specified (340 to 675 nm, with resolution of 2 nm) (those are intermediate outputs). With those intermediate 250 outputs (that carry information on aerosol types), DISAMAR can execute the forward simulation of AAI.
- We have rephrased this part into We obtain the size distribution function and complex refractive index at 440, 675, 880 and 1018 nm from AERONET, and apply the linear interpolation / extrapolation to derive the complex refractive index over the spectrum from 340 to 675 nm, with spectral resolutions of 2 nm (line 103-105).
- This is also mentioned in line 223-225 where we applied the same method for the case study: To constrain the spectral dependency of optical properties in the near-UV band, complex refractive index nr and ni in the UV band are linearly extrapolated using available data between 440 and 675 nm as mentioned in Section 2.2.
  In line 225-226 we mentioned the treatment for AERONET AOD and SSA (we use them for evaluating our results): Finally, the AERONET retrieved τ and ωo is also linearly interpolated to 550 nm.

260

265

**(3) Author's response to the editor**

275

Thank you very much for your comments on our manuscript. In the followings please find responses to your comments.

- Your manuscript # amt-2018-40 has received comments and suggestions from two anonymous reviews for which you have just submitted a response addressing their concerns. Thank you. Along the review process, I have read your paper and gave some thoughts on the research content. As I understood, the concept of the retrieving aerosol SSA in the near-UV region by constraining AOD quite resembles to the methods presented in Satheesh et al. (2008) and Gassó and Torres (2016). Both earlier studies, and the results of your own paper demonstrate that it is possible to retrieve aerosol SSA and layer height from the color ratio information (UVAI) in the near-UV given the AOD as an a priori and a pre-285 defined aerosol model. It was a surprise that your paper completely misses to discuss and cite these two papers relevant
- to the present study.

We have read the two research papers we missed out upon that you mention in the comment. We appreciate the studies you suggested here (the publication of Satheesh et al. was actually published in 2009). From our view, the goals and / or the methods of these two studies do not resemble to ours. Satheesh et al. (2009) provided a hybrid retrieval method by combing
OMI and MODIS measurements, and Gassó and Torres (2016) aimed to discuss the discrepancy between OMAERUV AOD and other independent measurements. But both studies did discuss the relation among aerosol layer height, aerosol concentration, and aerosol absorption, thus we have included them in the introduction part.

Satheesh et al. (2009) used a hybrid approach to retrieve aerosol layer height (ALH) and aerosol single scattering albedo (SSA).

295 They combined the OMI aerosol product (OMAERUV), which is sensitive to ALH and aerosol absorption, with MODIS's accurate aerosol optical depth (AOD), which is insensitive to ALH. Their study has a similar object as ours, that is to retrieve SSA from satellite measurements, but the method is not the same. Both studies retrieve ALH and SSA from given a priori aerosol models. Satheesh et al. used MODIS AOD as the parameter to constrain the operational OMAERUV retrieval, while we use the absorbing aerosol index (AAI). The role of AAI used in Satheesh et al. (2009) is a qualitive parameter to distinguish absorbing aerosols from non-absorbing ones.

Specifically, Satheesh et al. (2008) extrapolated MODIS AOD to the near UV band. Using this MODIS-produced AOD to constrain the standard OMI AOD inversion procedure (OMAERUV) allows to derive improved ALH and SSA (that specified in the LUT). In our study, we used MODIS standard AOD at 550 nm to constrain the radiative transfer calculation in the forward simulation, and used OMI measured AAI to constrain the backward retrieval of ALH and SSA. Furthermore, Satheesh

- 305 et al. (2009) only compared the SSA retrieved with the hybrid-algorithm with that retrieved with the standard OMI algorithm SSA. They did not validate it with measurements from other instruments such as AERONET. Satheesh et al. mentioned the difficulty of extrapolating MODIS AOD from visible band to 388 nm where OMI requires the AOD information for retrieval (especially difficult for small particles, e.g. biomass burning aerosols). They first applied a linear least square fitting (log-log scale) for AOD as a function of wavelength, then they improved the method by including
- 310 information on the AOD spectral curvature. The relation between UV and visible AOD may provide some clues to determine the relation between UV and visible SSA and refractive index, and it is worth to study in a separated research. Satheesh et al. also mentioned that measurements of aerosol absorption in the UV spectral range are rare, which makes it difficult to validate the retrieval results. We met similar difficulties in determining the spectral dependence of aerosol properties from the visible to the UV band. But most importantly, we did not notice that the use of MODIS AOD can eliminate the uncertainty of ALH
- 315 assumption. We have included this as one of the reason that use MODIS AOD to constrain the forward simulation.

Gassó and Torres (2016) compared the OMAERUV aerosol product with independent measurements (MODIS and AERONET) in order to: (1) assess the quality of the OMAERUV AOD retrieval over ocean (section 3); (2) estimate the impact of cloud contamination on their AOD retrieval (section 4); (3) demonstrate the effect of variations in aerosol concentration and

- 320 height on OMAERUV by comparing the operational OMAERUV algorithm and the OMI-MODIS joint algorithm presented in Satheesh et al. (2009) (section 5); (4) determine whether the assumed aerosol models that leads to discrepancies in AOD retrieval of section 5 (section 6). Gassó and Torres (2016) aimed to evaluate the influences of several factors on the AOD retrieval, rather than to retrieve the SSA from measured AAI (with MODIS AOD constrain).
- Through case studies, Gassó and Torres (2016) evaluated the hybrid retrieval method presented by Satheesh et al. (2009) via 325 case studies. At high AAI magnitude (AAI > 1.8), the retrieved ALH is comparable with CALIOP measurements, although their method is quite sensitive to small variations in the input AOD in case of low AAI. Although they seemed to discuss the interplay among AAI, ALH and AOD, the AAI was not used as the constraint to retrieved aerosol absorption. However, Gassó and Torres (2016) concluded that AAI alone cannot be used quantitatively if no ALH or AOD information is available. This is consistent with our finding that the accuracy of the retrieved SSA depends strongly on the uncertainties in AOD and ALH.

330

In summary, although these two studies discussed the interplay among AOD, ALH, and aerosol absorption (SSA / AAI / refractive index), but the purpose / method differ from ours. We use the measured AAI as an additional constraint to retrieve SSA, which is not the case the other two studies. But both studies provided useful information in aerosol research that we can include in our manuscript. Here are the corresponding contents that we have included study of Satheesh et al. (2009) and Gassó and Tarma (2016).

335 and Torres (2016):

In section 1 introduction: But this aerosol absorption over near-UV is highly sensitive to the assumption on aerosol layer height. Satheesh et al. (2009) therefore used the  $\tau$  from MODerate-resolution Imaging Spectroradiometer (MODIS), which is independent of aerosol layer height, to constrain the OMAERUV retrieval. The validation showed that compared with operational OMAERUV algorithm, the retrieved aerosol height by the hybrid method is in a better agreement with air-borne

340 measurements, implying a potential improvement in aerosol absorption retrieval. This OMI-MODIS joint retrieval was also evaluated by Gassó and Torres (2016). They found under less absorbing condition, the hybrid method is sensitive to the variations in the input  $\tau$ , which is used to select the retrieved pair of aerosol layer and  $\omega_0$ . (line 43 - 49)

In section 3.1.2 MODIS and OMI aerosol optical thickness: Besides, the MODIS retrieved  $\tau$  is free from the uncertainty triggered by assumed aerosol profile (Satheesh et al., 2009). (line 187 - 188)

345 In section 3.2 Methodology: Previous research suggested AAI cannot be quantitatively used without  $\tau$  or zaer information (Gassó and Torres, 2016). (line 257 - 258)

In this inversion process, your study employs an aerosol model developed using the AERONET data. It was not clear in the Methodology section what kind of spectral dependence is assumed between the two near-UV wavelengths. A 350 number of studies, using laboratory measurements and satellite data, have shown that the carbonaceous aerosols generated from the open field agricultural burning and wildfires exhibit a strong wavelength-dependent of absorption (imaginary part of the refractive index) in the near-UV region. In our earlier study (Jethva and Torres, 2011, ACT), we learned that the relative spectral dependence in the imaginary part of the refractive index needs to be increased to 20% with reference to no spectral dependence (equivalent to black carbon) in order to obtain a better agreement in

355 AOD and SSA between OMI and AERONET. A strong wavelength dependence in aerosol absorption in the UV has been shown as a proxy for the amount of organics in the biomass burning smoke. The color ratio (UVAI) information in the near-UV strongly varies with the Absorption Angstrom Exponent, a parameter that quantifies the spectral dependence of aerosol absorption. Since your paper lacks to provide this essential information, it is hard to interpret the SSA retrievals presented in the paper. Perhaps, a significant disagreement of 360 ~0.06 in SSA between the OMI retrievals and that of AERONET reflects the issue described above that you should consider while revising your manuscript. Please verify the relative spectral dependence in absorption assumed in the aerosol model and discuss/mention in the revision.

Apology for the unclear explanation in the Section 3 Methodology and datasets. Indeed, the spectral dependence of aerosol absorption has a significant influence on the retrieval of SSA. As you mentioned in Jethva and Torres (2011), compared with 365 the gray aerosol assumption used in OMAERUV algorithm, an aerosol model with wavelength dependence can significantly reduce the bias in retrieved AOD and achieve better agreement of retrieved SSA with AERONET measured SSA (70% retrieved data is within +/-0.03 difference).

Therefore, one has to be careful when relating the aerosol refractive index in the UV to the one in the visible band. Jethva and Torres (2011) finally adopted a spectral dependence with a 20% increase in the 354 nm imaginary refractive index with respect

- 370 to the value at 388 nm (absorbing Angstrom Exponent around 2.5 to 3.0 for seven carbonaceous aerosol models in OMAERUV). It is an improvement compared to the operational application, where the aerosol properties are pre-assumed. However, aerosol properties evolve over time and thus also over space. Also, even for a specific aerosol type, the aerosol properties vary from one study to another. Thus, we prefer to directly use AERONET measurements to define the aerosol type rather than to use to one of the pre-defined aerosol types.
- 375 The corresponding content in the manuscript has been rephrased into: The spectral bands of the AERONET instrument at this site only cover the visible band. Due to the absence of observations, assumptions have to be made on the spectral dependence of aerosol properties to obtain their values in the near-UV band. The properties of biomass burning aerosol depend on the type of fuel, the procedure producing the smoke, the age of the smoke, and also the atmospheric conditions (Reid et al., 2005). Using measurements to constrain the input aerosol refractive index can reduce the uncertainties due to a-priori knowledge.
- 380 Our treatment on the complex refractive index is as following: (1) take the complex refractive index at visible band (440 to 675 nm) from AERONET measurements; (2) linearly extrapolate the complex refractive index to near-UV band. The real part nr for radiative transfer calculation is obtained in this step. A slight wavelength dependence of nr is found from the measurements; (3) for the imaginary part nr, we multiply it (for the entire wavelength from UV to visible) with a scaling factor as we treat it as a free parameter. By varying the value of the scaling factor, both the magnitude and the wavelength dependence of nr ic can change to meet the requirement of retrieval (line 234 243).
- We have now also included the retrieved refractive index in Table 2. In our case the spectral dependence between 354 and 388 nm is less than 5%, which is smaller than the 20% in Jethva and Torres (2011). We also added a plot to show the original AERONET measured (dashed line) and retrieved complex refractive index (scaled by a factor) in Fig.9. Although the scaling factors can amplify the spectral dependence, it is not very large due to the flat character of the original refractive index. We
- 390 have added an explanation for this: There are many sources accounting for this discrepancy. First of all, the nearest site Santiago\_Beauchef is not exactly in the primary biomass burning regions as mentioned in section 3.1.3. The AERONET site is located in downtown, where reflective urban or industrial aerosols may have been mixed with the smoke enhancing the ω0. This would also affect the complex refractive index used in radiative transfer calculation, since we use the AERONET measured refractive index to constrain the forward simulations. According to Table 2, the retrieved n reveals that the difference
- **395** between 354 and 388 nm is less than 5%. This small spectral dependence of  $n_i$  is mainly determined by AERONET measurements in the visible band (dashed lines), whereas the effect of the scaling factor is minor in this case (Fig.9). We thus find a much weaker wavelength dependence than in the Jethva and Torres (2011) study, where a 20% difference between the two UV wavelengths was applied to OMAERUV algorithm to achieve the result that 70% of the retrieved  $\omega_0$  differ less than  $\pm 0.03$  from the  $\omega_0$  from the AERONET measurements. A stronger spectral dependence of  $n_i$  between 354 and 388 nm would
- 400 allow simulations to reach a higher AAI while keeping  $n_i$  at a relatively low level, so would retrieve a higher  $\omega_0$  at 550 nm. The presence of non-absorbing aerosols weakens the spectral dependence (particularly in the UV spectral range) and the linear

extension would overestimate the aerosol absorption. In this situation, the uncertainties due to assumed spectral aerosol properties might compensate the measurement errors to some extent (line 313 - 326).

In our case, we use the AERONET measurement to constrain the spectral dependence of aerosol properties, but it might be 405 improved if observations from other observational aerosol properties in UV band would be available as constraint. This is summarized in the conclusion: This study proves the potential of utilizing OMI measured AAI to quantitatively characterize aerosol optical properties like  $\omega_0$ . Currently, it is challenging to retrieve and validate results without reliable aerosol profile information. In the future, the availability of daily global aerosol profile data, e.g. the L2 aerosol layer height product

[revised manuscript text omitted]

|                  | (Formatted: Font: 12 pt, English (US)                                    |
|------------------|--------------------------------------------------------------------------|
| 7                | Formatted: Font: 10 pt, English (US)                                     |
| 2                | Formatted: Left                                                          |
|                  | Formatted: Left                                                          |
| Z                | Formatted: English (US), Not Highlight                                   |
| /                | Deleted: The absorbing aerosol index (AAI) based on the near             |
|                  | Ultra-Violet (near-UV) remote sensing techniques is a qualitative. [1]   |
| 1                | Deleted: isis presented exclusively on biomass burning aeroso[2]         |
| V                | Formatted: Font: 10 pt, English (US)                                     |
| 1                | Formatted: Not Highlight                                                 |
|                  | Deleted: relatively small data size                                      |
|                  | Formatted [3]                                                            |
|                  | Commented [S]1]: Because the data in this case is sensitive to[5] |
| 1                |                                                                          |
| h                | Formatted [4]                                                            |
|                  | Formatted [6]                                                            |
|                  | Commented [JV2]: Add a range instead of a single number?          |
|                  | Commented [Sj3R2]: Changed.                                              |
|                  | Deleted: are over                                                        |
| 1                | Formatted: Font: 10 pt, English (US)                                     |
| //               | Formatted: Font: 5.5 pt                                                  |
| 1                | Formatted [7]                                                            |
| 7                | Deleted: is                                                              |
| 2                | Formatted [8]                                                            |
|                  | Deleted: approximately                                                   |
|                  | Deleted: in the range                                                    |
|                  | Formatted [9]                                                            |
| /                | Deleted: 0.84,                                                           |
|                  | Pointed: English (US)                                                    |
|                  |                                                                          |
| $\left( \right)$ | Formatted [10]                                                           |
| V,               | Deleted: signify smaller than the value of 0.90 measured [11]            |
|                  | Formatted                                                                |
|                  | Deleted: Except for the observational errors, the impact of       |
|                  | Deleted: Except for the observational errors, the impact of              |
|                  | Formatted                                                                |
| <                | Deleted: also f great concern from the perspective of climate 177 |
| 1                | Formatted                                                                |
| /                | Deleted: ) one type of absorbing aerosol black carbon (BC) carbon |
| 1                | Formatted: Font: 10 pt. English (US)                                     |
| $\left( \right)$ | Deleted: (Huang et al., 2013 or by enhancing the absorption of on |
|                  | Formatted                                                                |
|                  | Deleted:                                                                 |
|                  | Formatted [22]                                                           |
| $l_{l}$          | Deleted:ut But                                                           |
| 1                | Deleted:defined as the ratio of the aerosol scattering over thest        |
|                  | Formatted [26]                                                           |
| -                | Deleted: Many implementations have been done for                         |
| 1                | Formatted [27]                                                           |
|                  |                                                                          |

[revised manuscript text omitted]

**Formatted**

**.....**

**Deleted:**  $\omega_0$  is defined as the ratio of the radiation scattered by aerosol particles to the total attenuation. Because aerosol compositions and properties are highly variable in space and time, measuring the global distribution of  $\omega_0$  relies on remote sensing techniques. The POLarization and Directionality of the Earth's Reflectances (POLDER)  $\omega_0$  from a combination of multi-angular, multi-spectral observations of the measures aerosol polarized phase function. This provides information directly related to  $\omega_0$  (Lercy et al., 1997). But However, there is no continuous temporal coverage because the first two instruments encountered technical hitches that problems on the satellite level. The third POLDER instrument mission covered the period 2004-2014.

... [28]

**Deleted:** As a result,  $\omega_0$  is usually retrieved by forward simulations that are adapted to observational parameters. Many implementations have been done for ground-based network measurements (Dubovik et al., 1998; Eck et al., 2003; Petters et al., 2003; Kassianov et al., 2005; Corr et al., 2009; Yin et al., 2015), while relatively fewer applications to satclifte instruments exist due to lack of validation (Lee et al., 2007; Ialongo et al., 2010; Eck et al., 2013). Moreover, a majority of those methods heavily depend on the aerosol optical thickness (t), either in forward model simulations or in validation procedures. This makes the derived  $\omega_0$  subject to large uncertainties. The reason is that retrieval requires assumptions on aerosol types, and the commonly used t that is retrieved in the visible band where the signal of bright surfaces is strong. Besides, the aerosol effect on radiance is inversely proportional to wavelength (Kaufman, 1993), and the sensitivity. Top Deleted: instruments

| Deleted: improved                                                                           |
|---------------------------------------------------------------------------------------------|
| Formatted [30]                                                                              |
| Deleted:                                                                                    |
| Formatted [31]                                                                              |
| Formatted [32]                                                                              |
| Formatted [33]                                                                              |
| Deleted: that constrains                                                                    |
| Formatted [34]                                                                              |
| Deleted: forward model simulations with the absorbing aeroso[35]                            |
| Formatted [36]                                                                              |
| Deleted: more than 35 years                                                                 |
| Formatted [37]                                                                              |
| Deleted: improving                                                                          |
| Deleted: foremost ost important advantage of the satellite [38]                             |
| Deleted: ce                                                                                 |
| Deleted: from n assumptions on aerosol types, which [39]                                    |
| Formatted [40]                                                                              |
| Deleted:Ginoux et al. (2004) suggested that comparing model1]                               |
| Formatted [42]                                                                              |
| Deleted: , the sensitivity of $\tau$ in the visible band to $\omega_0$ is lower [43] |
| Formatted [44]                                                                              |
| Formatted [45]                                                                              |
| Deleted: Empirical models were also developed to build [46]                                 |
| Formatted [47]                                                                              |
| Deleted: follows revious research to quantify the aerosol [48]                              |
| Formatted [49]                                                                              |
| Deleted:and caused massive losses of the local forestry industry                            |
| Deleted: the MODerate-resolution Imaging Spectroradiometer [51]                             |
| Formatted: Font: 10 pt, English (US)                                                        |

**2 AAI sensitivity studies based on DISAMAR**

In this section, we first introduce the near-UV AAI. In the sensitivity analysis, we show that the AAI depends not only on aerosol parameters, but also on the surface conditions and the observation geometry. The sensitivity analysis in this study is only designed for biomass burning aerosols.

**2.1 Near-UV AAI definition**

800

The concept of the near-UV AAI was first conceived to detect UV-absorbing aerosols from the spectral contrast provided by TOMS observations, known as the residue method (Herman et al., 1997). The basic idea of the residue method is that in a pure Rayleigh atmosphere, the reflectance (or equivalently the radiance (L2) decreases strongly with the wavelength. The presence of absorbing aerosols will reduce this spectral dependency of L2. The change in this wavelength dependency is summarized as the AAI, which is calculated from the L2 at the wavelength pair  $\lambda_1$  and  $\lambda_2(\lambda_1 < \lambda_2)$ :

$$AAI = -100 \left( log_{10} \left( \frac{l_{\lambda 1}}{d_{\lambda 2}} \right)^{obs} - log_{10} \left( \frac{l_{\lambda 1}}{d_{\lambda 2}} \right)^{Ray} \right),$$

(1)

The *obs* and *Ray* denote the Jk from the satellite measurement and calculated using a Rayleigh atmosphere, respectively. The longer wavelength  $\lambda_2$  is treated as reference wavelength where the surface albedo (as) is determined by fitting the

805 observed radiance, i.e.  $\int_{a_1}^{Ray} (g_{s_0}) = \int_{a_1}^{bbs} \sum_{a_2}^{a_1} F_{a_1}^{bbs}$  is done using an atmosphere containing only molecular scattering bounded by a Lambertian surface. The spectral dependence of the surface albedo is neglected thus  $\int_{a_1}^{Ray} \sum_{a_1}^{a_2} \sum_{a_2}^{a_3} \sum_{a_1}^{a_2} \sum_{a_2}^{a_3} \sum_{a_1}^{a_2} \sum_{a_2}^{a_3} \sum_{a_1}^{a_2} \sum_{a_2}^{a_3} \sum_{a_3}^{a_4} \sum_{a_4}^{a_4} \sum_{a_5}^{a_5} \sum_{a_1}^{a_5} \sum_{a_5}^{a_5} \sum_{a_5$

AAI =  $100 \log_{10} \left( \frac{\Delta I_{\lambda 1}}{I_{0}^{obs}} + 1 \right)$

(2)

Jt is advantageous to use Eq.(2) because the AAI can be simply interpreted as the ratio between the simulated and observed 810 radiance at  $\lambda_{1,x}$

**2.2 Near-UV AAI sensitivity studies**

In this section, we present results from sensitivity studies, performed with the radiative transfer model DISAMAR. DISAMAR can perform simulations of the forward L spectrum in a wide spectral coverage (270 nm to 2.4 μm) and models scattering and absorption by gases, aerosols and clouds, as well as reflection by the surface (De Haan, 2011). It uses either

- 815 the Doubling-Adding method or the Layer Based Orders of Scattering (LABOS) for the radiative transfer calculations. In this study the latter one is used, because it is less computationally intensive (De Haan et al., 1987; De Haan, 2011), DISAMAR allows to apply several aerosol scattering approximations. Here we assume Mie scattering aerosols. The parameters to describe Mie particles and their corresponding values are listed in Table 1. Considering the Chile wildfires plumes, which were dominated by biomass burning aerosols, these sensitivity studies are specifically performed for
- 820 parameterized smoke aerosols, with only fine mode particles and weak linearly wavelength dependency of the complex refractive index (nr and ni). The default values refer to observations of the daily average on January 27 of the AERONET station Santiago Beauchef (33.46°S, 70.66°W). We obtain the size distribution function and complex refractive index at 440, 675, 880 and 1018 nm from AERONET, and apply the linear interpolation / extrapolation to derive the complex refractive index over the spectrum from 340 to 675 nm, with spectral resolutions of 2 nm. Then DISAMAR uses above information to
- s25 calculate the aerosol phase function  $P(\Theta)$  and  $\omega_0$  over the full spectrum. The corresponding  $P(\Theta)$  at 354 nm is presented in Fig.2. DISAMAR requires  $\tau_1$  to be defined at reference wavelength 550 nm. Surface parameters include a spectrally flat as and the surface pressure Ps. The aerosol profile is parameterized as a single layer box shape, with its bottom at  $z_{aer}$ - $\Delta z/2$  and top at  $z_{aer}$ + $\Delta z/2$ , where  $z_{aer}$  and  $\Delta z$  are the geometric central height and the geometric thickness of the aerosol layer, respectively. The whole sensitivity analysis is performed for cloud-free conditions. The wavelength pair of OMI (354 and

| (                     | Formatted: Heading 2                                                                                                                  |               |
|-----------------------|---------------------------------------------------------------------------------------------------------------------------------------|---------------|
|                       | Formatted: Left                                                                                                                       |               |
|                       | Deleted: forn a pure Rayleigh atmosphere, wherehe                                                                              |               |
|                       | reflectance, (or equivalently the radiance (lb),                                                                                      | [53]          |
|                       | Formatted                                                                                                                             | [52]          |
| $\parallel \parallel$ |                                                                                                                                       | [54]          |
| / //                  | Formatted                                                                                                                             | [55]          |
|                       | Deleted: observedeasurement and the modelalculate Rayleigh atmosphere $I_{\lambda}$                                            | d using a     |
| ///                   | Formatted                                                                                                                             | [57]          |
| $\parallel \parallel$ | Formatted                                                                                                                             | [58]          |
| $\ \ $                | Deleted: The aerosol effect is assumed at $\lambda_2$ is negligible,                                                           | so that       |
|                       | the assumption of a Rayleigh atmosphere at this wavelength.
This The way the longer wavelength $\lambda_2$ is treated as reference | is valid.     |
|                       | wavelength where the surface albedo ( $a_s$ ) is determined by fin                                                                    | ting [159]    |
|                       | Deleted: the difference between $I_{\lambda 1}^{obs}$ and $I_{\lambda 1}^{Ray}$ normalized                                     | by.the641     |
|                       | Formatted                                                                                                                             | [60]          |
| 14                    | Formatted                                                                                                                             | [61]          |
|                       | Formatted                                                                                                                             | [62]          |
| $\parallel \mid$      | Deleted: Consequently,E(1) can be equivalently tran                                                                                   | sforined      |
| 1//                   | Deleted: rephrased                                                                                                                    | [00]          |
|                       | Formatted                                                                                                                             | [65]          |
| 1                     | Formatted                                                                                                                             | [66]          |
| -                     | Formatted                                                                                                                             | [67]          |
| 7                     | Formatted                                                                                                                             | [68]          |
| 1                     | Deleted: The advantage                                                                                                                | [00]          |
| 4                     | Formatted: Font: 10 pt, English (US), Not Highlight                                                                                   | $\rightarrow$ |
| (                     | Deleted: of                                                                                                                           |               |
| Y                     | Formatted                                                                                                                             | [69]          |
| $\mathcal{N}$         | Deleted: writing the AAI using equation 2                                                                                             |               |
| N                     | Formatted                                                                                                                             | [70]          |
|                       | Deleted: is that                                                                                                                      | [/ 0]         |
| $\langle \rangle$     | Formatted                                                                                                                             | [71]          |
| $\langle \rangle$     | Deleted: 1                                                                                                                            |               |
|                       | Formatted                                                                                                                             | [72]          |
| Å                     | Deleted: The sensitivity studies areperformed with the ra                                                                             | adiatives     |
| $\langle \rangle$     | Formatted                                                                                                                             | [74]          |
| Y                     | Deleted: DISAMARt uses either the Doubling-Adding                                                                                     | methods       |
| $\langle \rangle$     | Formatted                                                                                                                             | [76]          |
| Y                     | Formatted                                                                                                                             | [77]          |
| $\gamma$              | Deleted: Given size distribution function (rg), complex refi                                                                          | active 81     |
| $\geq$                | Deleted: thathich mainly producedere dominated by                                                                                     | biomayoj      |
| (                     | Commented [JV4]: Which period?                                                                                                        |               |
|                       | Commented [Sj5R4]: Added.                                                                                                             |               |
| Ŋ                     | Formatted: Font: 5.5 pt                                                                                                               |               |
| Y                     | Formatted                                                                                                                             | [80]          |
|                       | Deleted: and t                                                                                                                        |               |
|                       | Formatted                                                                                                                             | [81]          |
| (                     | Deleted: should                                                                                                                       |               |
| Y                     | Formatted                                                                                                                             | [821          |
| 14                    | Deleted: influences                                                                                                                   |               |
|                       | Deleted: s                                                                                                                            | $\rightarrow$ |

[revised manuscript text omitted]

| Deleted: make it                                                                                                                                                                                                                                                                                                                                                                                                                                                                                                                                                                                                                                                                                                                                                                                                                                                                                                                                                                                                                                                                                                                                                                                                                                              |                                                                                                                                                                                                                               |
|---------------------------------------------------------------------------------------------------------------------------------------------------------------------------------------------------------------------------------------------------------------------------------------------------------------------------------------------------------------------------------------------------------------------------------------------------------------------------------------------------------------------------------------------------------------------------------------------------------------------------------------------------------------------------------------------------------------------------------------------------------------------------------------------------------------------------------------------------------------------------------------------------------------------------------------------------------------------------------------------------------------------------------------------------------------------------------------------------------------------------------------------------------------------------------------------------------------------------------------------------------------|-------------------------------------------------------------------------------------------------------------------------------------------------------------------------------------------------------------------------------|
| Formatted                                                                                                                                                                                                                                                                                                                                                                                                                                                                                                                                                                                                                                                                                                                                                                                                                                                                                                                                                                                                                                                                                                                                                                                                                                                     | [83]                                                                                                                                                                                                                          |
| Deleted:                                                                                                                                                                                                                                                                                                                                                                                                                                                                                                                                                                                                                                                                                                                                                                                                                                                                                                                                                                                                                                                                                                                                                                                                                                                      |                                                                                                                                                                                                                               |
| Formatted                                                                                                                                                                                                                                                                                                                                                                                                                                                                                                                                                                                                                                                                                                                                                                                                                                                                                                                                                                                                                                                                                                                                                                                                                                                     | [84]                                                                                                                                                                                                                          |
| Deleted: ,                                                                                                                                                                                                                                                                                                                                                                                                                                                                                                                                                                                                                                                                                                                                                                                                                                                                                                                                                                                                                                                                                                                                                                                                                                                    |                                                                                                                                                                                                                               |
| Formatted                                                                                                                                                                                                                                                                                                                                                                                                                                                                                                                                                                                                                                                                                                                                                                                                                                                                                                                                                                                                                                                                                                                                                                                                                                                     | [85]                                                                                                                                                                                                                          |
| Deleted: ¶                                                                                                                                                                                                                                                                                                                                                                                                                                                                                                                                                                                                                                                                                                                                                                                                                                                                                                                                                                                                                                                                                                                                                                                                                                                    |                                                                                                                                                                                                                               |
| Formatted                                                                                                                                                                                                                                                                                                                                                                                                                                                                                                                                                                                                                                                                                                                                                                                                                                                                                                                                                                                                                                                                                                                                                                                                                                                     | [86]                                                                                                                                                                                                                          |
| Formatted                                                                                                                                                                                                                                                                                                                                                                                                                                                                                                                                                                                                                                                                                                                                                                                                                                                                                                                                                                                                                                                                                                                                                                                                                                                     | [87]                                                                                                                                                                                                                          |
| Formatted                                                                                                                                                                                                                                                                                                                                                                                                                                                                                                                                                                                                                                                                                                                                                                                                                                                                                                                                                                                                                                                                                                                                                                                                                                                     | [88]                                                                                                                                                                                                                          |
| Deleted: T                                                                                                                                                                                                                                                                                                                                                                                                                                                                                                                                                                                                                                                                                                                                                                                                                                                                                                                                                                                                                                                                                                                                                                                                                                                    | [00]                                                                                                                                                                                                                          |
| Deleted: effect                                                                                                                                                                                                                                                                                                                                                                                                                                                                                                                                                                                                                                                                                                                                                                                                                                                                                                                                                                                                                                                                                                                                                                                                                                               |                                                                                                                                                                                                                               |
| Formatted                                                                                                                                                                                                                                                                                                                                                                                                                                                                                                                                                                                                                                                                                                                                                                                                                                                                                                                                                                                                                                                                                                                                                                                                                                                     | [98]                                                                                                                                                                                                                          |
| Deleted:                                                                                                                                                                                                                                                                                                                                                                                                                                                                                                                                                                                                                                                                                                                                                                                                                                                                                                                                                                                                                                                                                                                                                                                                                                                      |                                                                                                                                                                                                                               |
| Formatted                                                                                                                                                                                                                                                                                                                                                                                                                                                                                                                                                                                                                                                                                                                                                                                                                                                                                                                                                                                                                                                                                                                                                                                                                                                     | 1001                                                                                                                                                                                                                          |
| Deleted: The asymmetry factor g is the averaged cosine                                                                                                                                                                                                                                                                                                                                                                                                                                                                                                                                                                                                                                                                                                                                                                                                                                                                                                                                                                                                                                                                                                                                                                                                        | of the [93]                                                                                                                                                                                                                   |
| Formatted                                                                                                                                                                                                                                                                                                                                                                                                                                                                                                                                                                                                                                                                                                                                                                                                                                                                                                                                                                                                                                                                                                                                                                                                                                                     | [01]                                                                                                                                                                                                                          |
| Formatted                                                                                                                                                                                                                                                                                                                                                                                                                                                                                                                                                                                                                                                                                                                                                                                                                                                                                                                                                                                                                                                                                                                                                                                                                                                     | [ 1 ]                                                                                                                                                                                                                  |
| Formatted                                                                                                                                                                                                                                                                                                                                                                                                                                                                                                                                                                                                                                                                                                                                                                                                                                                                                                                                                                                                                                                                                                                                                                                                                                                     | [J2]                                                                                                                                                                                                                          |
| Deleted:                                                                                                                                                                                                                                                                                                                                                                                                                                                                                                                                                                                                                                                                                                                                                                                                                                                                                                                                                                                                                                                                                                                                                                                                                                                      |                                                                                                                                                                                                                               |
| Formatted                                                                                                                                                                                                                                                                                                                                                                                                                                                                                                                                                                                                                                                                                                                                                                                                                                                                                                                                                                                                                                                                                                                                                                                                                                                     | TOF1                                                                                                                                                                                                                          |
| Deleted: geometry                                                                                                                                                                                                                                                                                                                                                                                                                                                                                                                                                                                                                                                                                                                                                                                                                                                                                                                                                                                                                                                                                                                                                                                                                                             | [95]                                                                                                                                                                                                                          |
| Formatted                                                                                                                                                                                                                                                                                                                                                                                                                                                                                                                                                                                                                                                                                                                                                                                                                                                                                                                                                                                                                                                                                                                                                                                                                                                     |                                                                                                                                                                                                                               |
| Formatted                                                                                                                                                                                                                                                                                                                                                                                                                                                                                                                                                                                                                                                                                                                                                                                                                                                                                                                                                                                                                                                                                                                                                                                                                                                     | [96]                                                                                                                                                                                                                          |
| Deleted: ( is 3 (c) and (d))                                                                                                                                                                                                                                                                                                                                                                                                                                                                                                                                                                                                                                                                                                                                                                                                                                                                                                                                                                                                                                                                                                                                                                                                                                  | [97]                                                                                                                                                                                                                          |
|                                                                                                                                                                                                                                                                                                                                                                                                                                                                                                                                                                                                                                                                                                                                                                                                                                                                                                                                                                                                                                                                                                                                                                                                                                                               | [98]                                                                                                                                                                                                                          |
| Formatted                                                                                                                                                                                                                                                                                                                                                                                                                                                                                                                                                                                                                                                                                                                                                                                                                                                                                                                                                                                                                                                                                                                                                                                                                                                     | [99]                                                                                                                                                                                                                          |
| Deleted:                                                                                                                                                                                                                                                                                                                                                                                                                                                                                                                                                                                                                                                                                                                                                                                                                                                                                                                                                                                                                                                                                                                                                                                                                                                      |                                                                                                                                                                                                                               |
| Deleted: As shown in Fig.3 (e) and (f), even with a dec                                                                                                                                                                                                                                                                                                                                                                                                                                                                                                                                                                                                                                                                                                                                                                                                                                                                                                                                                                                                                                                                                                                                                                                                | reasing 100]                                                                                                                                                                                                                  |
| Formatted                                                                                                                                                                                                                                                                                                                                                                                                                                                                                                                                                                                                                                                                                                                                                                                                                                                                                                                                                                                                                                                                                                                                                                                                                                                     | [101]                                                                                                                                                                                                                         |
| Formatted                                                                                                                                                                                                                                                                                                                                                                                                                                                                                                                                                                                                                                                                                                                                                                                                                                                                                                                                                                                                                                                                                                                                                                                                                                                     | [102]                                                                                                                                                                                                                         |
| Deleted: T                                                                                                                                                                                                                                                                                                                                                                                                                                                                                                                                                                                                                                                                                                                                                                                                                                                                                                                                                                                                                                                                                                                                                                                                                                                    | $ \longrightarrow $                                                                                                                                                                                                           |
| Commented [Sj6]: Deleted.                                                                                                                                                                                                                                                                                                                                                                                                                                                                                                                                                                                                                                                                                                                                                                                                                                                                                                                                                                                                                                                                                                                                                                                                                                     |                                                                                                                                                                                                                               |
| Formatted                                                                                                                                                                                                                                                                                                                                                                                                                                                                                                                                                                                                                                                                                                                                                                                                                                                                                                                                                                                                                                                                                                                                                                                                                                                     | [103]                                                                                                                                                                                                                         |
| Deleted: as its definition (Eq.(1))                                                                                                                                                                                                                                                                                                                                                                                                                                                                                                                                                                                                                                                                                                                                                                                                                                                                                                                                                                                                                                                                                                                                                                                                                           |                                                                                                                                                                                                                               |
| V -                                                                                                                                                                                                                                                                                                                                                                                                                                                                                                                                                                                                                                                                                                                                                                                                                                                                                                                                                                                                                                                                                                                                                                                                                                                           | N 1                                                                                                                                                                                                                           |
| Formatted                                                                                                                                                                                                                                                                                                                                                                                                                                                                                                                                                                                                                                                                                                                                                                                                                                                                                                                                                                                                                                                                                                                                                                                                                                                     | [104]                                                                                                                                                                                                                         |
| Formatted Formatted                                                                                                                                                                                                                                                                                                                                                                                                                                                                                                                                                                                                                                                                                                                                                                                                                                                                                                                                                                                                                                                                                                                                                                                                                                           | [104]                                                                                                                                                                                                                         |
| Formatted
Formatted
Deleted: it                                                                                                                                                                                                                                                                                                                                                                                                                                                                                                                                                                                                                                                                                                                                                                                                                                                                                                                                                                                                                                                                                                                                                                                                                         | [104]                                                                                                                                                                                                                         |
| Formatted
Formatted
Formatted                                                                                                                                                                                                                                                                                                                                                                                                                                                                                                                                                                                                                                                                                                                                                                                                                                                                                                                                                                                                                                                                                                                                                                                                            | [104]
[105]
[106]                                                                                                                                                                                                       |
| Formatted Formatted Deleted: it Formatted Deleted: an                                                                                                                                                                                                                                                                                                                                                                                                                                                                                                                                                                                                                                                                                                                                                                                                                                                                                                                                                                                                                                                                                                                                                                                                         | [104]
[105]
[106]                                                                                                                                                                                                       |
| Formatted
Formatted
Formatted
Deleted: possible eason could be that a larger ∆z ind                                                                                                                                                                                                                                                                                                                                                                                                                                                                                                                                                                                                                                                                                                                                                                                                                                                                                                                                                                                                                                                                                                                                    | [104]
[105]
[106]                                                                                                                                                                                                       |
| Formatted
Formatted
Formatted
Formatted                                                                                                                                                                                                                                                                                                                                                                                                                                                                                                                                                                                                                                                                                                                                                                                                                                                                                                                                                                                                                                                                                                                        | [104]
[105]
[106]
[106]
[108]                                                                                                                                                                                     |
| Formatted
Formatted
Formatted
Formatted
Deleted: the absorbing layer                                                                                                                                                                                                                                                                                                                                                                                                                                                                                                                                                                                                                                                                                                                                                                                                                                                                                                                                                                                                                                                                                       | [104]
[105]
[106]
icat es ₹ 107]
[108]                                                                                                                                                              |
| Formatted         Formatted         Deleted: it         Formatted         Deleted: an         Deleted: possible eason could be that a larger Δz ind         Formatted         Deleted: the absorbing layer         Formatted                                                                                                                                                                                                                                                                                                                                                                                                                                                                                                                                                                                                                                                                                                                                                                                                                                                                                                                                                                                                                                  | [104]
[105]
[106]
[106]
[108]
[108]                                                                                                                                                                            |
| Formatted         Formatted         Deleted: it         Formatted         Deleted: an         Deleted: possible cason could be that a larger Δz ind         Formatted         Deleted: the absorbing layer         Formatted         Deleted: 1                                                                                                                                                                                                                                                                                                                                                                                                                                                                                                                                                                                                                                                                                                                                                                                                                                                                                                                                                                                                               | [104]
[105]
[106]
[106]
[108]
[109]                                                                                                                                                                            |
| Formatted         Formatted         Deleted: it         Formatted         Deleted: an         Deleted: possible cason could be that a larger Δz ind         Formatted         Deleted: the absorbing layer         Formatted         Deleted: 1         Formatted         Deleted: 1                                                                                                                                                                                                                                                                                                                                                                                                                                                                                                                                                                                                                                                                                                                                                                                                                                                                                                                                                                          | [104]
[105]
[106]
[106]
[108]
[109]
[110]                                                                                                                                                                   |
| Formatted         Formatted         Deleted: it         Formatted         Deleted: an         Deleted: possible cason could be that a larger Δz ind         Formatted         Deleted: the absorbing layer         Formatted         Deleted: 1                                                                                                                                                                                                                                                                                                                                                                                                                                                                                                                                                                                                                                                                                                                                                                                                                                                           | [104]
[105]
[106]
[106]
[108]
[109]
[110]                                                                                                                                                                   |
| Formatted         Formatted         Deleted: it         Formatted         Deleted: an         Deleted: possible cason could be that a larger Δz ind         Formatted         Deleted: the absorbing layer         Formatted         Deleted: 1                                                                                                                                                                                                                                                                                                                                                                                                                                                                                                                                                                                                                                                                                                                                                                                                                      | [104]
[105]
[106]
[106]
[108]
[109]
[110]
[111]                                                                                                                                                          |
| Formatted         Formatted         Deleted: it         Formatted         Deleted: an         Deleted: possible cason could be that a larger Δz ind         Formatted         Deleted: the absorbing layer         Formatted         Deleted: 1                                                                                                                                                                                                                                                                                                                                                                                                                                                                                                                                                                                                                             | [104]
[105]
[106]
[106]
[108]
[109]
[110]
[111]                                                                                                                                                          |
| Formatted         Formatted         Deleted: it         Formatted         Deleted: an         Deleted: possible cason could be that a larger Δz ind         Formatted         Deleted: the absorbing layer         Formatted         Deleted: 1         Formatted         Formatted                                                                                                                                                                                                                                                                                                                                                           | [104]
[105]
[106]
[106]
[108]
[109]
[110]
[111]
[111]                                                                                                                                                 |
| Formatted         Formatted         Deleted: it         Formatted         Deleted: an         Deleted: possible cason could be that a larger Δz ind         Formatted         Deleted: the absorbing layer         Formatted         Deleted: 1         Formatted         Deleted: 1         Formatted         Deleted: 1         Formatted         Deleted: 2         Formatted         Deleted: 4         Formatted         Deleted: are not                                                                                                                                                                                                                                                                                                                                                                                                                                                                                                                                                                                                                                                                                                                                                                                                                | [104]
[105]
[106]
[106]
[108]
[109]
[110]
[111]
[111]                                                                                                                                                 |
| Formatted         Formatted         Deleted: it         Formatted         Deleted: an         Deleted: possible eason could be that a larger Δz ind         Formatted         Deleted: the absorbing layer         Formatted         Deleted: 1         Formatted         Deleted: 1         Formatted         Deleted: 2         Formatted         Deleted: 4         Formatted         Deleted: 5         Formatted         Deleted: 4         Formatted         Deleted: 4         Formatted         Deleted: 4         Formatted         Deleted: are not         Formatted                                                                                                                                                                                                                                                                                                                                                                                                                                                                                                                                                                                                                                                                               | [104]
[105]
[106]
[106]
[108]
[109]
[110]
[111]
[112]
[113]                                                                                                                                        |
| Formatted         Formatted         Deleted: it         Formatted         Deleted: an         Deleted: possible cason could be that a larger Δz ind         Formatted         Deleted: the absorbing layer         Formatted         Deleted: 1         Formatted         Deleted: 1         Formatted         Deleted: 2         Formatted         Deleted: 4         Formatted         Deleted: 4         Formatted         Deleted: are not         Formatted         Deleted: in                                                                                                                                                                                                                                                                                                                                                                                                                                                                                                                                                                                                                                                                                                                                                                          | [104]
[105]
[106]
[106]
[108]
[109]
[110]
[111]
[111]
[113]                                                                                                                                        |
| Formatted         Formatted         Deleted: it         Formatted         Deleted: an         Deleted: possible cason could be that a larger Δz ind         Formatted         Deleted: the absorbing layer         Formatted         Deleted: 1         Formatted         Deleted: 1         Formatted         Deleted: 2         Formatted         Deleted: 4         Formatted         Deleted: 1         Formatted         Deleted: are not         Formatted         Deleted: in         Formatted                                                                                                                                                                                                                                                                                                                                                                                                                                                                                                                                                                                                                                                                                                                                                        | [104]
[105]
[106]
[106]
[108]
[109]
[110]
[111]
[111]
[113]
[114]                                                                                                                               |
| Formatted         Formatted         Deleted: it         Formatted         Deleted: an         Deleted: possible cason could be that a larger Δz ind         Formatted         Deleted: the absorbing layer         Formatted         Deleted: 1         Formatted         Deleted: 1         Formatted         Deleted: 2         Formatted         Deleted: 4         Formatted         Deleted: are not         Formatted         Deleted: in         Formatted         Deleted: generally                                                                                                                                                                                                                                                                                                                                                                                                                                                                                                                                                                                                                                                                                                                                                                  | [104]
[105]
[106]
[106]
[108]
[109]
[110]
[111]
[111]
[113]
[114]                                                                                                                               |
| Formatted         Formatted         Deleted: it         Formatted         Deleted: an         Deleted: possible cason could be that a larger Δz ind         Formatted         Deleted: the absorbing layer         Formatted         Deleted: 1         Formatted         Deleted: ine         Formatted         Deleted: in         Formatted         Deleted: in         Formatted         Deleted: in         Formatted         Deleted: ine         Formatted         Deleted: ine                                                                                                                                                                                                                                                                                                                                                                                                                                                                                                                                                                                                                    | [104]
[105]
[106]
[106]
[108]
[109]
[110]
[111]
[111]
[113]
[114]                                                                                                                               |
| Formatted         Formatted         Deleted: it         Formatted         Deleted: an         Deleted: possiblecason could be that a larger Δz ind         Formatted         Deleted: the absorbing layer         Formatted         Deleted: 1         Formatted         Deleted: 1         Formatted         Deleted: 1         Formatted         Deleted: 1         Formatted         Deleted: are not         Formatted         Deleted: in         Formatted         Deleted: ine         Formatted         Deleted: ine         Formatted         Deleted: generally         Deleted: decline         Formatted                                                                                                                                                                                                                                                                                                                                                                                                                                                                                                                                                                                                                                          | [104]
[105]
[106]
[106]
[108]
[109]
[110]
[111]
[111]
[114]
[115]                                                                                                                               |
| Formatted         Formatted         Deleted: it         Formatted         Deleted: an         Deleted: possiblecason could be that a larger Δz ind         Formatted         Deleted: the absorbing layer         Formatted         Deleted: 1         Formatted         Deleted: 1         Formatted         Deleted: 1         Formatted         Deleted: 1         Formatted         Deleted: n         Formatted         Deleted: are not         Formatted         Deleted: in         Formatted         Deleted: generally         Deleted: decline         Formatted         Deleted:         Deleted:         Formatted         Deleted:         Deleted:         Formatted         Deleted:         Deleted:         Formatted                                                                                                                                                                                                                                                                                                                                                                                                                                                                                                                       | [104]
[105]
[106]
[106]
[108]
[109]
[110]
[111]
[111]
[112]
[114]                                                                                                                               |
| Formatted         Formatted         Deleted: it         Formatted         Deleted: an         Deleted: possiblecason could be that a larger Δz ind         Formatted         Deleted: the absorbing layer         Formatted         Deleted: 1         Formatted         Deleted: 1         Formatted         Deleted: 1         Formatted         Deleted: are not         Formatted         Deleted: in         Formatted         Deleted: in         Formatted         Deleted: ins         Formatted         Deleted: coline         Formatted         Deleted: decline         Formatted         Deleted:         Formatted         Deleted:         Formatted         Deleted:         Formatted         Deleted:         Formatted         Deleted:         Formatted         Deleted:         Formatted                                                                                                                                                                                                                                                                                                                                                                                                                                               | [104]
[105]
[106]
[108]
[108]
[109]
[110]
[111]
[111]
[111]
[114]
[115]
[116]                                                                                                             |
| Formatted
Formatted
Formatted
Formatted
Formatted
Formatted
Formatted
Formatted
Formatted
Formatted
Formatted
Formatted
Formatted
Formatted
Formatted
Deleted: . where it was found that the retrieved AAI co                                                                                                                                                                                                                                                                                                                                                                                                                                                                                                                                                                                                                                                                               | [104]
[105]
[106]
[108]
[108]
[109]
[110]
[111]
[111]
[112]
[114]
[115]
[116]
uld ½ [117]                                                                                              |
| Formatted         Formatted         Deleted: it         Formatted         Deleted: an         Deleted: possiblecason could be that a larger Δz ind         Formatted         Deleted: the absorbing layer         Formatted         Deleted: 1         Formatted         Deleted: 1         Formatted         Deleted: 1         Formatted         Deleted: 1         Formatted         Deleted: n         Formatted         Deleted: are not         Formatted         Deleted: in         Formatted         Deleted: generally         Deleted: decline         Formatted         Deleted: .This is         Formatted         Deleted: .This is         Formatted         Deleted: .the is found that the retrieved AAI co         Formatted                                                                                                                                                                                                                                                                                                                                                                                                                                                                                                                | [104]
[105]
[106]
[108]
[108]
[109]
[109]
[110]
[111]
[111]
[112]
[114]
[115]
[115]
[116]                                                                                           |
| Formatted         Formatted         Deleted: it         Formatted         Deleted: an         Deleted: possiblecason could be that a larger Δz ind         Formatted         Deleted: the absorbing layer         Formatted         Deleted: 1         Formatted         Deleted: 1         Formatted         Deleted: 1         Formatted         Deleted: 1         Formatted         Deleted: are not         Formatted         Deleted: in         Formatted         Deleted: generally         Deleted: .This is         Formatted         Deleted: .This is         Formatted         Deleted: .This is         Formatted         Deleted: .the retrieved AAI co                                                                                                                                                                                                                                                                                                                                                                                                                                                                                                                                                                                        | [104]
[105]
[106]
[108]
[108]
[109]
[110]
[111]
[111]
[112]
[114]
[115]
[115]
[116]
uld ½[117]
[118]
[119]                                                                    |
| Formatted         Formatted         Deleted: it         Formatted         Deleted: an         Deleted: possible cason could be that a larger Δz ind         Formatted         Deleted: the absorbing layer         Formatted         Deleted: 1         Formatted         Deleted: 1         Formatted         Deleted: 1         Formatted         Deleted: 1         Formatted         Deleted: are not         Formatted         Deleted: in         Formatted         Deleted: elemenally         Deleted:ensis         Formatted         Deleted:                                                                                                                                                                                                                                                                                                                                                                                                                                                                                                                                                                                                                                                                                                        | [104]
[105]
[106]
[108]
[108]
[109]
[110]
[111]
[111]
[112]
[113]
[114]
[115]
[115]
[116]
uld ½ [117]
[118]
[119]
[120]                                                 |
| Formatted         Formatted         Deleted: it         Formatted         Deleted: an         Deleted: possible cason could be that a larger Δz ind         Formatted         Deleted: the absorbing layer         Formatted         Deleted: 1         Formatted         Deleted: 1         Formatted         Deleted: 1         Formatted         Deleted: 1         Formatted         Deleted: are not         Formatted         Deleted: in         Formatted         Deleted: generally         Deleted: .This is         Formatted         Deleted: .This is         Formatted         Deleted: , where it was found that the retrieved AAI co         Formatted         Formatted         Deleted: , where it was found that the retrieved AAI co         Formatted         Formatted         Formatted                                                                                                                                                                                                                                                                                                                                                                                                                                                | [104]
[105]
[106]
[108]
[108]
[109]
[110]
[111]
[111]
[112]
[114]
[115]
[115]
[115]
[116]
uld ½º [117]
[118]
[119]
[120]
[121]                                       |
| Formatted         Formatted         Deleted: it         Formatted         Deleted: an         Deleted: possible cason could be that a larger Δz ind         Formatted         Deleted: the absorbing layer         Formatted         Deleted: 1         Formatted         Deleted: 1         Formatted         Deleted: 1         Formatted         Deleted: 1         Formatted         Deleted: are not         Formatted         Deleted: in         Formatted         Deleted: elemently         Deleted: cloine         Formatted         Deleted: This is         Formatted         Deleted: where it was found that the retrieved AAI co         Formatted                                                                                                                                                                                                                                                                                                                                                                                                         | [104]
[105]
[106]
[108]
[108]
[109]
[110]
[111]
[111]
[111]
[112]
[115]
[115]
[116]
uld hg [117]
[118]
[120]
[121]                                                         |
| Formatted         Formatted         Deleted: it         Formatted         Deleted: an         Deleted: possible cason could be that a larger Δz ind         Formatted         Deleted: the absorbing layer         Formatted         Deleted: 1         Formatted         Deleted: 1         Formatted         Deleted: 1         Formatted         Deleted: 1         Formatted         Deleted: are not         Formatted         Deleted: in         Formatted         Deleted: eleme         Formatted         Deleted: sequerally         Deleted: This is         Formatted         Deleted: where it was found that the retrieved AAI co         Formatted                                                                                                                                                                                                                                                                                                                                                                                                                           | [104]
[105]
[106]
[106]
[108]
[108]
[109]
[110]
[111]
[111]
[112]
[114]
[114]
[115]
[115]
[116]
uld hg [117]
[118]
[119]
[120]
[122]
[122]                     |
| Formatted         Formatted         Deleted: it         Formatted         Deleted: an         Deleted: possible cason could be that a larger Δz ind         Formatted         Deleted: the absorbing layer         Formatted         Deleted: 1         Formatted         Deleted: 1         Formatted         Deleted: 1         Formatted         Deleted: 1         Formatted         Deleted: are not         Formatted         Deleted: in         Formatted         Deleted: eleme         Formatted         Deleted: cline         Formatted         Deleted: This is         Formatted         Deleted: where it was found that the retrieved AAI co         Formatted         Formatted <th> [104]
 [105]
 [106]
 [108]
 [108]
 [109]
 [110]
 [111]
 [111]
 [112]
 [114]
 [115]
 [115]
 [115]
 [115]
 [116]
 [116]
 [117]
 [118]
 [119]
 [120]
 [121]
 [122]
 [123]</th>                                       | [104]
[105]
[106]
[108]
[108]
[109]
[110]
[111]
[111]
[112]
[114]
[115]
[115]
[115]
[115]
[116]
[116]
[117]
[118]
[119]
[120]
[121]
[122]
[123]          |
| Formatted         Formatted         Deleted: it         Formatted         Deleted: an         Deleted: possible cason could be that a larger Δz ind         Formatted         Deleted: the absorbing layer         Formatted         Deleted: 1         Formatted         Deleted: 1         Formatted         Deleted: 1         Formatted         Deleted: 1         Formatted         Deleted: are not         Formatted         Deleted: in         Formatted         Deleted: eleme         Formatted         Deleted: eleme         Formatted         Deleted: decline         Formatted         Deleted: This is         Formatted         Deleted: where it was found that the retrieved AAI co         Formatted                                                                                                                                                                                                                                                                                                                                                 | [104]
[105]
[106]
[108]
[108]
[109]
[110]
[111]
[111]
[112]
[113]
[114]
[114]
[115]
[115]
[116]
uld ½ [117]
[118]
[119]
[120]
[121]
[123]                      |
| Formatted         Formatted         Deleted: it         Formatted         Deleted: an         Deleted: possible cason could be that a larger Δz ind         Formatted         Deleted: the absorbing layer         Formatted         Deleted: 1         Formatted         Deleted: 1         Formatted         Deleted: 1         Formatted         Deleted: 1         Formatted         Deleted: are not         Formatted         Deleted: in         Formatted         Deleted: generally         Deleted: This is         Formatted         Deleted: where it was found that the retrieved AAI co         Formatted         Deleted: , where it was found that the retrieved AAI co         Formatted         Formatted <th> [104]
 [105]
 [106]
 [108]
 [108]
 [109]
 [110]
 [110]
 [111]
 [111]
 [112]
 [113]
 [114]
 [114]
 [115]
 [115]
 [115]
 [116]
 [116]
 [117]
 [118]
 [120]
 [121]
 [123]
 [124]</th> | [104]
[105]
[106]
[108]
[108]
[109]
[110]
[110]
[111]
[111]
[112]
[113]
[114]
[114]
[115]
[115]
[115]
[116]
[116]
[117]
[118]
[120]
[121]
[123]
[124] |

**3 Methodology and datasets**

[revised manuscript text omitted]

| Å  | Formatted: English (US)                                                                                                                                                                                                                                                                            |
|----|----------------------------------------------------------------------------------------------------------------------------------------------------------------------------------------------------------------------------------------------------------------------------------------------------|
| 4  | Deleted: http://dx.doi.org/10.5067/aura/omi/data2001                                                                                                                                                                                                                                               |
| -( | Deleted: by                                                                                                                                                                                                                                                                                        |
| -( | Formatted: Font: 10 pt, English (US)                                                                                                                                                                                                                                                               |
| -( | Formatted: Font: 10 pt, English (US)                                                                                                                                                                                                                                                               |
| -( | Deleted: measurement                                                                                                                                                                                                                                                                               |
| Y  | Formatted [134]                                                                                                                                                                                                                                                                                    |
| (  | Deleted: s                                                                                                                                                                                                                                                                                         |
| Y  | Formatted: Font: 10 pt, English (US)                                                                                                                                                                                                                                                               |
|    | Deleted: ares smaller than 60°, and if ground pixels are not
contaminated by sun-glint, clouds, row anomalies of the instrument,
etc. The simulation is only applied to ground plumeixels inside the
biomass burning plume, which ares defined for both OMI and
GOME-2 retrieved [135] |
| Y  | Deleted: ,                                                                                                                                                                                                                                                                                         |
| Y  | Formatted [136]                                                                                                                                                                                                                                                                                    |
| ľ  | Deleted: and AERONET                                                                                                                                                                                                                                                                               |
| ľ  | Deleted: aaily global coverage per 1 to 2 days The spatial resolution can reach 1 km [137]                                                                                                                                                                                                         |
| Y  | Formatted [138]                                                                                                                                                                                                                                                                                    |
| ľ  | Deleted:         the spectrum ranges fromt has 36 spectral bands in the wavelength range between 000 nm andto         [139]                                                                                                                                                                        |
| Y  | Formatted [140]                                                                                                                                                                                                                                                                                    |
| ľ  | Deleted:                                                                                                                                                                                                                                                                                           |
| Y  | Formatted [141]                                                                                                                                                                                                                                                                                    |
| Ì  | Commented [JV13]: add collection number                                                                                                                                                                                                                                                            |
| ľ  | Commented [Sj14R13]: Added                                                                                                                                                                                                                                                                         |
| ľ  | Formatted: Font: 5.5 pt                                                                                                                                                                                                                                                                            |
| Y  | Formatted [142]                                                                                                                                                                                                                                                                                    |
| Y  | Formatted [143]                                                                                                                                                                                                                                                                                    |
| Y  | Deleted: .                                                                                                                                                                                                                                                                                         |
| Y  | Formatted [144]                                                                                                                                                                                                                                                                                    |
|    | Deleted: The τ retrieved from AERONET also has to be converted to 550 nm to make them comparable.                                                                                                                                                                                                  |
| Y  | Formatted [145]                                                                                                                                                                                                                                                                                    |
|    |                                                                                                                                                                                                                                                                                                    |

| 1    | 3.1.3 AERONET aerosol properties                                                                                                                                                                                                                                                                                                                                                                                                                                                                                                                                                                                                                                                                                                                                                                                                                                                                                                                                                                                                                                                                                                                                                                                                                                                                                                                                                                                                                                                                                                                                                                                                                                                                                                                                                                                                                                                                                                                                                                                                                                                                                              |                                 | Formatted: Font: 10 pt, English (US)                                                                          |                            |
|------|-------------------------------------------------------------------------------------------------------------------------------------------------------------------------------------------------------------------------------------------------------------------------------------------------------------------------------------------------------------------------------------------------------------------------------------------------------------------------------------------------------------------------------------------------------------------------------------------------------------------------------------------------------------------------------------------------------------------------------------------------------------------------------------------------------------------------------------------------------------------------------------------------------------------------------------------------------------------------------------------------------------------------------------------------------------------------------------------------------------------------------------------------------------------------------------------------------------------------------------------------------------------------------------------------------------------------------------------------------------------------------------------------------------------------------------------------------------------------------------------------------------------------------------------------------------------------------------------------------------------------------------------------------------------------------------------------------------------------------------------------------------------------------------------------------------------------------------------------------------------------------------------------------------------------------------------------------------------------------------------------------------------------------------------------------------------------------------------------------------------------------|---------------------------------|---------------------------------------------------------------------------------------------------------------|----------------------------|
|      | APPONITY is an example of standard strengthened and the standard strengthened in the strengthened and | $\langle - \rangle$             | Deleted: micro-physical parameters                                                                            |                            |
| 1.70 | AERONE I is an aerosol monitoring network of ground-based sun photometers. With standardized instruments, calibration,                                                                                                                                                                                                                                                                                                                                                                                                                                                                                                                                                                                                                                                                                                                                                                                                                                                                                                                                                                                                                                                                                                                                                                                                                                                                                                                                                                                                                                                                                                                                                                                                                                                                                                                                                                                                                                                                                                                                                                                                        | $\langle \rangle$               | Formatted: English (US)                                                                                       |                            |
| 170  | processing and distribution, AERONET provides a long-term global database for aerosol research and air-borne and space-

---

## Referee Report (RR1)

**Referee Report**

This manuscript presents a technique to retrieve aerosol SSA from satellite data by simulating near UV AAI and constraining AOD. This is similar to- and draws from the same philosophical lines as the study of Satheesh et al., 2009. The authors have chosen to retrieve SSA at 550 nm by using MODIS-AOD at 550 nm from the simulated AAI. This makes the UV-Vis spectral dependency of aerosols under investigation a key parameter. Since the simulation of AAI is driven by aerosol microphysics, complex refractive index from the nearest AERONET site which determines this spectral dependency. This raises a flag on its representativeness on the simulation and in-turn the retrieved properties far away from the site (at least for the case presented here – smoke plume travels up to 3000 km away from source fires and AERONET site). I think the authors should include some discussion in the manuscript about this representativeness and reliability of technique for studying such long-range transport of aerosols.

Few general points (in no specific order) the author should take care of, before the paper is ready for a final publication.

1) The purpose of the study mentioned in L 73 – 74, should also be presented in the abstract before they say "In the first part of this study…….and later we present…." with the reference wavelength of the SSA retrieval.
2) Line 17 : This sentence sounds like it is the instrument's deficiency. It is not. Please make it clear. Something like "The CALIOP overpasses over the region failed to capture the complete evolution…."
3) Line 21 : replace 'observations' with 'satellite observations'
4) Line 69 – 71 : Poorly phrased. Please re-write it.
5) Line 108 :'linearly' should be 'linear'.
6) Line 207 – 209 : Poorly phrased. Please re-write it.
7) Line 234 – 235 : I am not sure what the authors are saying here. Firstly, the Santiago_Beauchef site has near-UV AERONET measurements. Secondly, the aerosol complex refractive index is provided in their Inversion product that is always only for 440, 675, 880 and 1020 nm. So, what does "site only covers the visible band" and "absence of observations" mean there ?
8) Line 286 : Please mention the wavelength of SSA retrieval.
9) Line 295 : What do mean by "may even fail to capture the elevated plume" ?
10) Line 305, 307 : It should be '... for 26 January ….'.
11) Line 306 : low RMSE ? Not really. For a quantity like AAI which is varying from 0 – 4, an RMSE of 0.5 or 0.6 is not low. Further the measures RMSE and correlation alone cannot confirm an agreement or disagreement between model and observations. Probably one could try using a T-test and say at what confidence level the agreement is statistically significant, I am not sure if your sample size allows for it ?
12) In Table 2, please also provide RMSE for the SSA to be consistent with the AAI report.
13) Please provide the sample size (number of pixels) for which statistics are derived in the Table 2 and also in figure 8 (i – l).
14) Line 313 : Is this 'accounting' or 'contributing' ?

15) Line 319 – 321 : I think, the authors should also report absolute difference in SSA with the AERONET before drawing parallels with the study mentioned here. It is confusing to compare percent difference in ni and absolute difference in SSA.

16) The spectral dependency retrieved in the study should also be compared with the in-situ measurements and other studies on biomass burning in the literature and include in the discussion (L 320 – 330).

17) Line 374 : "The retrieved SSA is out of typical uncertainty…." this statement does not make sense and should be removed or re-phrase appropriately. At least for the case presented here the authors did not come up with a quantitative estimate of the uncertainty in SSA and clearly it is difficult if not impossible because of the multiple error sources.

---

## Author Response (AR2)

This document contains the responses to the referees' comment on the manuscript version 3, followed by a version 4 manuscript (starts in page 8). The reviewers' comments and questions are in bold. For each comment / question, the authors' reply / answer is in black, and the corresponding modifications in the manuscript version 3 are marked in blue colour. The line index mentioned in the response correspond to the line number in manuscript version 4 (without markups)

**(1) Author's response to Anonymous Referee #1**
Thank you very much for your comments. The following are the responses to your specific comments.

**Summary**
**This manuscript uses a case study to illustrate the derivation of the aerosol single scattering albedo (SSA) using observations from the OMI sensor of a large biomass-burning event in south Chile in 2017.**
**UV remote sensing of aerosols has not received the same level of attention as other sensors (such as MODIS/VIIRS) and studies like this one are a welcome addition to the general body of knowledge regarding the measurement of aerosol absorption from space. The paper is well laid out and its novelty resides in the combination of sensors utilized, its focus on single scattering albedo and aerosol height retrievals (as opposed to AOD and SSA) and its application using OMI observations. Overall, I think this paper can be published in its present form after some clarifications and corrections are included (mostly technicallso, I second the Editor's comments that the Satheesh et al and the Gassó and Torres papers should be consider in this analysis plus the lack of mentioning of the recent SSA studies using OMI done by Jethva and Torres.**
Thanks for your advice. We already have added the literatures you mentioned to the manuscript.

Satheesh et al. (2009) is referred at the following locations:
'But this aerosol absorption over near-UV is highly sensitive to the assumption on aerosol layer height. Satheesh et al. (2009) therefore used the $\tau$ from MODerate-resolution Imaging Spectroradiometer (MODIS), which is independent of aerosol layer height, to constrain the OMAERUV retrieval.' (Line 44-46)

'Besides, the MODIS retrieved $\tau$ is free from the uncertainty triggered by assumed aerosol profile (Satheesh et al., 2009).' (Line 193-194)

Gassó and Torres (2016) is referred at the following locations:
'This OMI-MODIS joint retrieval was also evaluated by Gassó and Torres (2016). They found under less absorbing condition, the hybrid method is sensitive to the variations in the input $\tau$, which is used to select the retrieved pair of aerosol layer and $\omega_0$.' (Line 48-50)

'It is noted that the observed aerosol vertical distribution is limited for the Chile wildfires. Previous research suggested AAI cannot be quantitatively used without $\tau$ or $z_{aer}$ information (Gassó and Torres, 2016).' (Line 268-269)

Jethva and Torres (2011) is referred at the following locations:
'We thus find a much weaker wavelength dependence than the value in Jethva and Torres (2011) study, where a 20% difference between the two UV wavelengths was applied to OMAERUV algorithm to achieve the result that 70% of the retrieved $\omega_0$ differ less than $\pm 0.03$ from the $\omega_0$ from the AERONET measurements.' (Line 329-331)

'From the sensitivity study of Jethva and Torres (2011), a stronger spectral dependence of $n_i$ between 354 and 388 nm would allow simulations to reach a higher AAI while keeping $n_i$ at a relatively low level. In our study, this means to retrieve a higher $\omega_0$ at 550 nm.' (Line 338-340)

**Details**
**L325-326: the explanation provided does not make much sense to me. If there is little sensitivity to deltaZ , why the claim that there is an amplification of absorption in the layer?. My interpretation is that if there is an amplification of absorption, the AAI increases which means there is sensitivity to deltaZ. Please, revise the sentence.**
Sorry for the misunderstanding. We just wanted to claim that $\Delta z$ has an effect on AAI value though it is very limited. We have rephrased into: 'The reason could be that a larger $\Delta z$ indicates the coming sunlight has a higher possibility to be absorbed by aerosols, slightly enhancing the aerosol absorption. Although the sensitivity exists, the impact is only up to 5%, which is negligible for practical purposes.' (Line 139-141)

**Line 332-333: A similar conclusion was reached by Colarco et al (2017) when analyzing the sensitivity of OMI UVAI .**
**Colarco, P. R., Gassó, S., Ahn, C., Buchard, V., da Silva, A. M., and Torres, O.: Simulation of the Ozone Monitoring Instrument aerosol index using the NASA Goddard Earth Observing System aerosol reanalysis products, Atmos. Meas. Tech., 10, 4121-4134, https://doi.org/10.5194/amt-10-4121-2017, 2017.**

Thank you for your advice. We have added to the manuscript: 'As a result, the AAI decreases significantly due to smaller $\Delta l_{\lambda_1}$, in agreement with a previous study (de Graaf et al., 2005; Colarco et al., 2017).' (Line 147-148)

65 **Lines 343-347: It should be mentioned that while the sensitivity is exists, in this case, it is very small; color range in the respective figure is from 0 to 1% which is negligible for practical purposes.**
There might be a misunderstanding here. We assume you are referring to Fig.6. The figure shows the normalized parameter, namely the value is divided by the maximum among all the cases in the comparison group. In terms of percentage, the difference due to different measuring geometry can be up to 20% (i.e. $(1-0.8) * 100\%$), which is not negligible. This
70 dependence on measuring geometry can also be explained from the view of AAI's definition. AAI is directly calculated from the measuring radiance, which is a parameter relies on the directions of incident lights and measuring angles. That is exactly the reason we have to use the same measuring geometry as OMI satellite to simulate the AAI.

However, this suggestion can be applied to the sensitivity to the $\Delta z$. As shown in Fig.4, the difference due to different $\Delta z$ is
75 only up to 5% (i.e. $(1-0.95) * 100\%$).

**Section 3.1.1 Include the time difference between GOME-2 and OMI overpasses the respective type of orbits (ascending/descending)**
We have added it to the manuscript: 'Note the GOME-2 and OMI have different equator crossing time (9:30 LT descending
80 node for GOME-2 and 13:45 LT ascending node for OMI) that may affect the inter-comparison of the two satellite measurements.' (Line 174-175)

**Line 378: please remove this sentence unless you can make a more convincing case. The data set used in this study is not good enough for making assessments of the MODIS SSA assumption (which certainly needs to be addressed). You**
85 **need at least well-collocated CALIPSO and Aeronet data. This does not mean that the comments regarding MODIS biases should be removed. The results are suggestive at most and it should be included.**
Thank you for your correction. We simply have rephrased into: 'In addition, the $\tau$ measured by OMI and AERONET are compared with MODIS.' (Line 197)

90 **Line 444: what is a " ground pixel"? in this context it seems like the word pixel is enough.**
We have changed accordingly. 'It should be noted that for all pixels in the plume we assume the same aerosol microphysical properties as well as the same aerosol layer height.' (Line 275-276)

**Line 540: This method does not "retrieve an aerosol profile", it retrieves an average aerosol height. Please correct this**
95 **case and other that are included in the paper.**
Sorry for the misunderstanding. We have rephrased them into proper statements.

**Figure 2: I see very little value in including a single-phase function in the figure. At reference to a paper should be enough and publication charges will be saved. Alternatively, you may want include additional phase functions to**
100 **point out differences between aerosol models.**
To your question, the phase function is actually range from 0 to 180 degree with 1degree resolution as the continuous line in Fig.2, the marker just indicates the scattering angle we are interested.
Since in this study our focus is on how the AAI response to a specific phase function and the measuring geometry rather than how the aerosol microphysical parameters determine the phase function, the discussion in the sensitivity studies is
105 exclusively on the default case. We prefer to mention other phase functions in the Appendix A as shown in Fig.A1 to A3 (also attached below).

[Figure]

**Figure.A1: Phase function p(Θ) at 354 nm of the parameterized Mie scattering aerosol in sensitivity analysis as a function of $r_g$ (with $n_r$ = 1.5 and $n_i$ = 0.06). The markers in the plot correspond to values when Θ=60°, 90°, 120°, 150°, 180°.**

[Figure]

**Figure.A2: Phase function p(Θ) at 354 nm of the parameterized Mie scattering aerosol in sensitivity analysis as a function of $n_r$ (with $r_g$ = 0.15 μm and $n_i$ = 0.06). The markers in the plot correspond to values when Θ=60°, 90°, 120°, 150°, 180°.**

[Figure]

**Figure.A3: Phase function p(Θ) at 354 nm of the parameterized Mie scattering aerosol in sensitivity analysis as a function of $n_i$ (with $r_g$ = 0.15 μm and $n_r$ = 1.5). The markers in the plot correspond to values when Θ=60°, 90°, 120°, 150°, 180°.**

**Figure 5: a) and b) seem to be inverted**
Thank you for the correction. We have changed accordingly.

**Figure 6a: Put the same ticks in both x and y axis.**
Thank you for the correction. We have changed accordingly.

**Figure 6: caption does not include reference or explanation of figure 6b.**
Thank you for the correction. We have changed the figure caption into: 'Figure.6 AAI sensitivity to θ and $θ_0$ at φ=180˚. The black dashed contour in (a) indicates the Θ=60˚, 90˚, 120˚, 150˚. The white dashed line in (a) indicates the cross sections, with its corresponding normalized AAI, $\Delta I_{\lambda 1}$ and $I_{\lambda 1}^{obs}$ in (b).' (Line 657-658)

**Figure 7: since the colorbar is not the standard colorbar used in the official CALIPSO quicklooks, can you clarify if the dark red are cloud? The contrast between in smoke and cloud appears to be between orange and red but a clarification will be appreciated.**
We have clarified the caption accordingly: 'Figure.7 CALIOP backscatter coefficient β at 532 nm. The solid and dashed line indicate the retrieved $z_{aer}$ and Δz, respectively. The red to black dots indicate clouds and the orange dots indicate aerosol layers, respectively.' (Line 676-677)

**Figure 8 Caption: nowhere in the caption mentions the word AI. Is this the parameter displayed?**
We have added accordingly: 'Figure.8 AAI from OMI observations (a–d) and DISAMAR simulations (e–h) of the Chile wildfires on 26, 27, 29 and 30 January 2017. The black and red cross symbols are the AERONET station and the main fire sources (Pichilemu W34.39˚ S72.00˚ and Consititución S35.33˚, W72.42˚), respectively. The grey dashed line indicates the CALIOP paths in the region of interest, where the paths used to validate the plume height are marked by black dashed line. The scatter plots (i–l) present the OMI observations against DISAMAR simulations for only qualified data (red dot) and all data (blue dot), respectively.' (Line 687-691)

**(2) Author's response to Anonymous Referee #3**
Thank you very much for your comments to our manuscript. The following are the responses to your specific comments.

**This manuscript presents a technique to retrieve aerosol SSA from satellite data by simulating near UV AAI and constraining AOD. This is similar to and draws from the same philosophical lines as the study of Satheesh et al., 2009. The authors have chosen to retrieve SSA at 550 nm by using MODIS-AOD at 550 nm from the simulated AAI. This makes the UV-Vis spectral dependence of aerosols under investigation a key parameter. Since the simulation of AAI is driven by aerosol microphysics, complex refractive index from the nearest AERONET site which determines this spectral dependence. This raises a flag on its representativeness on the simulation and in-turn the retrieved properties far away from the site (at least for the case presented here – smoke plume travels up to 3000 km away from source fires and AERONET site). I think the authors should include some discussion in the manuscript about this representativeness and reliability of technique for studying such long-range transport of aerosols.**

Thank you for the advice. In the manuscript, we have put emphasis on the fact that the location of the AERONET site is not in the source area, while little attention was paid on the spatial representativeness of the AERONET data. We only briefly mentioned the location difference between AERONET and the plume detected by the satellite. Here we added discussion on the spatial representativeness of the AERONET:

In section 4 Results and discussion:
'Last but not least, the spatial representation of the in situ instrument also concerns. Santese et al. (2007) showed that the selected AERONET aerosol parameters can be representative of a $300\times300$ km$^2$ southeast Italy area. For the Chile wildfires with the most remote pixel over 3000 km away from the continent, the measurements at AERONET cannot fully represent the plume detected by satellites.' (Line 348-351)

In section 5 Conclusions:
'the insufficient spatial representativeness of a single AERONET site;' (Line 395)

'It is also reliable to retrieve aerosol absorption for each individual pixel with constraint of the aerosol layer height product. The problem due to the poor spatial representativeness of in situ measurement can then be eased by comparing with the nearby satellite pixels.' (Line 404-407)

**Few general points (in no specific order) the author should take care of, before the paper is ready for a final publication.**
**1) The purpose of the study mentioned in L 73 – 74, should also be presented in the abstract before they say "In the first part of this study.......and later we present...." with the reference wavelength of the SSA retrieval.**
We have added the purpose accordingly: 'In this study, we attempt to quantify the aerosol absorption by retrieving the single scattering albedo ($\omega_0$) at 550 nm from the satellite measured AAI in near-UV channel.' (Line 10-11)

**2) Line 17 : This sentence sounds like it is the instrument's deficiency. It is not. Please make it clear. Something like "The CALIOP overpasses over the region failed to capture the complete evolution...."**
Thank you for the correction. We have rephrased into: 'The Cloud and Aerosol Lidar with Orthogonal Polarization (CALIOP) overpasses failed to capture the complete evolution of the smoke plume over the research region, therefore the aerosol profile is parameterized.' (Line 17-19)

**3) Line 21 : replace 'observations' with 'satellite observations'**
Thank you for the correction. We have rephrased into: 'The results show that the AAI simulated by DISAMAR is consistent with satellite observations.' (Line 21-22)

**4) Line 69 – 71 : Poorly phrased. Please re-write it.**
Sorry for the misunderstanding. We have rephrased into: 'This study is inspired by previous research to quantify the aerosol absorption from AAI. We use the near-UV AAI provided by OMI on-board Aura, the successor of TOMS, to derive the aerosol properties of the central Chile (Pichilemu 34.39°S, 72.00°W and Consititución 35.33°S, 72.42°W) wildfires in January 2017. The series of fires were triggered by a combination of long-term drought and high temperature, and was regarded as the worst wildfire season in the national history (The Guardian, 2017).' (Line 68-72)

**5) Line 108 :'linearly' should be 'linear'.**

Thank you for the correction. We have rephrased into: 'with only fine mode particles and weak linear wavelength

200 dependence of the complex refractive index ($n_r$ and $n_i$).' (Line 109 -110)

**6) Line 207 – 209 : Poorly phrased. Please re-write it.**
Sorry for the misunderstanding. We have rephrased into: 'Note that the AERONET level 1.5 dataset is not quality-assured.

205 In addition, the location of this AERONET site is in downtown of Santiago City and close to major roads, where the

presence of scattering urban aerosols may bias the measurements of the plume.'(Line 213-215)

**7) Line 234 – 235 : I am not sure what the authors are saying here. Firstly, the Santiago_Beauchef site has near-UV AERONET measurements. Secondly, the aerosol complex refractive index is provided in their Inversion product that is always only for 440, 675, 880 and 1020 nm. So, what does "site only covers the visible band" and "absence of**
210 **observations" mean there ?**
Sorry for the misunderstanding. 'only covers the visible band' refers to that the site only has sky radiance measurements

since visible band. In other word, the inversion product of this site is only available in visible and longer wavelength range.

'absence of observations' therefore indicates the absence of observations of microphysical parameter in UV band. We have

rephrased into a clearer way: 'The AERONET instrument at this site only cover the visible and infrared band (440 nm to

215 1020 nm) for sky radiance measurements, i.e. no aerosol inversion products at UV band. Due to the absence of observations,

assumptions have to be made on the spectral dependence of aerosol properties to obtain their values in the near-UV band.'

(Line 240-242)

**8) Line 286 : Please mention the wavelength of SSA retrieval.**
220 Thank you for the correction. We have added the wavelength at which the SSA is retrieved to the manuscript: 'By applying

the methodology described in the previous section, we quantitatively retrieved the aerosol profile and $\omega_0$ at 550 nm of the

Chile 2017 wildfires by AAI simulation.' (Line 295-296)

**9) Line 295 : What do mean by "may even fail to capture the elevated plume" ?**
225 As shown in Fig.8(a), the CALIOP overpass we selected (the black dashed line) is on the east side of the plume, while the

elevated plume detected by OMI was blown away from the continent to the open ocean. From Fig.7(a), one can also see that

the CALIOP overpass only captures the smoke at the source (near surface rather than elevated into sky). We mentioned this

in order to emphasize that OMI and CALIOP did not always measure the same target due to spatial coverage and measuring

time difference. To make it clearer, we have rephrased it into: 'It is noted that due to the spatial coverage and the measuring

230 time difference, CALIOP are not able to represent the entire plume captured by the OMI.' (Line 303-304)

**10) Line 305, 307 : It should be '... for 26 January ....'.**

Thank you for the correction. We have rephrased into:

'Except for 26 January, the median of simulated AAI in other cases is in good agreement with the measurements, with
235 relative differences within $\pm6\%$.' (Line 315-316)
'The majority of the simulated AAI for 26 January is negatively biased, which is reflected by the small slope without an

intercept correction in Fig.8 (i)' (Line 317-318)

**11) Line 306 : low RMSE ? Not really. For a quantity like AAI which is varying from 0 – 4, an RMSE of 0.5 or 0.6 is**
240 **not low. Further the measures RMSE and correlation alone cannot confirm an agreement or disagreement between model and observations. Probably one could try using a T-test and say at what confidence level the agreement is statistically significant, I am not sure if your sample size allows for it ?**
Thank you for your correction. To some extent, a RMSE of 0.5 for AAI is not low but just acceptable, considering the assumption of homogenous aerosol properties over the entire plume and the lack of aerosol profile information. The sample
245 size is provided in Table 2. The sample size for this event is indeed limited. But we appreciate your advice and may apply statistical tests in future cases where more sample points available.
We have rephrased into: 'The RMSE is only acceptable reflects that part of the plume cannot be fit by the assumptions in the forward simulation.' (Line 316-317)

250 **12) In Table 2, please also provide RMSE for the SSA to be consistent with the AAI report.**

As we assumed the homogeneous properties of the plume, i.e. all pixels in the plume have the same SSA. The retrieved SSA is an average level for the entire plume and is compared to the AERONET measurement. In Table 2, we already have provided the absolute and relative difference between retrieved and measured SSA, respectively.

**13) Please provide the sample size (number of pixels) for which statistics are derived in the Table 2 and also in figure 8 (i – l).**
Thank you for the reminding. We have added in Table.2.

**14) Line 313 : Is this 'accounting' or 'contributing' ?**
Thank you for the correction. We have rephrased into: 'There are many sources contributing to this discrepancy in $\omega_0$.' (Line 323)

**15) Line 319 – 321 : I think, the authors should also report absolute difference in SSA with the AERONET before drawing parallels with the study mentioned here. It is confusing to compare percent difference in ni and absolute difference in SSA.**
Sorry for the confusion. To be clear, the percent in this part is not the relative difference of imaginary refractive index, but the difference of the imaginary refractive index between two wavelengths, i.e. 354 and 388 nm. This relative difference between the two wavelength indicates the spectral dependence of the imaginary refractive index in the near-UV channel. This dependence further determines the relation between the near-UV AAI and the visible SSA: the larger this spectral dependence, the higher SSA can be retrieved at the same AAI level, and the other way around for weaker spectral dependence. We have rephrased into: 'This would also affect the spectral dependence of complex refractive index used to constrain the radiative transfer calculation.' (Line 325-326).

The following part then discusses how this wavelength dependence affects the AAI simulation and SSA retrieval.

**16) The spectral dependence retrieved in the study should also be compared with the in- situ measurements and other studies on biomass burning in the literature and include in the discussion (L 320 – 330).**
Thank you for the advice. We have added the following content in the discussion: 'We thus find a much weaker wavelength dependence than the value in Jethva and Torres (2011) study, where a 20% difference between the two UV wavelengths was applied to OMAERUV algorithm to achieve the result that 70% of the retrieved $\omega_0$ differ less than $\pm 0.03$ from the $\omega_0$ from the AERONET measurements. This 20% spectral dependence adopted in their work is associated with findings of Hoffer et al. (2006). They conducted in situ measurements on humic-like substances (HULIS) of Amazonia biomass burning aerosols and found that around 35% - 50% light absorption at 300 nm, whereas only around 15% at 400 nm. In terms of the absorbing Ångström exponent (AÅE), a 20% increase at 354 nm with respect to the value at 388 nm is equivalent to an AÅE value between 2.5 and 3, depending on the aerosol models of OMAERUV. According to Kirchstetter et al. (2004), the AÅE of urban pollution is near unit and biomass burning aerosols ranges is approximately 2 between 300 nm to 1 $\mu$m. Bergstrom et al. (2007) also confirmed this conclusion from several field programs (SAFARI 2000, ACE Asia, PRIDE, TARFOX, INTEX-A). From the sensitivity study of Jethva and Torres (2011), z stronger spectral dependence of $n_i$ between 354 and 388 nm would allow simulations to reach a higher AAI while keeping $n_i$ at a relatively low level. In our study, this means to retrieve a higher $\omega_0$ at 550 nm.' (Line 329-340)

**17) Line 374 : "The retrieved SSA is out of typical uncertainty...." this statement does not make sense and should be removed or re-phrase appropriately. At least for the case presented here the authors did not come up with a quantitative estimate of the uncertainty in SSA and clearly it is difficult if not impossible because of the multiple error sources.**
Thanks for the correction, we have removed this sentence.

[revised manuscript text omitted]

English (UK)

| Page 7: [2] Formatted | Sun ji | 6/11/18 1:55:00 PM |
|---|---|---|

Font: 12 pt, English (US)

| Page 7: [3] Formatted | Sun ji | 6/11/18 1:55:00 PM |
|---|---|---|

Font: 10 pt, English (US)

| Page 7: [4] Formatted | Sun ji | 5/15/18 10:32:00 PM |
|---|---|---|

Left

| Page 7: [5] Formatted | Sun ji | 5/15/18 10:32:00 PM |
|---|---|---|

Left

| Page 7: [6] Formatted | Sun ji | 6/11/18 1:55:00 PM |
|---|---|---|

English (US), Not Highlight

| Page 7: [6] Formatted | Sun ji | 6/11/18 1:55:00 PM |
|---|---|---|

English (US), Not Highlight

| Page 7: [6] Formatted | Sun ji | 6/11/18 1:55:00 PM |
|---|---|---|

English (US), Not Highlight

| Page 7: [6] Formatted | Sun ji | 6/11/18 1:55:00 PM |
|---|---|---|

English (US), Not Highlight

| Page 7: [6] Formatted | Sun ji | 6/11/18 1:55:00 PM |
|---|---|---|

English (US), Not Highlight

| Page 7: [7] Deleted | Sun ji | 5/22/18 4:38:00 PM |
|---|---|---|

 The absorbing aerosol index (AAI) based on the near Ultra-Violet (near-UV) remote sensing techniques is a qualitative parameter that allows to retrieve aerosol optical properties with confidence

| Page 7: [8] Formatted | Sun ji | 6/11/18 1:55:00 PM |
|---|---|---|

Font: 10 pt, English (US)

| Page 7: [8] Formatted | Sun ji | 6/11/18 1:55:00 PM |
|---|---|---|

Font: 10 pt, English (US)

| Page 7: [9] Deleted | Sun ji | 8/10/18 2:14:00 PM |
|---|---|---|

a series of

| Page 7: [9] Deleted | Sun ji | 8/10/18 2:14:00 PM |
|---|---|---|

a series of

| Page 7: [9] Deleted | Sun ji | 8/10/18 2:14:00 PM |
|---|---|---|

a series of

| Page 7: [10] Formatted | Sun ji | 6/11/18 1:55:00 PM |
|---|---|---|

Font: 10 pt, English (US)

| Page 7: [11] Formatted | Sun ji | 6/11/18 1:55:00 PM |
|---|---|---|

Font: 10 pt, English (US)

| Page 7: [12] Formatted | Sun ji | 6/11/18 1:55:00 PM |
|---|---|---|

Font: 10 pt, English (US)

| Page 7: [13] Formatted | Sun ji | 6/11/18 1:55:00 PM |
|---|---|---|

Font: 10 pt, English (US)

| Page 7: [14] Deleted | Sun ji | 5/8/18 11:59:00 AM |
|---|---|---|

and

| Page 7: [14] Deleted | Sun ji | 5/8/18 11:59:00 AM |
|---|---|---|

and

| Page 7: [14] Deleted | Sun ji | 5/8/18 11:59:00 AM |
|---|---|---|

and

| Page 7: [15] Formatted | Sun ji | 6/11/18 1:55:00 PM |
|---|---|---|

Font: 10 pt, English (US)

| Page 7: [16] Deleted | Sun ji | 8/10/18 1:31:00 PM |
|---|---|---|

The

| Page 7: [16] Deleted | Sun ji | 8/10/18 1:31:00 PM |
|---|---|---|

The

| Page 7: [17] Formatted | Sun ji | 6/11/18 1:55:00 PM |
|---|---|---|

Font: 10 pt, English (US)

| Page 7: [17] Formatted | Sun ji | 6/11/18 1:55:00 PM |
|---|---|---|

Font: 10 pt, English (US)

| Page 7: [17] Formatted | Sun ji | 6/11/18 1:55:00 PM |
|---|---|---|

Font: 10 pt, English (US)

| Page 7: [17] Formatted | Sun ji | 6/11/18 1:55:00 PM |
|---|---|---|

Font: 10 pt, English (US)

| Page 7: [18] Formatted | Sun ji | 6/11/18 1:55:00 PM |
|---|---|---|

English (US), Not Highlight

| Page 7: [18] Formatted | Sun ji | 6/11/18 1:55:00 PM |

English (US), Not Highlight

| Page 7: [19] Formatted | Sun ji | 6/11/18 1:55:00 PM |

Font: 10 pt, (Asian) Chinese (China), (Other) English (US)

| Page 7: [20] Formatted | Sun ji | 6/11/18 1:55:00 PM |

Font: 10 pt, English (US)

| Page 7: [21] Commented [Sj1] | Sun ji | 5/15/18 4:25:00 PM |

Because the data in this case is sensitive to outliers, we applied outlier detection to remove outlier first.

| Page 7: [22] Formatted | Sun ji | 6/11/18 1:55:00 PM |

Font: 5.5 pt

| Page 7: [22] Formatted | Sun ji | 6/11/18 1:55:00 PM |

Font: 5.5 pt

| Page 7: [22] Formatted | Sun ji | 6/11/18 1:55:00 PM |

Font: 5.5 pt

| Page 7: [22] Formatted | Sun ji | 6/11/18 1:55:00 PM |

Font: 5.5 pt

| Page 7: [23] Formatted | Sun ji | 6/11/18 1:55:00 PM |

Font: 10 pt, English (US)

| Page 7: [24] Formatted | Sun ji | 6/11/18 1:55:00 PM |

Font: 5.5 pt

| Page 7: [25] Formatted | Sun ji | 6/11/18 1:55:00 PM |

Font: 10 pt, English (US)

| Page 7: [25] Formatted | Sun ji | 6/11/18 1:55:00 PM |

Font: 10 pt, English (US)

| Page 7: [25] Formatted | Sun ji | 6/11/18 1:55:00 PM |

Font: 10 pt, English (US)

| Page 7: [25] Formatted | Sun ji | 6/11/18 1:55:00 PM |

Font: 10 pt, English (US)

| Page 7: [26] Formatted | Sun ji | 6/11/18 1:55:00 PM |
|---|---|---|

Font: 10 pt, (Asian) Chinese (China), (Other) English (US)

| Page 7: [27] Formatted | Sun ji | 6/11/18 1:55:00 PM |
|---|---|---|

English (US)

| Page 7: [27] Formatted | Sun ji | 6/11/18 1:55:00 PM |
|---|---|---|

English (US)

| Page 7: [28] Formatted | Sun ji | 6/11/18 1:55:00 PM |
|---|---|---|

English (US)

| Page 7: [29] Formatted | Sun ji | 6/11/18 1:55:00 PM |
|---|---|---|

Font: 10 pt, English (US)

| Page 7: [29] Formatted | Sun ji | 6/11/18 1:55:00 PM |
|---|---|---|

Font: 10 pt, English (US)

| Page 7: [29] Formatted | Sun ji | 6/11/18 1:55:00 PM |
|---|---|---|

Font: 10 pt, English (US)

| Page 7: [30] Deleted | JP Veefkind | 5/15/18 8:23:00 AM |
|---|---|---|

 slightly smaller than the value of 0.90 measured independently by the AERONET instrument.

| Page 7: [31] Deleted | Sun ji | 5/8/18 12:03:00 PM |
|---|---|---|

relative distance

| Page 7: [31] Deleted | Sun ji | 5/8/18 12:03:00 PM |
|---|---|---|

relative distance

| Page 7: [31] Deleted | Sun ji | 5/8/18 12:03:00 PM |
|---|---|---|

relative distance

| Page 7: [32] Formatted | Sun ji | 6/11/18 1:55:00 PM |
|---|---|---|

Font: 10 pt, English (US)

| Page 7: [32] Formatted | Sun ji | 6/11/18 1:55:00 PM |
|---|---|---|

Font: 10 pt, English (US)

| Page 7: [33] Deleted | Sun ji | 5/8/18 12:07:00 PM |
|---|---|---|

Except for the observational errors, the impact of remaining error sources on $\omega_0$ retrieval is difficult to quantify.

| Page 7: [34] Deleted | Sun ji | 4/23/18 2:30:00 PM |
|---|---|---|

They consist of fine particles (aerodynamic diameter smaller than 2.5 μm) that have adverse impacts on the

environment and human health (Bäumer et al., 2008; Adler et al., 2011). Biomass burning aerosols

| Page 7: [35] Formatted | Sun ji | 6/11/18 1:55:00 PM |

Font: 10 pt, English (US)

| Page 7: [35] Formatted | Sun ji | 6/11/18 1:55:00 PM |

Font: 10 pt, English (US)

| Page 7: [35] Formatted | Sun ji | 6/11/18 1:55:00 PM |

Font: 10 pt, English (US)

| Page 7: [35] Formatted | Sun ji | 6/11/18 1:55:00 PM |

Font: 10 pt, English (US)

| Page 7: [36] Deleted | Sun ji | 4/23/18 2:28:00 PM |

also

| Page 7: [36] Deleted | Sun ji | 4/23/18 2:28:00 PM |

also

| Page 7: [37] Formatted | Sun ji | 6/11/18 1:55:00 PM |

Font: 10 pt, English (US)

| Page 7: [37] Formatted | Sun ji | 6/11/18 1:55:00 PM |

Font: 10 pt, English (US)

| Page 7: [38] Deleted | Sun ji | 8/10/18 1:44:00 PM |

| Page 7: [38] Deleted | Sun ji | 8/10/18 1:44:00 PM |

| Page 7: [39] Formatted | Sun ji | 6/11/18 1:55:00 PM |

Font: 10 pt, English (US)

| Page 7: [39] Formatted | Sun ji | 6/11/18 1:55:00 PM |

Font: 10 pt, English (US)

| Page 7: [39] Formatted | Sun ji | 6/11/18 1:55:00 PM |

Font: 10 pt, English (US)

| Page 7: [40] Deleted | Sun ji | 4/23/18 2:09:00 PM |

), one type of absorbing aerosol, black carbon (BC), can be considered as the second important warming agent after carbon dioxide. Absorbing aerosols heat the atmosphere primarily by interaction with solar radiation. They directly absorb the incoming or reflected sunlight. They are also able to reduce the reflectivity of the planet by depositing on bright surfaces

| Page 7: [41] Formatted | Sun ji | 6/11/18 1:55:00 PM |

Font: 10 pt, English (US)

| Page 7: [42] Deleted | Sun ji | 4/23/18 2:09:00 PM |

(Huang et al., 2013

| Page 7: [42] Deleted | Sun ji | 4/23/18 2:09:00 PM |

(Huang et al., 2013

| Page 7: [43] Formatted | Sun ji | 6/11/18 1:55:00 PM |

Font: 10 pt, English (US)

| Page 7: [43] Formatted | Sun ji | 6/11/18 1:55:00 PM |

Font: 10 pt, English (US)

| Page 7: [43] Formatted | Sun ji | 6/11/18 1:55:00 PM |

Font: 10 pt, English (US)

| Page 7: [43] Formatted | Sun ji | 6/11/18 1:55:00 PM |

Font: 10 pt, English (US)

| Page 7: [43] Formatted | Sun ji | 6/11/18 1:55:00 PM |

Font: 10 pt, English (US)

| Page 7: [43] Formatted | Sun ji | 6/11/18 1:55:00 PM |

Font: 10 pt, English (US)

| Page 7: [43] Formatted | Sun ji | 6/11/18 1:55:00 PM |

Font: 10 pt, English (US)

| Page 7: [44] Deleted | Sun ji | 4/23/18 2:12:00 PM |

Quantifying the climate effect of absorbing aerosols is therefore important.

| Page 7: [44] Deleted | Sun ji | 4/23/18 2:12:00 PM |

Quantifying the climate effect of absorbing aerosols is therefore important.

| Page 7: [45] Formatted | Sun ji | 6/11/18 1:55:00 PM |

Font: 10 pt, English (US)

| Page 7: [45] Formatted | Sun ji | 6/11/18 1:55:00 PM |

Font: 10 pt, English (US)

| Page 7: [45] Formatted | Sun ji | 6/11/18 1:55:00 PM |

Font: 10 pt, English (US)

| Page 7: [45] Formatted | Sun ji | 6/11/18 1:55:00 PM |

Font: 10 pt, English (US)

| Page 7: [46] Deleted | JP Veefkind | 5/14/18 11:53:00 AM |

.

| Page 7: [46] Deleted | JP Veefkind | 5/14/18 11:53:00 AM |

.

| Page 7: [47] Deleted | Sun ji | 5/22/18 5:41:00 PM |

,

| Page 7: [47] Deleted | Sun ji | 5/22/18 5:41:00 PM |

,

| Page 7: [48] Formatted | Sun ji | 6/11/18 1:55:00 PM |

Font: 10 pt, English (US)

| Page 7: [48] Formatted | Sun ji | 6/11/18 1:55:00 PM |

Font: 10 pt, English (US)

| Page 7: [48] Formatted | Sun ji | 6/11/18 1:55:00 PM |

Font: 10 pt, English (US)

| Page 7: [48] Formatted | Sun ji | 6/11/18 1:55:00 PM |

Font: 10 pt, English (US)

| Page 7: [48] Formatted | Sun ji | 6/11/18 1:55:00 PM |

Font: 10 pt, English (US)

| Page 7: [49] Formatted | Sun ji | 6/11/18 1:55:00 PM |

English (US), Not Highlight

| Page 7: [49] Formatted | Sun ji | 6/11/18 1:55:00 PM |

English (US), Not Highlight

| Page 7: [49] Formatted | Sun ji | 6/11/18 1:55:00 PM |

English (US), Not Highlight

| Page 7: [49] Formatted | Sun ji | 6/11/18 1:55:00 PM |

English (US), Not Highlight

| Page 7: [49] Formatted | Sun ji | 6/11/18 1:55:00 PM |

English (US), Not Highlight

| Page 7: [49] Formatted | Sun ji | 6/11/18 1:55:00 PM |

English (US), Not Highlight

| Page 8: [50] Formatted | Sun ji | 6/11/18 1:55:00 PM |

English (US)

| Page 8: [50] Formatted | Sun ji | 6/11/18 1:55:00 PM |

English (US)

| Page 8: [50] Formatted | Sun ji | 6/11/18 1:55:00 PM |

English (US)

| Page 8: [50] Formatted | Sun ji | 6/11/18 1:55:00 PM |

English (US)

| Page 8: [51] Deleted | Sun ji | 5/22/18 11:30:00 AM |

$\omega_0$ is defined as the ratio of the radiation scattered by aerosol particles to the total attenuation. Because aerosol compositions and properties are highly variable in space and time, measuring the global distribution of $\omega_0$ relies on remote sensing techniques. The POLarization and Directionality of the Earth's Reflectances (POLDER) $\omega_0$ from a combination of multi-angular, multi-spectral observations of the measures aerosol polarized phase function. This provides information directly related to $\omega_0$ (Leroy et al., 1997). But However, there is no continuous temporal coverage because the first two instruments encountered technical hitches that prematurely ended the missions ended prematurely due to technical problems on the satellite level. The third POLDER instrument mission covered the period 2004-2014.

| Page 8: [52] Formatted | Sun ji | 6/11/18 1:55:00 PM |

Font: 10 pt, (Asian) Chinese (China), (Other) English (US)

| Page 8: [53] Deleted | Sun ji | 5/22/18 12:33:00 PM |

As a result, $\omega_0$ is usually retrieved by forward simulations that are adapted to observational parameters. Many implementations have been done for ground-based network measurements (Dubovik et al., 1998; Eck et al., 2003; Petters et al., 2003; Kassianov et al., 2005; Corr et al., 2009; Yin et al., 2015), while relatively fewer applications to satellite instruments exist due to lack of validation (Lee et al., 2007; Ialongo et al., 2010; Eck et al., 2013). Moreover, a majority of those methods heavily depend on the aerosol optical thickness ($\tau$), either in forward model simulations or in validation procedures. This makes the derived $\omega_0$ subject to large uncertainties. The reason is that $\tau$ retrieval requires assumptions on aerosol types, and the commonly used $\tau$ that is retrieved in the visible band where the signal of bright surfaces is strong. Besides, the aerosol effect on radiance is inversely proportional to wavelength (Kaufman, 1993), and the sensitivity to $\omega_0$ is not significant for most $\tau$ measurements in the visible and infrared band (Kaufman et al., 1997).

The

| Page 8: [54] Formatted | Sun ji | 6/11/18 1:55:00 PM |
|---|---|---|

Font: 10 pt, (Asian) Chinese (China), (Other) English (US)

| Page 8: [54] Formatted | Sun ji | 6/11/18 1:55:00 PM |
|---|---|---|

Font: 10 pt, (Asian) Chinese (China), (Other) English (US)

| Page 8: [54] Formatted | Sun ji | 6/11/18 1:55:00 PM |
|---|---|---|

Font: 10 pt, (Asian) Chinese (China), (Other) English (US)

| Page 8: [54] Formatted | Sun ji | 6/11/18 1:55:00 PM |
|---|---|---|

Font: 10 pt, (Asian) Chinese (China), (Other) English (US)

| Page 8: [54] Formatted | Sun ji | 6/11/18 1:55:00 PM |
|---|---|---|

Font: 10 pt, (Asian) Chinese (China), (Other) English (US)

| Page 8: [54] Formatted | Sun ji | 6/11/18 1:55:00 PM |
|---|---|---|

Font: 10 pt, (Asian) Chinese (China), (Other) English (US)

| Page 8: [54] Formatted | Sun ji | 6/11/18 1:55:00 PM |
|---|---|---|

Font: 10 pt, (Asian) Chinese (China), (Other) English (US)

| Page 8: [54] Formatted | Sun ji | 6/11/18 1:55:00 PM |
|---|---|---|

Font: 10 pt, (Asian) Chinese (China), (Other) English (US)

| Page 8: [54] Formatted | Sun ji | 6/11/18 1:55:00 PM |
|---|---|---|

Font: 10 pt, (Asian) Chinese (China), (Other) English (US)

| Page 8: [54] Formatted | Sun ji | 6/11/18 1:55:00 PM |
|---|---|---|

Font: 10 pt, (Asian) Chinese (China), (Other) English (US)

| Page 8: [54] Formatted | Sun ji | 6/11/18 1:55:00 PM |
|---|---|---|

Font: 10 pt, (Asian) Chinese (China), (Other) English (US)

| Page 8: [54] Formatted | Sun ji | 6/11/18 1:55:00 PM |
|---|---|---|

Font: 10 pt, (Asian) Chinese (China), (Other) English (US)

| Page 8: [55] Formatted | Sun ji | 6/11/18 1:55:00 PM |
|---|---|---|

Font: 10 pt, English (US)

| Page 8: [55] Formatted | Sun ji | 6/11/18 1:55:00 PM |
|---|---|---|

Font: 10 pt, English (US)

| Page 8: [56] Formatted | Sun ji | 6/11/18 1:55:00 PM |
|---|---|---|

Font: 10 pt, English (US)

| Page 8: [56] Formatted | Sun ji | 6/11/18 1:55:00 PM |

Font: 10 pt, English (US)

| Page 8: [56] Formatted | Sun ji | 6/11/18 1:55:00 PM |

Font: 10 pt, English (US)

| Page 8: [57] Formatted | Sun ji | 6/11/18 1:55:00 PM |

English (US), Not Highlight

| Page 8: [57] Formatted | Sun ji | 6/11/18 1:55:00 PM |

English (US), Not Highlight

| Page 8: [58] Formatted | Sun ji | 6/11/18 1:55:00 PM |

English (US), Not Highlight

| Page 8: [58] Formatted | Sun ji | 6/11/18 1:55:00 PM |

English (US), Not Highlight

| Page 8: [59] Deleted | Sun ji | 5/22/18 3:58:00 PM |

 forward model simulations with the absorbing aerosol index (AAI) (Herman et al., 1997)

| Page 8: [60] Formatted | Sun ji | 6/11/18 1:55:00 PM |

Font: 10 pt, English (US)

| Page 8: [60] Formatted | Sun ji | 6/11/18 1:55:00 PM |

Font: 10 pt, English (US)

| Page 8: [60] Formatted | Sun ji | 6/11/18 1:55:00 PM |

Font: 10 pt, English (US)

| Page 8: [60] Formatted | Sun ji | 6/11/18 1:55:00 PM |

Font: 10 pt, English (US)

| Page 8: [60] Formatted | Sun ji | 6/11/18 1:55:00 PM |

Font: 10 pt, English (US)

| Page 8: [61] Formatted | Sun ji | 6/11/18 1:55:00 PM |

English (US)

| Page 8: [61] Formatted | Sun ji | 6/11/18 1:55:00 PM |

English (US)

| Page 8: [62] Deleted | Sun ji | 5/8/18 12:15:00 PM |

improving

| Page 8: [62] Deleted | Sun ji | 5/8/18 12:15:00 PM |

improving

| Page 8: [63] Formatted | Sun ji | 6/11/18 1:55:00 PM |

Font: 10 pt, English (US)

| Page 8: [64] Formatted | Sun ji | 6/11/18 1:55:00 PM |

Font: 10 pt, English (US)

| Page 8: [64] Formatted | Sun ji | 6/11/18 1:55:00 PM |

Font: 10 pt, English (US)

| Page 8: [65] Deleted | Sun ji | 5/8/18 12:16:00 PM |

foremost

| Page 8: [65] Deleted | Sun ji | 5/8/18 12:16:00 PM |

foremost

| Page 8: [65] Deleted | Sun ji | 5/8/18 12:16:00 PM |

foremost

| Page 8: [65] Deleted | Sun ji | 5/8/18 12:16:00 PM |

foremost

| Page 8: [66] Formatted | Sun ji | 6/11/18 1:55:00 PM |

Font: 10 pt, English (US)

| Page 8: [67] Deleted | Sun ji | 8/10/18 2:02:00 PM |

assumptions on aerosol types, which significantly reduce the retrieval uncertainty

| Page 8: [68] Formatted | Sun ji | 6/11/18 1:55:00 PM |

Font: 10 pt, English (US)

| Page 8: [68] Formatted | Sun ji | 6/11/18 1:55:00 PM |

Font: 10 pt, English (US)

| Page 8: [68] Formatted | Sun ji | 6/11/18 1:55:00 PM |

Font: 10 pt, English (US)

| Page 8: [68] Formatted | Sun ji | 6/11/18 1:55:00 PM |

Font: 10 pt, English (US)

| Page 8: [68] Formatted | Sun ji | 6/11/18 1:55:00 PM |

Font: 10 pt, English (US)

| Page 8: [68] Formatted | Sun ji | 6/11/18 1:55:00 PM |

Font: 10 pt, English (US)

| Page 8: [68] Formatted | Sun ji | 6/11/18 1:55:00 PM |
|---|---|---|

Font: 10 pt, English (US)

| Page 8: [69] Deleted | Sun ji | 4/23/18 6:58:00 PM |
|---|---|---|

.

| Page 8: [69] Deleted | Sun ji | 4/23/18 6:58:00 PM |
|---|---|---|

.

| Page 8: [70] Formatted | Sun ji | 6/11/18 1:55:00 PM |
|---|---|---|

Font: 10 pt, English (US)

| Page 8: [70] Formatted | Sun ji | 6/11/18 1:55:00 PM |
|---|---|---|

Font: 10 pt, English (US)

| Page 8: [71] Formatted | Sun ji | 6/11/18 1:55:00 PM |
|---|---|---|

English (US)

| Page 8: [71] Formatted | Sun ji | 6/11/18 1:55:00 PM |
|---|---|---|

English (US)

| Page 8: [71] Formatted | Sun ji | 6/11/18 1:55:00 PM |
|---|---|---|

English (US)

| Page 8: [71] Formatted | Sun ji | 6/11/18 1:55:00 PM |
|---|---|---|

English (US)

| Page 8: [72] Deleted | Sun ji | 4/23/18 7:02:00 PM |
|---|---|---|

, the sensitivity of $\tau$ in the visible band to $\omega_0$ is lower over dark surfaces (Kaufman et al., 1997), while

| Page 8: [73] Formatted | Sun ji | 6/11/18 1:55:00 PM |
|---|---|---|

Font: 10 pt, English (US)

| Page 8: [73] Formatted | Sun ji | 6/11/18 1:55:00 PM |
|---|---|---|

Font: 10 pt, English (US)

| Page 8: [74] Formatted | Sun ji | 6/11/18 1:55:00 PM |
|---|---|---|

Font: 10 pt, English (US)

| Page 8: [75] Formatted | Sun ji | 6/11/18 1:55:00 PM |
|---|---|---|

Font: 10 pt, English (US)

| Page 8: [76] Formatted | Sun ji | 6/11/18 1:55:00 PM |
|---|---|---|

Font: 10 pt, English (US)

| Page 8: [76] Formatted | Sun ji | 6/11/18 1:55:00 PM |
|---|---|---|

Font: 10 pt, English (US)

| Page 8: [77] Deleted | Sun ji | 4/23/18 3:58:00 PM |
|---|---|---|

Empirical models were also developed to build connections between the AAI and parameters it depends on. Hsu et al. (1999) found a linear relation between the TOMS retrieved AAI and Sun-photometer measured $\tau$ over regions with biomass burning and regions covered by African dust. Ginoux and Torres (2003) implemented an empirical relation between the AAI retrieved from TOMS with $\tau$, $\omega_0$ and surface pressure ($P_s$) to characterize the dust aerosols. Although requiring less computational cost, applying these empirical models is either limited by specific conditions or subject to large errors. Thus, these methods have not been widely used.

| Page 8: [77] Deleted | Sun ji | 4/23/18 3:58:00 PM |
|---|---|---|

Empirical models were also developed to build connections between the AAI and parameters it depends on. Hsu et al. (1999) found a linear relation between the TOMS retrieved AAI and Sun-photometer measured $\tau$ over regions with biomass burning and regions covered by African dust. Ginoux and Torres (2003) implemented an empirical relation between the AAI retrieved from TOMS with $\tau$, $\omega_0$ and surface pressure ($P_s$) to characterize the dust aerosols. Although requiring less computational cost, applying these empirical models is either limited by specific conditions or subject to large errors. Thus, these methods have not been widely used.

| Page 8: [78] Formatted | Sun ji | 6/11/18 1:55:00 PM |
|---|---|---|

English (US)

| Page 8: [78] Formatted | Sun ji | 6/11/18 1:55:00 PM |
|---|---|---|

English (US)

| Page 8: [79] Deleted | Sun ji | 5/8/18 12:19:00 PM |
|---|---|---|

, that

| Page 8: [79] Deleted | Sun ji | 5/8/18 12:19:00 PM |
|---|---|---|

, that

| Page 8: [79] Deleted | Sun ji | 5/8/18 12:19:00 PM |
|---|---|---|

, that

| Page 8: [80] Formatted | Sun ji | 6/11/18 1:55:00 PM |
|---|---|---|

Font: 10 pt, English (US)

| Page 8: [80] Formatted | Sun ji | 6/11/18 1:55:00 PM |
|---|---|---|

Font: 10 pt, English (US)

| Page 8: [80] Formatted | Sun ji | 6/11/18 1:55:00 PM |
|---|---|---|

Font: 10 pt, English (US)

| Page 8: [81] Formatted | Sun ji | 6/11/18 1:55:00 PM |
|---|---|---|

Font: 10 pt, English (US)

| Page 8: [82] Deleted | Sun ji | 8/2/18 10:20:00 AM |
|---|---|---|

, this series of fires occurring in central Chile (Pichilemu 34.39°S 72.00°W and Consititución 35.33°S, 72.42°W) was

| Page 8: [83] Formatted | Sun ji | 6/11/18 1:55:00 PM |
|---|---|---|

Font: 10 pt, English (US)

| Page 8: [83] Formatted | Sun ji | 6/11/18 1:55:00 PM |
|---|---|---|

Font: 10 pt, English (US)

| Page 8: [83] Formatted | Sun ji | 6/11/18 1:55:00 PM |
|---|---|---|

Font: 10 pt, English (US)

| Page 8: [84] Formatted | Sun ji | 6/11/18 1:55:00 PM |
|---|---|---|

Font: 10 pt, English (US)

| Page 8: [85] Deleted | Sun ji | 6/17/18 10:17:00 AM |
|---|---|---|

the MODerate-resolution Imaging Spectroradiometer (

| Page 8: [85] Deleted | Sun ji | 6/17/18 10:17:00 AM |
|---|---|---|

the MODerate-resolution Imaging Spectroradiometer (

| Page 8: [85] Deleted | Sun ji | 6/17/18 10:17:00 AM |
|---|---|---|

the MODerate-resolution Imaging Spectroradiometer (

| Page 8: [86] Formatted | Sun ji | 6/11/18 1:55:00 PM |
|---|---|---|

Font: 10 pt, English (US)

| Page 9: [87] Formatted | Sun ji | 5/15/18 10:32:00 PM |
|---|---|---|

Heading 2

| Page 9: [88] Formatted | Sun ji | 5/15/18 10:32:00 PM |
|---|---|---|

Left

| Page 9: [89] Formatted | Sun ji | 6/11/18 1:55:00 PM |
|---|---|---|

Font: 10 pt, English (US)

| Page 9: [89] Formatted | Sun ji | 6/11/18 1:55:00 PM |
|---|---|---|

Font: 10 pt, English (US)

| Page 9: [90] Deleted | Sun ji | 4/23/18 11:42:00 AM |
|---|---|---|

for

| Page 9: [91] Formatted | Sun ji | 6/11/18 1:55:00 PM |
|---|---|---|

Font: 10 pt, English (US)

| Page 9: [92] Formatted | Sun ji | 6/11/18 1:55:00 PM |
|---|---|---|

Font: 10 pt, English (US)

| Page 9: [93] Formatted | Sun ji | 6/11/18 1:55:00 PM |
|---|---|---|

Font: 10 pt, English (US)

Font: 10 pt, English (US)

| Page 9: [93] Formatted | Sun ji | 6/11/18 1:55:00 PM |
|---|---|---|

Font: 10 pt, English (US)

| Page 9: [93] Formatted | Sun ji | 6/11/18 1:55:00 PM |
|---|---|---|

Font: 10 pt, English (US)

| Page 9: [93] Formatted | Sun ji | 6/11/18 1:55:00 PM |
|---|---|---|

Font: 10 pt, English (US)

| Page 9: [93] Formatted | Sun ji | 6/11/18 1:55:00 PM |
|---|---|---|

Font: 10 pt, English (US)

| Page 9: [93] Formatted | Sun ji | 6/11/18 1:55:00 PM |
|---|---|---|

Font: 10 pt, English (US)

| Page 9: [93] Formatted | Sun ji | 6/11/18 1:55:00 PM |
|---|---|---|

Font: 10 pt, English (US)

| Page 9: [93] Formatted | Sun ji | 6/11/18 1:55:00 PM |
|---|---|---|

Font: 10 pt, English (US)

| Page 9: [93] Formatted | Sun ji | 6/11/18 1:55:00 PM |
|---|---|---|

Font: 10 pt, English (US)

| Page 9: [93] Formatted | Sun ji | 6/11/18 1:55:00 PM |
|---|---|---|

Font: 10 pt, English (US)

| Page 9: [94] Formatted | Sun ji | 6/11/18 1:55:00 PM |
|---|---|---|

Font: 10 pt, English (US)

| Page 9: [95] Deleted | JP Veefkind | 5/14/18 12:25:00 PM |
|---|---|---|

the model

| Page 9: [95] Deleted | JP Veefkind | 5/14/18 12:25:00 PM |
|---|---|---|

the model

| Page 9: [96] Formatted | Sun ji | 6/11/18 1:55:00 PM |
|---|---|---|

Font: 10 pt, English (US)

| Page 9: [97] Formatted | Sun ji | 6/11/18 1:55:00 PM |
|---|---|---|

Font: 10 pt, English (US)

| Page 9: [97] Formatted | Sun ji | 6/11/18 1:55:00 PM |
|---|---|---|

Font: 10 pt, English (US)

| Page 9: [97] Formatted | Sun ji | 6/11/18 1:55:00 PM |
|---|---|---|

Font: 10 pt, English (US)

| Page 9: [97] Formatted | Sun ji | 6/11/18 1:55:00 PM |
|---|---|---|

Font: 10 pt, English (US)

| Page 9: [97] Formatted | Sun ji | 6/11/18 1:55:00 PM |
|---|---|---|

Font: 10 pt, English (US)

| Page 9: [97] Formatted | Sun ji | 6/11/18 1:55:00 PM |
|---|---|---|

Font: 10 pt, English (US)

| Page 9: [97] Formatted | Sun ji | 6/11/18 1:55:00 PM |
|---|---|---|

Font: 10 pt, English (US)

| Page 9: [97] Formatted | Sun ji | 6/11/18 1:55:00 PM |
|---|---|---|

Font: 10 pt, English (US)

| Page 9: [97] Formatted | Sun ji | 6/11/18 1:55:00 PM |
|---|---|---|

Font: 10 pt, English (US)

| Page 9: [97] Formatted | Sun ji | 6/11/18 1:55:00 PM |
|---|---|---|

Font: 10 pt, English (US)

| Page 9: [97] Formatted | Sun ji | 6/11/18 1:55:00 PM |
|---|---|---|

Font: 10 pt, English (US)

| Page 9: [98] Formatted | Sun ji | 6/11/18 1:55:00 PM |
|---|---|---|

Font: 10 pt, English (US)

| Page 9: [98] Formatted | Sun ji | 6/11/18 1:55:00 PM |
|---|---|---|

Font: 10 pt, English (US)

| Page 9: [98] Formatted | Sun ji | 6/11/18 1:55:00 PM |
|---|---|---|

Font: 10 pt, English (US)

| Page 9: [98] Formatted | Sun ji | 6/11/18 1:55:00 PM |
|---|---|---|

Font: 10 pt, English (US)

| Page 9: [99] Deleted | JP Veefkind | 5/14/18 12:15:00 PM |
|---|---|---|

The aerosol effect is assumed at $\lambda_2$ is negligible, so that the assumption of a Rayleigh atmosphere at this wavelength is valid. This The way the longer wavelength $\lambda_2$ is treated as reference wavelength where the surface albedo ($a_s$) is determined by fitting the observed radiance, i.e. $I_{\lambda2}^{Ray}(a_s) = I_{\lambda2}^{obs}$. This $a_s$ is also assumed used at $\lambda_1$ to compute $I_{\lambda1}^{Ray}$.

| Page 9: [100] Formatted | Sun ji | 6/11/18 1:55:00 PM |
|---|---|---|

Font: 10 pt, English (US)

| Page 9: [100] Formatted | Sun ji | 6/11/18 1:55:00 PM |
|---|---|---|

Font: 10 pt, English (US)

| Page 9: [100] Formatted | Sun ji | 6/11/18 1:55:00 PM |
|---|---|---|

Font: 10 pt, English (US)

| Page 9: [100] Formatted | Sun ji | 6/11/18 1:55:00 PM |
|---|---|---|

Font: 10 pt, English (US)

| Page 9: [100] Formatted | Sun ji | 6/11/18 1:55:00 PM |
|---|---|---|

Font: 10 pt, English (US)

| Page 9: [100] Formatted | Sun ji | 6/11/18 1:55:00 PM |
|---|---|---|

Font: 10 pt, English (US)

| Page 9: [100] Formatted | Sun ji | 6/11/18 1:55:00 PM |
|---|---|---|

Font: 10 pt, English (US)

| Page 9: [100] Formatted | Sun ji | 6/11/18 1:55:00 PM |
|---|---|---|

Font: 10 pt, English (US)

| Page 9: [100] Formatted | Sun ji | 6/11/18 1:55:00 PM |
|---|---|---|

Font: 10 pt, English (US)

| Page 9: [100] Formatted | Sun ji | 6/11/18 1:55:00 PM |
|---|---|---|

Font: 10 pt, English (US)

| Page 9: [100] Formatted | Sun ji | 6/11/18 1:55:00 PM |
|---|---|---|

Font: 10 pt, English (US)

| Page 9: [101] Formatted | Sun ji | 6/11/18 1:55:00 PM |
|---|---|---|

Font: 10 pt, English (US)

| Page 9: [101] Formatted | Sun ji | 6/11/18 1:55:00 PM |
|---|---|---|

Font: 10 pt, English (US)

| Page 9: [101] Formatted | Sun ji | 6/11/18 1:55:00 PM |
|---|---|---|

Font: 10 pt, English (US)

| Page 9: [101] Formatted | Sun ji | 6/11/18 1:55:00 PM |

Font: 10 pt, English (US)

| Page 9: [102] Formatted | Sun ji | 6/11/18 1:55:00 PM |

Font: 10 pt, English (US)

| Page 9: [102] Formatted | Sun ji | 6/11/18 1:55:00 PM |

Font: 10 pt, English (US)

| Page 9: [102] Formatted | Sun ji | 6/11/18 1:55:00 PM |

Font: 10 pt, English (US)

| Page 9: [102] Formatted | Sun ji | 6/11/18 1:55:00 PM |

Font: 10 pt, English (US)

| Page 9: [102] Formatted | Sun ji | 6/11/18 1:55:00 PM |

Font: 10 pt, English (US)

| Page 9: [102] Formatted | Sun ji | 6/11/18 1:55:00 PM |

Font: 10 pt, English (US)

| Page 9: [102] Formatted | Sun ji | 6/11/18 1:55:00 PM |

Font: 10 pt, English (US)

| Page 9: [102] Formatted | Sun ji | 6/11/18 1:55:00 PM |

Font: 10 pt, English (US)

| Page 9: [102] Formatted | Sun ji | 6/11/18 1:55:00 PM |

Font: 10 pt, English (US)

| Page 9: [102] Formatted | Sun ji | 6/11/18 1:55:00 PM |

Font: 10 pt, English (US)

| Page 9: [103] Deleted | Sun ji | 5/8/18 11:15:00 AM |

Consequently,

| Page 9: [103] Deleted | Sun ji | 5/8/18 11:15:00 AM |

Consequently,

| Page 9: [103] Deleted | Sun ji | 5/8/18 11:15:00 AM |

Consequently,

| Page 9: [104] Deleted | Sun ji | 4/23/18 4:08:00 PM |

the difference between $I_{\lambda1}^{obs}$ and $I_{\lambda1}^{Ray}$ normalized by the measured radiance $I_{\lambda1}^{obs}$:

| Page 9: [105] Formatted | Sun ji | 6/11/18 1:55:00 PM |
|---|---|---|

Font: 10 pt, English (US)

| Page 9: [105] Formatted | Sun ji | 6/11/18 1:55:00 PM |
|---|---|---|

Font: 10 pt, English (US)

| Page 9: [105] Formatted | Sun ji | 6/11/18 1:55:00 PM |
|---|---|---|

Font: 10 pt, English (US)

| Page 9: [105] Formatted | Sun ji | 6/11/18 1:55:00 PM |
|---|---|---|

Font: 10 pt, English (US)

| Page 9: [106] Formatted | Sun ji | 6/11/18 1:55:00 PM |
|---|---|---|

Font: 10 pt, English (US)

| Page 9: [106] Formatted | Sun ji | 6/11/18 1:55:00 PM |
|---|---|---|

Font: 10 pt, English (US)

| Page 9: [106] Formatted | Sun ji | 6/11/18 1:55:00 PM |
|---|---|---|

Font: 10 pt, English (US)

| Page 9: [106] Formatted | Sun ji | 6/11/18 1:55:00 PM |
|---|---|---|

Font: 10 pt, English (US)

| Page 9: [107] Formatted | Sun ji | 6/11/18 1:55:00 PM |
|---|---|---|

Font: 10 pt, English (US)

| Page 9: [107] Formatted | Sun ji | 6/11/18 1:55:00 PM |
|---|---|---|

Font: 10 pt, English (US)

| Page 9: [107] Formatted | Sun ji | 6/11/18 1:55:00 PM |
|---|---|---|

Font: 10 pt, English (US)

| Page 9: [107] Formatted | Sun ji | 6/11/18 1:55:00 PM |
|---|---|---|

Font: 10 pt, English (US)

| Page 9: [108] Formatted | Sun ji | 6/11/18 1:55:00 PM |
|---|---|---|

Font: 10 pt, English (US)

| Page 9: [108] Formatted | Sun ji | 6/11/18 1:55:00 PM |
|---|---|---|

Font: 10 pt, English (US)

| Page 9: [108] Formatted | Sun ji | 6/11/18 1:55:00 PM |
|---|---|---|

Font: 10 pt, English (US)

| Page 9: [108] Formatted | Sun ji | 6/11/18 1:55:00 PM |
|---|---|---|

Font: 10 pt, English (US)

| Page 9: [108] Formatted | Sun ji | 6/11/18 1:55:00 PM |
|---|---|---|

Font: 10 pt, English (US)

| Page 9: [108] Formatted | Sun ji | 6/11/18 1:55:00 PM |
|---|---|---|

Font: 10 pt, English (US)

| Page 9: [108] Formatted | Sun ji | 6/11/18 1:55:00 PM |
|---|---|---|

Font: 10 pt, English (US)

| Page 9: [108] Formatted | Sun ji | 6/11/18 1:55:00 PM |
|---|---|---|

Font: 10 pt, English (US)

| Page 9: [108] Formatted | Sun ji | 6/11/18 1:55:00 PM |
|---|---|---|

Font: 10 pt, English (US)

| Page 9: [108] Formatted | Sun ji | 6/11/18 1:55:00 PM |
|---|---|---|

Font: 10 pt, English (US)

| Page 9: [109] Formatted | Sun ji | 6/11/18 1:55:00 PM |
|---|---|---|

Font: 10 pt, English (US), Not Highlight

| Page 9: [110] Formatted | Sun ji | 6/11/18 1:55:00 PM |
|---|---|---|

Font: 10 pt, English (US)

| Page 9: [110] Formatted | Sun ji | 6/11/18 1:55:00 PM |
|---|---|---|

Font: 10 pt, English (US)

| Page 9: [111] Formatted | Sun ji | 6/11/18 1:55:00 PM |
|---|---|---|

Font: 10 pt, English (US)

| Page 9: [111] Formatted | Sun ji | 6/11/18 1:55:00 PM |
|---|---|---|

Font: 10 pt, English (US)

| Page 9: [111] Formatted | Sun ji | 6/11/18 1:55:00 PM |
|---|---|---|

Font: 10 pt, English (US)

| Page 9: [111] Formatted | Sun ji | 6/11/18 1:55:00 PM |
|---|---|---|

Font: 10 pt, English (US)

| Page 9: [111] Formatted | Sun ji | 6/11/18 1:55:00 PM |
|---|---|---|

Font: 10 pt, English (US)

| Page 9: [111] Formatted | Sun ji | 6/11/18 1:55:00 PM |
|---|---|---|

Font: 10 pt, English (US)

| Page 9: [112] Formatted | Sun ji | 6/11/18 1:55:00 PM |
|---|---|---|

Font: 10 pt, English (US), Highlight

| Page 9: [112] Formatted | Sun ji | 6/11/18 1:55:00 PM |
|---|---|---|

Font: 10 pt, English (US), Highlight

| Page 9: [113] Formatted | Sun ji | 6/11/18 1:55:00 PM |
|---|---|---|

English (US)

| Page 9: [113] Formatted | Sun ji | 6/11/18 1:55:00 PM |
|---|---|---|

English (US)

| Page 9: [114] Formatted | Sun ji | 6/11/18 1:55:00 PM |
|---|---|---|

Font: 10 pt, English (US)

| Page 9: [115] Deleted | JP Veefkind | 5/14/18 12:41:00 PM |
|---|---|---|

It

| Page 9: [115] Deleted | JP Veefkind | 5/14/18 12:41:00 PM |
|---|---|---|

It

| Page 9: [115] Deleted | JP Veefkind | 5/14/18 12:41:00 PM |
|---|---|---|

It

| Page 9: [115] Deleted | JP Veefkind | 5/14/18 12:41:00 PM |
|---|---|---|

It

| Page 9: [115] Deleted | JP Veefkind | 5/14/18 12:41:00 PM |
|---|---|---|

It

| Page 9: [115] Deleted | JP Veefkind | 5/14/18 12:41:00 PM |
|---|---|---|

It

| Page 9: [115] Deleted | JP Veefkind | 5/14/18 12:41:00 PM |
|---|---|---|

It

| Page 9: [116] Deleted | JP Veefkind | 5/14/18 12:40:00 PM |
|---|---|---|

various

| Page 9: [116] Deleted | JP Veefkind | 5/14/18 12:40:00 PM |
|---|---|---|

various

| Page 9: [116] Deleted | JP Veefkind | 5/14/18 12:40:00 PM |

various

| Page 9: [116] Deleted | JP Veefkind | 5/14/18 12:40:00 PM |

various

| Page 9: [116] Deleted | JP Veefkind | 5/14/18 12:40:00 PM |

various

| Page 9: [116] Deleted | JP Veefkind | 5/14/18 12:40:00 PM |

various

| Page 9: [117] Formatted | Sun ji | 6/11/18 1:55:00 PM |

Font: 10 pt, English (US)

| Page 9: [117] Formatted | Sun ji | 6/11/18 1:55:00 PM |

Font: 10 pt, English (US)

| Page 9: [118] Deleted | JP Veefkind | 5/14/18 12:41:00 PM |

DISAMAR

| Page 9: [118] Deleted | JP Veefkind | 5/14/18 12:41:00 PM |

DISAMAR

| Page 9: [118] Deleted | JP Veefkind | 5/14/18 12:41:00 PM |

DISAMAR

| Page 9: [119] Formatted | Sun ji | 6/11/18 1:55:00 PM |

Font: 10 pt, English (US)

| Page 9: [119] Formatted | Sun ji | 6/11/18 1:55:00 PM |

Font: 10 pt, English (US)

| Page 9: [119] Formatted | Sun ji | 6/11/18 1:55:00 PM |

Font: 10 pt, English (US)

| Page 9: [120] Formatted | Sun ji | 5/15/18 10:32:00 PM |

Left, Don't add space between paragraphs of the same style

| Page 9: [121] Deleted | Sun ji | 4/23/18 11:54:00 AM |

Given size distribution function ($r_g$), complex refractive index ($n_r$ and $n_i$) at specific wavelengths and a certain wavelength interpolation method, DISAMAR calculates the spectrally dependent optical properties (e.g. $\omega_0$ and phase function $P(\Theta)$) within the specified wavelength range. In this study, we use the linear interpolation and the spectrum coverage from 340 to 675 nm.

| Page 9: [122] Deleted | JP Veefkind | 5/14/18 12:43:00 PM |

that

| Page 9: [122] Deleted | JP Veefkind | 5/14/18 12:43:00 PM |

that

| Page 9: [123] Deleted | Sun ji | 8/2/18 10:21:00 AM |

ly

| Page 9: [124] Formatted | Sun ji | 6/11/18 1:55:00 PM |

Font: 10 pt, English (US)

| Page 9: [125] Formatted | Sun ji | 6/11/18 1:55:00 PM |

Font: 5.5 pt

| Page 9: [126] Formatted | Sun ji | 6/11/18 1:55:00 PM |

English (US)

| Page 9: [127] Formatted | Sun ji | 6/11/18 1:55:00 PM |

Font: 10 pt, English (US)

| Page 9: [128] Formatted | Sun ji | 6/11/18 1:55:00 PM |

Font: 10 pt, English (US)

| Page 9: [129] Formatted | Sun ji | 6/11/18 1:55:00 PM |

Font: 10 pt, English (US)

| Page 9: [129] Formatted | Sun ji | 6/11/18 1:55:00 PM |
|---|---|---|

Font: 10 pt, English (US)

| Page 9: [130] Formatted | Sun ji | 6/11/18 1:55:00 PM |
|---|---|---|

English (US)

| Page 9: [130] Formatted | Sun ji | 6/11/18 1:55:00 PM |
|---|---|---|

English (US)

| Page 10: [131] Formatted | Sun ji | 6/11/18 1:55:00 PM |
|---|---|---|

Font: 10 pt, English (US)

| Page 10: [132] Formatted | Sun ji | 6/11/18 1:55:00 PM |
|---|---|---|

Font: 10 pt, English (US)

| Page 10: [133] Deleted | Sun ji | 8/10/18 2:45:00 PM |
|---|---|---|

each

| Page 10: [133] Deleted | Sun ji | 8/10/18 2:45:00 PM |
|---|---|---|

each

| Page 10: [133] Deleted | Sun ji | 8/10/18 2:45:00 PM |
|---|---|---|

each

| Page 10: [134] Formatted | Sun ji | 6/11/18 1:55:00 PM |
|---|---|---|

Font: 10 pt, English (US)

| Page 10: [135] Formatted | Sun ji | 6/11/18 1:55:00 PM |
|---|---|---|

Font: 10 pt, English (US)

| Page 10: [136] Formatted | Sun ji | 5/15/18 10:32:00 PM |
|---|---|---|

Left

| Page 10: [137] Formatted | Sun ji | 6/11/18 1:55:00 PM |
|---|---|---|

Font: 10 pt, English (US)

| Page 10: [137] Formatted | Sun ji | 6/11/18 1:55:00 PM |
|---|---|---|

Font: 10 pt, English (US)

| Page 10: [138] Formatted | Sun ji | 6/11/18 1:55:00 PM |
|---|---|---|

Font: 10 pt, English (US)

| Page 10: [138] Formatted | Sun ji | 6/11/18 1:55:00 PM |
|---|---|---|

Font: 10 pt, English (US)

| Page 10: [138] Formatted | Sun ji | 6/11/18 1:55:00 PM |
|---|---|---|

Font: 10 pt, English (US)

| Page 10: [139] Formatted | Sun ji | 6/11/18 1:55:00 PM |
|---|---|---|

Font: 10 pt, English (US)

| Page 10: [139] Formatted | Sun ji | 6/11/18 1:55:00 PM |
|---|---|---|

Font: 10 pt, English (US)

| Page 10: [139] Formatted | Sun ji | 6/11/18 1:55:00 PM |
|---|---|---|

Font: 10 pt, English (US)

| Page 10: [139] Formatted | Sun ji | 6/11/18 1:55:00 PM |
|---|---|---|

Font: 10 pt, English (US)

| Page 10: [139] Formatted | Sun ji | 6/11/18 1:55:00 PM |
|---|---|---|

Font: 10 pt, English (US)

| Page 10: [140] Formatted | Sun ji | 6/11/18 1:55:00 PM |
|---|---|---|

Font: 10 pt, English (US)

| Page 10: [141] Formatted | Sun ji | 6/11/18 1:55:00 PM |
|---|---|---|

English (US)

| Page 10: [142] Formatted | Sun ji | 6/11/18 1:55:00 PM |
|---|---|---|

Font: 10 pt, English (US)

| Page 10: [142] Formatted | Sun ji | 6/11/18 1:55:00 PM |
|---|---|---|

Font: 10 pt, English (US)

| Page 10: [142] Formatted | Sun ji | 6/11/18 1:55:00 PM |
|---|---|---|

Font: 10 pt, English (US)

| Page 10: [143] Formatted | Sun ji | 6/11/18 1:55:00 PM |
|---|---|---|

Font: 10 pt, English (US)

| Page 10: [143] Formatted | Sun ji | 6/11/18 1:55:00 PM |
|---|---|---|

Font: 10 pt, English (US)

| Page 10: [143] Formatted | Sun ji | 6/11/18 1:55:00 PM |
|---|---|---|

Font: 10 pt, English (US)

| Page 10: [143] Formatted | Sun ji | 6/11/18 1:55:00 PM |
|---|---|---|

Font: 10 pt, English (US)

| Page 10: [144] Formatted | Sun ji | 6/11/18 1:55:00 PM |
| --- | --- | --- |

Font: 10 pt, English (US)

| Page 10: [144] Formatted | Sun ji | 6/11/18 1:55:00 PM |
| --- | --- | --- |

Font: 10 pt, English (US)

| Page 10: [144] Formatted | Sun ji | 6/11/18 1:55:00 PM |
| --- | --- | --- |

Font: 10 pt, English (US)

| Page 10: [145] Formatted | Sun ji | 6/11/18 1:55:00 PM |
| --- | --- | --- |

Font: 10 pt, English (US)

| Page 10: [146] Deleted | JP Veefkind | 5/14/18 1:04:00 PM |
| --- | --- | --- |

The asymmetry factor g is the averaged cosine of the scattering angle Θ, weighted by P(Θ).

| Page 10: [147] Formatted | Sun ji | 6/11/18 1:55:00 PM |
| --- | --- | --- |

English (US)

| Page 10: [148] Formatted | Sun ji | 6/11/18 1:55:00 PM |
| --- | --- | --- |

Font: 10 pt, English (US)

| Page 10: [149] Formatted | Sun ji | 6/11/18 1:55:00 PM |
| --- | --- | --- |

English (US)

| Page 10: [149] Formatted | Sun ji | 6/11/18 1:55:00 PM |
| --- | --- | --- |

English (US)

| Page 10: [150] Formatted | Sun ji | 6/11/18 1:55:00 PM |
| --- | --- | --- |

Font: 10 pt, English (US)

| Page 10: [150] Formatted | Sun ji | 6/11/18 1:55:00 PM |
| --- | --- | --- |

Font: 10 pt, English (US)

| Page 10: [150] Formatted | Sun ji | 6/11/18 1:55:00 PM |
| --- | --- | --- |

Font: 10 pt, English (US)

| Page 10: [150] Formatted | Sun ji | 6/11/18 1:55:00 PM |
| --- | --- | --- |

Font: 10 pt, English (US)

| Page 10: [150] Formatted | Sun ji | 6/11/18 1:55:00 PM |
| --- | --- | --- |

Font: 10 pt, English (US)

| Page 10: [151] Deleted | JP Veefkind | 5/14/18 1:05:00 PM |

(

| Page 10: [151] Deleted | JP Veefkind | 5/14/18 1:05:00 PM |

(

| Page 10: [151] Deleted | JP Veefkind | 5/14/18 1:05:00 PM |

(

| Page 10: [152] Formatted | Sun ji | 6/11/18 1:55:00 PM |

Font: 10 pt, English (US)

| Page 10: [153] Formatted | Sun ji | 6/11/18 1:55:00 PM |

Font: 10 pt, English (US)

| Page 10: [153] Formatted | Sun ji | 6/11/18 1:55:00 PM |

Font: 10 pt, English (US)

| Page 10: [153] Formatted | Sun ji | 6/11/18 1:55:00 PM |

Font: 10 pt, English (US)

| Page 10: [154] Deleted | JP Veefkind | 5/14/18 3:56:00 PM |

As shown in Fig.3 (e) and (f), even with a decreasing $\omega_0$ and an increasing g, or alternatively a decreasing $I_{\lambda 1}^{obs}$, the AAI primarily follows the behaviour of $\Delta I_{\lambda 1}$. The significant reduction in the spectral dependency of $I_\lambda$ overwhelms the high reflectivity for small particles ($r_g$=0.1μm).

| Page 10: [155] Formatted | Sun ji | 6/11/18 1:55:00 PM |

Font: 10 pt, English (US)

| Page 10: [155] Formatted | Sun ji | 6/11/18 1:55:00 PM |

Font: 10 pt, English (US)

| Page 10: [155] Formatted | Sun ji | 6/11/18 1:55:00 PM |

Font: 10 pt, English (US)

| Page 10: [155] Formatted | Sun ji | 6/11/18 1:55:00 PM |

Font: 10 pt, English (US)

| Page 10: [156] Formatted | Sun ji | 6/11/18 1:55:00 PM |

Font: 10 pt, English (US)

| Page 10: [156] Formatted | Sun ji | 6/11/18 1:55:00 PM |

Font: 10 pt, English (US)

| Page 10: [156] Formatted | Sun ji | 6/11/18 1:55:00 PM |

Font: 10 pt, English (US)

| Page 10: [157] Formatted | Sun ji | 6/11/18 1:55:00 PM |
|---|---|---|

Font: 10 pt, English (US)

| Page 10: [158] Formatted | Sun ji | 6/11/18 1:55:00 PM |
|---|---|---|

Font: 10 pt, English (US)

| Page 10: [158] Formatted | Sun ji | 6/11/18 1:55:00 PM |
|---|---|---|

Font: 10 pt, English (US)

| Page 10: [158] Formatted | Sun ji | 6/11/18 1:55:00 PM |
|---|---|---|

Font: 10 pt, English (US)

| Page 10: [159] Formatted | Sun ji | 6/11/18 1:55:00 PM |
|---|---|---|

Font: 10 pt, English (US), Highlight

| Page 10: [159] Formatted | Sun ji | 6/11/18 1:55:00 PM |
|---|---|---|

Font: 10 pt, English (US), Highlight

| Page 10: [159] Formatted | Sun ji | 6/11/18 1:55:00 PM |
|---|---|---|

Font: 10 pt, English (US), Highlight

| Page 10: [159] Formatted | Sun ji | 6/11/18 1:55:00 PM |
|---|---|---|

Font: 10 pt, English (US), Highlight

| Page 10: [159] Formatted | Sun ji | 6/11/18 1:55:00 PM |
|---|---|---|

Font: 10 pt, English (US), Highlight

| Page 10: [159] Formatted | Sun ji | 6/11/18 1:55:00 PM |
|---|---|---|

Font: 10 pt, English (US), Highlight

| Page 10: [159] Formatted | Sun ji | 6/11/18 1:55:00 PM |
|---|---|---|

Font: 10 pt, English (US), Highlight

| Page 10: [159] Formatted | Sun ji | 6/11/18 1:55:00 PM |
|---|---|---|

Font: 10 pt, English (US), Highlight

| Page 10: [160] Formatted | Sun ji | 6/11/18 1:55:00 PM |
|---|---|---|

Font: 10 pt, English (US)

| Page 10: [160] Formatted | Sun ji | 6/11/18 1:55:00 PM |
|---|---|---|

Font: 10 pt, English (US)

| Page 10: [160] Formatted | Sun ji | 6/11/18 1:55:00 PM |
|---|---|---|

Font: 10 pt, English (US)

| Page 10: [160] Formatted | Sun ji | 6/11/18 1:55:00 PM |
|---|---|---|

Font: 10 pt, English (US)

| Page 10: [161] Formatted | Sun ji | 6/11/18 1:55:00 PM |
|---|---|---|

Font: 10 pt, English (US)

| Page 10: [161] Formatted | Sun ji | 6/11/18 1:55:00 PM |
|---|---|---|

Font: 10 pt, English (US)

| Page 10: [161] Formatted | Sun ji | 6/11/18 1:55:00 PM |
|---|---|---|

Font: 10 pt, English (US)

| Page 10: [162] Deleted | Sun ji | 5/25/18 10:59:00 AM |
|---|---|---|

possible

| Page 10: [162] Deleted | Sun ji | 5/25/18 10:59:00 AM |
|---|---|---|

possible

| Page 10: [163] Formatted | Sun ji | 6/11/18 1:55:00 PM |
|---|---|---|

English (US)

| Page 10: [164] Formatted | Sun ji | 6/11/18 1:55:00 PM |
|---|---|---|

Font: 10 pt, English (US)

| Page 10: [164] Formatted | Sun ji | 6/11/18 1:55:00 PM |
|---|---|---|

Font: 10 pt, English (US)

| Page 10: [164] Formatted | Sun ji | 6/11/18 1:55:00 PM |
|---|---|---|

Font: 10 pt, English (US)

| Page 10: [165] Formatted | Sun ji | 6/11/18 1:55:00 PM |
|---|---|---|

Font: 10 pt, English (US)

| Page 10: [165] Formatted | Sun ji | 6/11/18 1:55:00 PM |
|---|---|---|

Font: 10 pt, English (US)

| Page 10: [165] Formatted | Sun ji | 6/11/18 1:55:00 PM |
|---|---|---|

Font: 10 pt, English (US)

| Page 10: [166] Deleted | Sun ji | 8/10/18 2:53:00 PM |
|---|---|---|

of the aerosol layer

| Page 10: [166] Deleted | Sun ji | 8/10/18 2:53:00 PM |
|---|---|---|

of the aerosol layer

| Page 10: [167] Formatted | Sun ji | 6/11/18 1:55:00 PM |
|---|---|---|

Font: 10 pt, English (US)

| Page 10: [168] Formatted | Sun ji | 6/11/18 1:55:00 PM |
|---|---|---|

Font: 10 pt, English (US)

| Page 10: [168] Formatted | Sun ji | 6/11/18 1:55:00 PM |
|---|---|---|

Font: 10 pt, English (US)

| Page 10: [169] Formatted | Sun ji | 6/11/18 1:55:00 PM |
|---|---|---|

Font: 10 pt, English (US)

| Page 10: [169] Formatted | Sun ji | 6/11/18 1:55:00 PM |
|---|---|---|

Font: 10 pt, English (US)

| Page 10: [170] Formatted | Sun ji | 6/11/18 1:55:00 PM |
|---|---|---|

English (US)

| Page 10: [171] Formatted | Sun ji | 6/11/18 1:55:00 PM |
|---|---|---|

Font: 10 pt, English (US)

| Page 10: [171] Formatted | Sun ji | 6/11/18 1:55:00 PM |
|---|---|---|

Font: 10 pt, English (US)

| Page 10: [172] Formatted | Sun ji | 6/11/18 1:55:00 PM |
|---|---|---|

Font: 10 pt, English (US)

| Page 10: [173] Formatted | Sun ji | 6/11/18 1:55:00 PM |
|---|---|---|

Font: 10 pt, English (US)

| Page 10: [173] Formatted | Sun ji | 6/11/18 1:55:00 PM |
|---|---|---|

Font: 10 pt, English (US)

| Page 10: [173] Formatted | Sun ji | 6/11/18 1:55:00 PM |
|---|---|---|

Font: 10 pt, English (US)

| Page 10: [174] Formatted | Sun ji | 6/11/18 1:55:00 PM |
|---|---|---|

Font: 10 pt, English (US)

| Page 10: [175] Deleted | JP Veefkind | 5/14/18 4:04:00 PM |

, where it was found that the retrieved AAI could be highly overestimated without correction for terrain height

| Page 10: [176] Formatted | Sun ji | 6/11/18 1:55:00 PM |

Font: 10 pt, English (US)

| Page 10: [177] Formatted | Sun ji | 6/11/18 1:55:00 PM |

Font: 10 pt, English (US)

| Page 10: [177] Formatted | Sun ji | 6/11/18 1:55:00 PM |

Font: 10 pt, English (US)

| Page 10: [177] Formatted | Sun ji | 6/11/18 1:55:00 PM |

Font: 10 pt, English (US)

| Page 10: [177] Formatted | Sun ji | 6/11/18 1:55:00 PM |

Font: 10 pt, English (US)

| Page 10: [178] Formatted | Sun ji | 6/11/18 1:55:00 PM |

Font: 10 pt, English (US)

| Page 10: [178] Formatted | Sun ji | 6/11/18 1:55:00 PM |

Font: 10 pt, English (US)

| Page 10: [178] Formatted | Sun ji | 6/11/18 1:55:00 PM |

Font: 10 pt, English (US)

| Page 10: [179] Formatted | Sun ji | 6/11/18 1:55:00 PM |

Font: 10 pt, English (US)

| Page 10: [180] Formatted | Sun ji | 6/11/18 1:55:00 PM |

Font: 10 pt, English (US)

| Page 10: [181] Deleted | Sun ji | 8/10/18 2:57:00 PM |

also

| Page 10: [181] Deleted | Sun ji | 8/10/18 2:57:00 PM |

also

| Page 10: [182] Formatted | Sun ji | 6/11/18 1:55:00 PM |

Font: 10 pt, English (US)

| Page 10: [182] Formatted | Sun ji | 6/11/18 1:55:00 PM |

Font: 10 pt, English (US)

| | | |
|---|---|---|
| **Page 10: [182] Formatted** | **Sun ji** | **6/11/18 1:55:00 PM** |

Font: 10 pt, English (US)

| | | |
|---|---|---|
| **Page 10: [182] Formatted** | **Sun ji** | **6/11/18 1:55:00 PM** |

Font: 10 pt, English (US)

| | | |
|---|---|---|
| **Page 10: [182] Formatted** | **Sun ji** | **6/11/18 1:55:00 PM** |

Font: 10 pt, English (US)

| | | |
|---|---|---|
| **Page 10: [182] Formatted** | **Sun ji** | **6/11/18 1:55:00 PM** |

Font: 10 pt, English (US)

| | | |
|---|---|---|
| **Page 10: [182] Formatted** | **Sun ji** | **6/11/18 1:55:00 PM** |

Font: 10 pt, English (US)

| | | |
|---|---|---|
| **Page 10: [183] Formatted** | **Sun ji** | **6/11/18 1:55:00 PM** |

Font: 10 pt, English (US)

| | | |
|---|---|---|
| **Page 10: [184] Formatted** | **Sun ji** | **6/11/18 1:55:00 PM** |

Font: 10 pt, English (US)

| | | |
|---|---|---|
| **Page 10: [185] Formatted** | **Sun ji** | **6/11/18 1:55:00 PM** |

Font: 10 pt, English (US)

| | | |
|---|---|---|
| **Page 10: [185] Formatted** | **Sun ji** | **6/11/18 1:55:00 PM** |

Font: 10 pt, English (US)

| | | |
|---|---|---|
| **Page 10: [185] Formatted** | **Sun ji** | **6/11/18 1:55:00 PM** |

Font: 10 pt, English (US)

| | | |
|---|---|---|
| **Page 10: [185] Formatted** | **Sun ji** | **6/11/18 1:55:00 PM** |

Font: 10 pt, English (US)

| | | |
|---|---|---|
| **Page 10: [186] Formatted** | **Sun ji** | **6/11/18 1:55:00 PM** |

Font: 10 pt, English (US)

| | | |
|---|---|---|
| **Page 10: [186] Formatted** | **Sun ji** | **6/11/18 1:55:00 PM** |

Font: 10 pt, English (US)

| | | |
|---|---|---|
| **Page 10: [186] Formatted** | **Sun ji** | **6/11/18 1:55:00 PM** |

Font: 10 pt, English (US)

| | | |
|---|---|---|
| **Page 10: [186] Formatted** | **Sun ji** | **6/11/18 1:55:00 PM** |

Font: 10 pt, English (US)

| Page 10: [186] Formatted | Sun ji | 6/11/18 1:55:00 PM |

Font: 10 pt, English (US)

| Page 11: [187] Formatted | Sun ji | 6/11/18 1:55:00 PM |

Font: 10 pt, English (US)

| Page 11: [187] Formatted | Sun ji | 6/11/18 1:55:00 PM |

Font: 10 pt, English (US)

| Page 11: [187] Formatted | Sun ji | 6/11/18 1:55:00 PM |

Font: 10 pt, English (US)

| Page 11: [187] Formatted | Sun ji | 6/11/18 1:55:00 PM |

Font: 10 pt, English (US)

| Page 11: [187] Formatted | Sun ji | 6/11/18 1:55:00 PM |

Font: 10 pt, English (US)

| Page 11: [187] Formatted | Sun ji | 6/11/18 1:55:00 PM |

Font: 10 pt, English (US)

| Page 11: [187] Formatted | Sun ji | 6/11/18 1:55:00 PM |

Font: 10 pt, English (US)

| Page 11: [187] Formatted | Sun ji | 6/11/18 1:55:00 PM |

Font: 10 pt, English (US)

| Page 11: [188] Deleted | Sun ji | 5/16/18 5:25:00 PM |

corresponding to the selected $\Theta$ does not strictly follow the changes in $P(\Theta)$

| Page 11: [189] Formatted | Sun ji | 6/11/18 1:55:00 PM |

Font: 10 pt, English (US)

| Page 11: [190] Formatted | Sun ji | 6/11/18 1:55:00 PM |

English (US), Not Highlight

| Page 11: [190] Formatted | Sun ji | 6/11/18 1:55:00 PM |

English (US), Not Highlight

| Page 11: [191] Deleted | Sun ji | 5/16/18 3:18:00 PM |

the reason that the length of the light path through the aerosol layer also varies with the measurement geometry.

Although the overall change in $P(\Theta)$ with an increasing $\Theta$ is negative, the light path within the aerosol layer also

decreases. Less absorption occurring in the aerosol layer overwhelms the decrease in reflectivity for larger Θ, resulting in an increase in $I_{\lambda 1}^{obs}$ with Θ. [Sj1]

| Page 11: [192] Formatted | Sun ji | 6/11/18 1:55:00 PM |
|---|---|---|

Font: 10 pt, English (US)

| Page 11: [192] Formatted | Sun ji | 6/11/18 1:55:00 PM |
|---|---|---|

Font: 10 pt, English (US)

| Page 11: [192] Formatted | Sun ji | 6/11/18 1:55:00 PM |
|---|---|---|

Font: 10 pt, English (US)

| Page 11: [192] Formatted | Sun ji | 6/11/18 1:55:00 PM |
|---|---|---|

Font: 10 pt, English (US)

| Page 11: [193] Commented [Sj9] | Sun ji | 5/16/18 12:11:00 PM |
|---|---|---|

I reconsider the reason, and also set up a simulation under the same condition but without aerosol (not included in the manuscript). It turns out the Rayleigh scattering at forward and backward direction is strong.

| Page 11: [194] Formatted | Sun ji | 6/11/18 1:55:00 PM |
|---|---|---|

English (US), Not Highlight

| Page 11: [194] Formatted | Sun ji | 6/11/18 1:55:00 PM |
|---|---|---|

English (US), Not Highlight

| Page 11: [195] Formatted | Sun ji | 6/11/18 1:55:00 PM |
|---|---|---|

English (US), Not Highlight

| Page 11: [195] Formatted | Sun ji | 6/11/18 1:55:00 PM |
|---|---|---|

English (US), Not Highlight

| Page 11: [196] Formatted | Sun ji | 6/11/18 1:55:00 PM |
|---|---|---|

Font: 10 pt, English (US)

| Page 11: [197] Deleted | Sun ji | 5/15/18 4:41:00 PM |
|---|---|---|

Although the DISAMAR can calculate wavelength dependent, ω₀ at 550 nm is used as retrieved value for the consistent comparison with AERONET measurements. [JV2][Sj3]

| Page 11: [198] Formatted | Sun ji | 6/11/18 1:55:00 PM |
|---|---|---|

Font: 5.5 pt

| Page 11: [199] Formatted | Sun ji | 6/11/18 1:55:00 PM |
|---|---|---|

Font: 10 pt, English (US)

| Page 11: [200] Formatted | Sun ji | 6/11/18 1:55:00 PM |
|---|---|---|

English (US)

| Page 11: [200] Formatted | Sun ji | 6/11/18 1:55:00 PM |
|---|---|---|

English (US)

| Page 11: [201] Formatted | Sun ji | 6/11/18 1:55:00 PM |
|---|---|---|

Font: 10 pt, English (US)

| Page 11: [201] Formatted | Sun ji | 6/11/18 1:55:00 PM |
|---|---|---|

Font: 10 pt, English (US)

| Page 11: [202] Formatted | Sun ji | 6/11/18 1:55:00 PM |
|---|---|---|

Font: 10 pt, English (US)

| Page 11: [202] Formatted | Sun ji | 6/11/18 1:55:00 PM |
|---|---|---|

Font: 10 pt, English (US)

| Page 11: [202] Formatted | Sun ji | 6/11/18 1:55:00 PM |
|---|---|---|

Font: 10 pt, English (US)

| Page 11: [203] Formatted | Sun ji | 6/11/18 1:55:00 PM |
|---|---|---|

English (US)

| Page 11: [204] Formatted | Sun ji | 6/11/18 1:55:00 PM |
|---|---|---|

Font: 10 pt, English (US)

| Page 11: [205] Formatted | Sun ji | 6/11/18 1:55:00 PM |
|---|---|---|

Font: 10 pt, English (US)

| Page 11: [206] Formatted | Sun ji | 6/11/18 1:55:00 PM |
|---|---|---|

English (US)

| Page 11: [206] Formatted | Sun ji | 6/11/18 1:55:00 PM |
|---|---|---|

English (US)

| Page 11: [207] Formatted | Sun ji | 6/11/18 1:55:00 PM |
|---|---|---|

Font: 10 pt, English (US)

| Page 11: [208] Formatted | Sun ji | 6/11/18 1:55:00 PM |
|---|---|---|

Font: 10 pt, (Asian) Chinese (China), (Other) English (US)

| Page 11: [209] Deleted | JP Veefkind | 5/14/18 4:13:00 PM |
|---|---|---|

plume

| Page 11: [209] Deleted | JP Veefkind | 5/14/18 4:13:00 PM |
|---|---|---|

plume

| Page 11: [210] Formatted | Sun ji | 6/11/18 1:55:00 PM |
|---|---|---|

Font: 10 pt, English (US)

| Page 11: [211] Deleted | Sun ji | 8/10/18 3:11:00 PM |
|---|---|---|

,

| Page 11: [211] Deleted | Sun ji | 8/10/18 3:11:00 PM |
|---|---|---|

,

| Page 11: [212] Formatted | Sun ji | 6/11/18 1:55:00 PM |
|---|---|---|

Font: 10 pt, English (US)

| Page 11: [213] Formatted | Sun ji | 6/11/18 1:55:00 PM |
|---|---|---|

English (US)

| Page 11: [213] Formatted | Sun ji | 6/11/18 1:55:00 PM |
|---|---|---|

English (US)

| Page 11: [214] Deleted | JP Veefkind | 5/14/18 4:15:00 PM |
|---|---|---|

a

| Page 11: [214] Deleted | JP Veefkind | 5/14/18 4:15:00 PM |
|---|---|---|

a

| Page 11: [214] Deleted | JP Veefkind | 5/14/18 4:15:00 PM |
|---|---|---|

a

| Page 11: [215] Formatted | Sun ji | 6/11/18 1:55:00 PM |
|---|---|---|

Font: 10 pt, English (US), Highlight

| Page 11: [215] Formatted | Sun ji | 6/11/18 1:55:00 PM |
|---|---|---|

Font: 10 pt, English (US), Highlight

| Page 11: [215] Formatted | Sun ji | 6/11/18 1:55:00 PM |
|---|---|---|

Font: 10 pt, English (US), Highlight

| Page 11: [216] Deleted | JP Veefkind | 5/14/18 4:17:00 PM |
|---|---|---|

the spectrum ranges from

| Page 11: [216] Deleted | JP Veefkind | 5/14/18 4:17:00 PM |
|---|---|---|

the spectrum ranges from

| Page 11: [216] Deleted | JP Veefkind | 5/14/18 4:17:00 PM |
|---|---|---|

the spectrum ranges from

| Page 11: [217] Formatted | Sun ji | 6/11/18 1:55:00 PM |

Font: 10 pt, English (US)

| Page 11: [217] Formatted | Sun ji | 6/11/18 1:55:00 PM |

Font: 10 pt, English (US)

| Page 11: [218] Formatted | Sun ji | 6/11/18 1:55:00 PM |

Font: 10 pt, English (US)

| Page 11: [218] Formatted | Sun ji | 6/11/18 1:55:00 PM |

Font: 10 pt, English (US)

| Page 11: [218] Formatted | Sun ji | 6/11/18 1:55:00 PM |

Font: 10 pt, English (US)

| Page 11: [218] Formatted | Sun ji | 6/11/18 1:55:00 PM |

Font: 10 pt, English (US)

| Page 11: [218] Formatted | Sun ji | 6/11/18 1:55:00 PM |

Font: 10 pt, English (US)

| Page 11: [218] Formatted | Sun ji | 6/11/18 1:55:00 PM |

Font: 10 pt, English (US)

| Page 11: [218] Formatted | Sun ji | 6/11/18 1:55:00 PM |

Font: 10 pt, English (US)

| Page 11: [218] Formatted | Sun ji | 6/11/18 1:55:00 PM |

Font: 10 pt, English (US)

| Page 11: [218] Formatted | Sun ji | 6/11/18 1:55:00 PM |

Font: 10 pt, English (US)

| Page 11: [219] Formatted | Sun ji | 6/11/18 1:55:00 PM |

Font: 10 pt, English (US)

| Page 11: [219] Formatted | Sun ji | 6/11/18 1:55:00 PM |

Font: 10 pt, English (US)

| Page 11: [219] Formatted | Sun ji | 6/11/18 1:55:00 PM |

Font: 10 pt, English (US)

| Page 11: [220] Formatted | Sun ji | 6/11/18 1:55:00 PM |

Font: 5.5 pt

| Page 11: [221] Formatted | Sun ji | 6/11/18 1:55:00 PM |
|---|---|---|

Font: 10 pt, English (US)

| Page 11: [221] Formatted | Sun ji | 6/11/18 1:55:00 PM |
|---|---|---|

Font: 10 pt, English (US)

| Page 11: [221] Formatted | Sun ji | 6/11/18 1:55:00 PM |
|---|---|---|

Font: 10 pt, English (US)

| Page 11: [222] Deleted | Sun ji | 8/7/18 9:16:00 AM |
|---|---|---|

treated as a reference dataset to evaluate potential biases in

| Page 11: [223] Formatted | Sun ji | 6/11/18 1:55:00 PM |
|---|---|---|

Font: 10 pt, English (US)

| Page 11: [223] Formatted | Sun ji | 6/11/18 1:55:00 PM |
|---|---|---|

Font: 10 pt, English (US)

| Page 11: [223] Formatted | Sun ji | 6/11/18 1:55:00 PM |
|---|---|---|

Font: 10 pt, English (US)

| Page 12: [224] Formatted | Sun ji | 6/11/18 1:55:00 PM |
|---|---|---|

Font: 10 pt, English (US)

| Page 12: [224] Formatted | Sun ji | 6/11/18 1:55:00 PM |
|---|---|---|

Font: 10 pt, English (US)

| Page 12: [224] Formatted | Sun ji | 6/11/18 1:55:00 PM |
|---|---|---|

Font: 10 pt, English (US)

| Page 12: [224] Formatted | Sun ji | 6/11/18 1:55:00 PM |
|---|---|---|

Font: 10 pt, English (US)

| Page 12: [225] Formatted | Sun ji | 6/11/18 1:55:00 PM |
|---|---|---|

English (US)

| Page 12: [225] Formatted | Sun ji | 6/11/18 1:55:00 PM |
|---|---|---|

English (US)

| Page 12: [225] Formatted | Sun ji | 6/11/18 1:55:00 PM |
|---|---|---|

English (US)

| Page 12: [226] Formatted | Sun ji | 6/11/18 1:55:00 PM |
|---|---|---|

Font: 10 pt, English (US)

| Page 12: [226] Formatted | Sun ji | 6/11/18 1:55:00 PM |

Font: 10 pt, English (US)

| Page 12: [227] Formatted | Sun ji | 6/11/18 1:55:00 PM |

Font: 10 pt, English (US), Highlight

| Page 12: [227] Formatted | Sun ji | 6/11/18 1:55:00 PM |

Font: 10 pt, English (US), Highlight

| Page 12: [227] Formatted | Sun ji | 6/11/18 1:55:00 PM |

Font: 10 pt, English (US), Highlight

| Page 12: [227] Formatted | Sun ji | 6/11/18 1:55:00 PM |

Font: 10 pt, English (US), Highlight

| Page 12: [228] Formatted | Sun ji | 6/11/18 1:55:00 PM |

English (US)

| Page 12: [228] Formatted | Sun ji | 6/11/18 1:55:00 PM |

English (US)

| Page 12: [229] Formatted | Sun ji | 6/11/18 1:55:00 PM |

Font: 10 pt, English (US)

| Page 12: [229] Formatted | Sun ji | 6/11/18 1:55:00 PM |

Font: 10 pt, English (US)

| Page 12: [229] Formatted | Sun ji | 6/11/18 1:55:00 PM |

Font: 10 pt, English (US)

| Page 12: [229] Formatted | Sun ji | 6/11/18 1:55:00 PM |

Font: 10 pt, English (US)

| Page 12: [229] Formatted | Sun ji | 6/11/18 1:55:00 PM |

Font: 10 pt, English (US)

| Page 12: [229] Formatted | Sun ji | 6/11/18 1:55:00 PM |

Font: 10 pt, English (US)

| Page 12: [229] Formatted | Sun ji | 6/11/18 1:55:00 PM |

Font: 10 pt, English (US)

| Page 12: [229] Formatted | Sun ji | 6/11/18 1:55:00 PM |

Font: 10 pt, English (US)

| Page 12: [229] Formatted | Sun ji | 6/11/18 1:55:00 PM |

Font: 10 pt, English (US)

| Page 12: [230] Formatted | Sun ji | 6/11/18 1:55:00 PM |

English (US)

| Page 12: [230] Formatted | Sun ji | 6/11/18 1:55:00 PM |

English (US)

| Page 12: [231] Formatted | Sun ji | 6/11/18 1:55:00 PM |

English (US)

| Page 12: [231] Formatted | Sun ji | 6/11/18 1:55:00 PM |

English (US)

| Page 12: [232] Formatted | Sun ji | 6/11/18 1:55:00 PM |

Font: 10 pt, English (US)

| Page 12: [232] Formatted | Sun ji | 6/11/18 1:55:00 PM |

Font: 10 pt, English (US)

| Page 12: [232] Formatted | Sun ji | 6/11/18 1:55:00 PM |

Font: 10 pt, English (US)

| Page 12: [233] Deleted | Sun ji | 5/16/18 11:16:00 AM |

inversion

| Page 12: [233] Deleted | Sun ji | 5/16/18 11:16:00 AM |

inversion

| Page 12: [233] Deleted | Sun ji | 5/16/18 11:16:00 AM |

inversion

| Page 12: [233] Deleted | Sun ji | 5/16/18 11:16:00 AM |

inversion

| Page 12: [234] Formatted | Sun ji | 6/11/18 1:55:00 PM |

Font: 10 pt, English (US)

| Page 12: [234] Formatted | Sun ji | 6/11/18 1:55:00 PM |

Font: 10 pt, English (US)

| Page 12: [235] Deleted | JP Veefkind | 5/14/18 4:20:00 PM |

when

| Page 12: [235] Deleted | JP Veefkind | 5/14/18 4:20:00 PM |
|---|---|---|

when

| Page 12: [236] Formatted | Sun ji | 6/11/18 1:55:00 PM |
|---|---|---|

English (US)

| Page 12: [236] Formatted | Sun ji | 6/11/18 1:55:00 PM |
|---|---|---|

English (US)

| Page 12: [236] Formatted | Sun ji | 6/11/18 1:55:00 PM |
|---|---|---|

English (US)

| Page 12: [237] Formatted | Sun ji | 6/11/18 1:55:00 PM |
|---|---|---|

Font: 10 pt, English (US)

| Page 12: [237] Formatted | Sun ji | 6/11/18 1:55:00 PM |
|---|---|---|

Font: 10 pt, English (US)

| Page 12: [238] Formatted | Sun ji | 6/11/18 1:55:00 PM |
|---|---|---|

English (US)

| Page 12: [238] Formatted | Sun ji | 6/11/18 1:55:00 PM |
|---|---|---|

English (US)

| Page 12: [238] Formatted | Sun ji | 6/11/18 1:55:00 PM |
|---|---|---|

English (US)

| Page 12: [238] Formatted | Sun ji | 6/11/18 1:55:00 PM |
|---|---|---|

English (US)

| Page 12: [238] Formatted | Sun ji | 6/11/18 1:55:00 PM |
|---|---|---|

English (US)

| Page 12: [239] Formatted | Sun ji | 6/11/18 1:55:00 PM |
|---|---|---|

Font: 10 pt, English (US)

| Page 12: [239] Formatted | Sun ji | 6/11/18 1:55:00 PM |
|---|---|---|

Font: 10 pt, English (US)

| Page 12: [240] Formatted | Sun ji | 6/11/18 1:55:00 PM |
|---|---|---|

Font: 10 pt, English (US)

| Page 12: [240] Formatted | Sun ji | 6/11/18 1:55:00 PM |
|---|---|---|

Font: 10 pt, English (US)

| Page 12: [240] Formatted | | Sun ji | | 6/11/18 1:55:00 PM |
|---|---|---|---|---|

Font: 10 pt, English (US)

| Page 12: [240] Formatted | | Sun ji | | 6/11/18 1:55:00 PM |
|---|---|---|---|---|

Font: 10 pt, English (US)

| Page 12: [240] Formatted | | Sun ji | | 6/11/18 1:55:00 PM |
|---|---|---|---|---|

Font: 10 pt, English (US)

| Page 12: [240] Formatted | | Sun ji | | 6/11/18 1:55:00 PM |
|---|---|---|---|---|

Font: 10 pt, English (US)

| Page 12: [240] Formatted | | Sun ji | | 6/11/18 1:55:00 PM |
|---|---|---|---|---|

Font: 10 pt, English (US)

| Page 12: [240] Formatted | | Sun ji | | 6/11/18 1:55:00 PM |
|---|---|---|---|---|

Font: 10 pt, English (US)

| Page 12: [240] Formatted | | Sun ji | | 6/11/18 1:55:00 PM |
|---|---|---|---|---|

Font: 10 pt, English (US)

| Page 12: [240] Formatted | | Sun ji | | 6/11/18 1:55:00 PM |
|---|---|---|---|---|

Font: 10 pt, English (US)

| Page 12: [240] Formatted | | Sun ji | | 6/11/18 1:55:00 PM |
|---|---|---|---|---|

Font: 10 pt, English (US)

| Page 12: [240] Formatted | | Sun ji | | 6/11/18 1:55:00 PM |
|---|---|---|---|---|

Font: 10 pt, English (US)

| Page 12: [240] Formatted | | Sun ji | | 6/11/18 1:55:00 PM |
|---|---|---|---|---|

Font: 10 pt, English (US)

| Page 12: [240] Formatted | | Sun ji | | 6/11/18 1:55:00 PM |
|---|---|---|---|---|

Font: 10 pt, English (US)

| Page 12: [240] Formatted | | Sun ji | | 6/11/18 1:55:00 PM |
|---|---|---|---|---|

Font: 10 pt, English (US)

| Page 12: [240] Formatted | | Sun ji | | 6/11/18 1:55:00 PM |
|---|---|---|---|---|

Font: 10 pt, English (US)

| Page 12: [240] Formatted | | Sun ji | | 6/11/18 1:55:00 PM |
|---|---|---|---|---|

Font: 10 pt, English (US)

| Page 12: [240] Formatted | Sun ji | 6/11/18 1:55:00 PM |
|---|---|---|

Font: 10 pt, English (US)

| Page 12: [240] Formatted | Sun ji | 6/11/18 1:55:00 PM |
|---|---|---|

Font: 10 pt, English (US)

| Page 12: [240] Formatted | Sun ji | 6/11/18 1:55:00 PM |
|---|---|---|

Font: 10 pt, English (US)

| Page 12: [240] Formatted | Sun ji | 6/11/18 1:55:00 PM |
|---|---|---|

Font: 10 pt, English (US)

| Page 12: [240] Formatted | Sun ji | 6/11/18 1:55:00 PM |
|---|---|---|

Font: 10 pt, English (US)

| Page 12: [241] Formatted | Sun ji | 6/11/18 1:55:00 PM |
|---|---|---|

Font: 10 pt, English (US)

| Page 12: [241] Formatted | Sun ji | 6/11/18 1:55:00 PM |
|---|---|---|

Font: 10 pt, English (US)

| Page 12: [241] Formatted | Sun ji | 6/11/18 1:55:00 PM |
|---|---|---|

Font: 10 pt, English (US)

| Page 12: [241] Formatted | Sun ji | 6/11/18 1:55:00 PM |
|---|---|---|

Font: 10 pt, English (US)

| Page 12: [241] Formatted | Sun ji | 6/11/18 1:55:00 PM |
|---|---|---|

Font: 10 pt, English (US)

| Page 12: [241] Formatted | Sun ji | 6/11/18 1:55:00 PM |
|---|---|---|

Font: 10 pt, English (US)

| Page 12: [241] Formatted | Sun ji | 6/11/18 1:55:00 PM |
|---|---|---|

Font: 10 pt, English (US)

| Page 12: [241] Formatted | Sun ji | 6/11/18 1:55:00 PM |
|---|---|---|

Font: 10 pt, English (US)

| Page 12: [241] Formatted | Sun ji | 6/11/18 1:55:00 PM |
|---|---|---|

Font: 10 pt, English (US)

| Page 12: [241] Formatted | Sun ji | 6/11/18 1:55:00 PM |
|---|---|---|

Font: 10 pt, English (US)

| Page 12: [241] Formatted | Sun ji | 6/11/18 1:55:00 PM |

Font: 10 pt, English (US)

| Page 12: [242] Formatted | Sun ji | 6/11/18 1:55:00 PM |

Font: 10 pt, English (US)

| Page 12: [242] Formatted | Sun ji | 6/11/18 1:55:00 PM |

Font: 10 pt, English (US)

| Page 12: [242] Formatted | Sun ji | 6/11/18 1:55:00 PM |

Font: 10 pt, English (US)

| Page 12: [242] Formatted | Sun ji | 6/11/18 1:55:00 PM |

Font: 10 pt, English (US)

| Page 12: [242] Formatted | Sun ji | 6/11/18 1:55:00 PM |

Font: 10 pt, English (US)

| Page 12: [242] Formatted | Sun ji | 6/11/18 1:55:00 PM |

Font: 10 pt, English (US)

| Page 12: [243] Deleted | JP Veefkind | 5/14/18 4:22:00 PM |

.

| Page 12: [243] Deleted | JP Veefkind | 5/14/18 4:22:00 PM |

.

| Page 12: [243] Deleted | JP Veefkind | 5/14/18 4:22:00 PM |

.

| Page 12: [244] Deleted | Sun ji | 4/23/18 7:24:00 PM |

, and the optical properties of

| Page 12: [244] Deleted | Sun ji | 4/23/18 7:24:00 PM |

, and the optical properties of

| Page 12: [244] Deleted | Sun ji | 4/23/18 7:24:00 PM |

, and the optical properties of

| Page 12: [245] Formatted | Sun ji | 6/11/18 1:55:00 PM |

Font: 10 pt, English (US)

| Page 12: [245] Formatted | Sun ji | 6/11/18 1:55:00 PM |

Font: 10 pt, English (US)

| Page 12: [245] Formatted | Sun ji | 6/11/18 1:55:00 PM |

Font: 10 pt, English (US)

| Page 12: [245] Formatted | Sun ji | 6/11/18 1:55:00 PM |
|---|---|---|

Font: 10 pt, English (US)

| Page 12: [245] Formatted | Sun ji | 6/11/18 1:55:00 PM |
|---|---|---|

Font: 10 pt, English (US)

| Page 12: [245] Formatted | Sun ji | 6/11/18 1:55:00 PM |
|---|---|---|

Font: 10 pt, English (US)

| Page 12: [245] Formatted | Sun ji | 6/11/18 1:55:00 PM |
|---|---|---|

Font: 10 pt, English (US)

| Page 12: [245] Formatted | Sun ji | 6/11/18 1:55:00 PM |
|---|---|---|

Font: 10 pt, English (US)

| Page 12: [245] Formatted | Sun ji | 6/11/18 1:55:00 PM |
|---|---|---|

Font: 10 pt, English (US)

| Page 12: [245] Formatted | Sun ji | 6/11/18 1:55:00 PM |
|---|---|---|

Font: 10 pt, English (US)

| Page 12: [245] Formatted | Sun ji | 6/11/18 1:55:00 PM |
|---|---|---|

Font: 10 pt, English (US)

| Page 12: [245] Formatted | Sun ji | 6/11/18 1:55:00 PM |
|---|---|---|

Font: 10 pt, English (US)

| Page 12: [245] Formatted | Sun ji | 6/11/18 1:55:00 PM |
|---|---|---|

Font: 10 pt, English (US)

| Page 12: [245] Formatted | Sun ji | 6/11/18 1:55:00 PM |
|---|---|---|

Font: 10 pt, English (US)

| Page 12: [245] Formatted | Sun ji | 6/11/18 1:55:00 PM |
|---|---|---|

Font: 10 pt, English (US)

| Page 12: [245] Formatted | Sun ji | 6/11/18 1:55:00 PM |
|---|---|---|

Font: 10 pt, English (US)

| Page 12: [245] Formatted | Sun ji | 6/11/18 1:55:00 PM |
|---|---|---|

Font: 10 pt, English (US)

| Page 12: [245] Formatted | Sun ji | 6/11/18 1:55:00 PM |
|---|---|---|

Font: 10 pt, English (US)

| Page 12: [245] Formatted | Sun ji | 6/11/18 1:55:00 PM |
|---|---|---|

Font: 10 pt, English (US)

| Page 12: [245] Formatted | Sun ji | 6/11/18 1:55:00 PM |
|---|---|---|

Font: 10 pt, English (US)

| Page 12: [245] Formatted | Sun ji | 6/11/18 1:55:00 PM |
|---|---|---|

Font: 10 pt, English (US)

| Page 12: [245] Formatted | Sun ji | 6/11/18 1:55:00 PM |
|---|---|---|

Font: 10 pt, English (US)

| Page 12: [245] Formatted | Sun ji | 6/11/18 1:55:00 PM |
|---|---|---|

Font: 10 pt, English (US)

| Page 12: [245] Formatted | Sun ji | 6/11/18 1:55:00 PM |
|---|---|---|

Font: 10 pt, English (US)

| Page 12: [245] Formatted | Sun ji | 6/11/18 1:55:00 PM |
|---|---|---|

Font: 10 pt, English (US)

| Page 12: [245] Formatted | Sun ji | 6/11/18 1:55:00 PM |
|---|---|---|

Font: 10 pt, English (US)

| Page 12: [245] Formatted | Sun ji | 6/11/18 1:55:00 PM |
|---|---|---|

Font: 10 pt, English (US)

| Page 12: [245] Formatted | Sun ji | 6/11/18 1:55:00 PM |
|---|---|---|

Font: 10 pt, English (US)

| Page 12: [246] Formatted | Sun ji | 6/11/18 1:55:00 PM |
|---|---|---|

Font: 10 pt, English (US)

| Page 12: [246] Formatted | Sun ji | 6/11/18 1:55:00 PM |
|---|---|---|

Font: 10 pt, English (US)

| Page 12: [246] Formatted | Sun ji | 6/11/18 1:55:00 PM |
|---|---|---|

Font: 10 pt, English (US)

| Page 12: [246] Formatted | Sun ji | 6/11/18 1:55:00 PM |
|---|---|---|

Font: 10 pt, English (US)

| Page 12: [246] Formatted | Sun ji | 6/11/18 1:55:00 PM |
|---|---|---|

Font: 10 pt, English (US)

| Page 12: [246] Formatted | Sun ji | 6/11/18 1:55:00 PM |
|---|---|---|

Font: 10 pt, English (US)

| Page 12: [247] Formatted | Sun ji | 6/11/18 1:55:00 PM |
|---|---|---|

Font: 10 pt, English (US)

| Page 12: [247] Formatted | Sun ji | 6/11/18 1:55:00 PM |
|---|---|---|

Font: 10 pt, English (US)

| Page 12: [247] Formatted | Sun ji | 6/11/18 1:55:00 PM |
|---|---|---|

Font: 10 pt, English (US)

| Page 12: [248] Formatted | Sun ji | 6/11/18 1:55:00 PM |
|---|---|---|

Font: 10 pt, English (US)

| Page 12: [248] Formatted | Sun ji | 6/11/18 1:55:00 PM |
|---|---|---|

Font: 10 pt, English (US)

| Page 12: [248] Formatted | Sun ji | 6/11/18 1:55:00 PM |
|---|---|---|

Font: 10 pt, English (US)

| Page 12: [248] Formatted | Sun ji | 6/11/18 1:55:00 PM |
|---|---|---|

Font: 10 pt, English (US)

| Page 12: [248] Formatted | Sun ji | 6/11/18 1:55:00 PM |
|---|---|---|

Font: 10 pt, English (US)

| Page 12: [248] Formatted | Sun ji | 6/11/18 1:55:00 PM |
|---|---|---|

Font: 10 pt, English (US)

| Page 12: [248] Formatted | Sun ji | 6/11/18 1:55:00 PM |
|---|---|---|

Font: 10 pt, English (US)

| Page 12: [248] Formatted | Sun ji | 6/11/18 1:55:00 PM |
|---|---|---|

Font: 10 pt, English (US)

| Page 12: [248] Formatted | Sun ji | 6/11/18 1:55:00 PM |
|---|---|---|

Font: 10 pt, English (US)

| Page 12: [248] Formatted | Sun ji | 6/11/18 1:55:00 PM |
|---|---|---|

Font: 10 pt, English (US)

| Page 12: [248] Formatted | Sun ji | 6/11/18 1:55:00 PM |
|---|---|---|

Font: 10 pt, English (US)

| Page 12: [248] Formatted | Sun ji | 6/11/18 1:55:00 PM |
|---|---|---|

Font: 10 pt, English (US)

| Page 12: [248] Formatted | Sun ji | 6/11/18 1:55:00 PM |
|---|---|---|

Font: 10 pt, English (US)

| Page 12: [248] Formatted | Sun ji | 6/11/18 1:55:00 PM |
|---|---|---|

Font: 10 pt, English (US)

| Page 12: [248] Formatted | Sun ji | 6/11/18 1:55:00 PM |
|---|---|---|

Font: 10 pt, English (US)

| Page 12: [248] Formatted | Sun ji | 6/11/18 1:55:00 PM |
|---|---|---|

Font: 10 pt, English (US)

| Page 13: [249] Formatted | Sun ji | 6/11/18 1:55:00 PM |
|---|---|---|

Font: 10 pt, English (US)

| Page 13: [249] Formatted | Sun ji | 6/11/18 1:55:00 PM |
|---|---|---|

Font: 10 pt, English (US)

| Page 13: [249] Formatted | Sun ji | 6/11/18 1:55:00 PM |
|---|---|---|

Font: 10 pt, English (US)

| Page 13: [249] Formatted | Sun ji | 6/11/18 1:55:00 PM |
|---|---|---|

Font: 10 pt, English (US)

| Page 13: [249] Formatted | Sun ji | 6/11/18 1:55:00 PM |
|---|---|---|

Font: 10 pt, English (US)

| Page 13: [249] Formatted | Sun ji | 6/11/18 1:55:00 PM |
|---|---|---|

Font: 10 pt, English (US)

| Page 13: [250] Formatted | Sun ji | 6/11/18 1:55:00 PM |
|---|---|---|

Font: 10 pt, English (US)

| Page 13: [250] Formatted | Sun ji | 6/11/18 1:55:00 PM |
|---|---|---|

Font: 10 pt, English (US)

| Page 13: [250] Formatted | Sun ji | 6/11/18 1:55:00 PM |
|---|---|---|

Font: 10 pt, English (US)

| Page 13: [250] Formatted | Sun ji | 6/11/18 1:55:00 PM |
|---|---|---|

Font: 10 pt, English (US)

| Page 13: [250] Formatted | Sun ji | 6/11/18 1:55:00 PM |
|---|---|---|

Font: 10 pt, English (US)

| Page 13: [250] Formatted | Sun ji | 6/11/18 1:55:00 PM |
|---|---|---|

Font: 10 pt, English (US)

| Page 13: [250] Formatted | Sun ji | 6/11/18 1:55:00 PM |
|---|---|---|

Font: 10 pt, English (US)

| Page 13: [250] Formatted | Sun ji | 6/11/18 1:55:00 PM |
|---|---|---|

Font: 10 pt, English (US)

| Page 13: [250] Formatted | Sun ji | 6/11/18 1:55:00 PM |
|---|---|---|

Font: 10 pt, English (US)

| Page 13: [250] Formatted | Sun ji | 6/11/18 1:55:00 PM |
|---|---|---|

Font: 10 pt, English (US)

| Page 13: [250] Formatted | Sun ji | 6/11/18 1:55:00 PM |
|---|---|---|

Font: 10 pt, English (US)

| Page 13: [250] Formatted | Sun ji | 6/11/18 1:55:00 PM |
|---|---|---|

Font: 10 pt, English (US)

| Page 13: [250] Formatted | Sun ji | 6/11/18 1:55:00 PM |
|---|---|---|

Font: 10 pt, English (US)

| Page 13: [250] Formatted | Sun ji | 6/11/18 1:55:00 PM |
|---|---|---|

Font: 10 pt, English (US)

| Page 13: [250] Formatted | Sun ji | 6/11/18 1:55:00 PM |
|---|---|---|

Font: 10 pt, English (US)

| Page 13: [250] Formatted | Sun ji | 6/11/18 1:55:00 PM |

Font: 10 pt, English (US)

| Page 13: [250] Formatted | Sun ji | 6/11/18 1:55:00 PM |

Font: 10 pt, English (US)

| Page 13: [250] Formatted | Sun ji | 6/11/18 1:55:00 PM |

Font: 10 pt, English (US)

| Page 13: [250] Formatted | Sun ji | 6/11/18 1:55:00 PM |

Font: 10 pt, English (US)

| Page 13: [250] Formatted | Sun ji | 6/11/18 1:55:00 PM |

Font: 10 pt, English (US)

| Page 13: [250] Formatted | Sun ji | 6/11/18 1:55:00 PM |

Font: 10 pt, English (US)

| Page 13: [250] Formatted | Sun ji | 6/11/18 1:55:00 PM |

Font: 10 pt, English (US)

| Page 13: [250] Formatted | Sun ji | 6/11/18 1:55:00 PM |

Font: 10 pt, English (US)

| Page 13: [251] Formatted | Sun ji | 6/11/18 1:55:00 PM |

Font: 10 pt, English (US)

| Page 13: [251] Formatted | Sun ji | 6/11/18 1:55:00 PM |

Font: 10 pt, English (US)

| Page 13: [251] Formatted | Sun ji | 6/11/18 1:55:00 PM |

Font: 10 pt, English (US)

| Page 13: [251] Formatted | Sun ji | 6/11/18 1:55:00 PM |

Font: 10 pt, English (US)

| Page 13: [251] Formatted | Sun ji | 6/11/18 1:55:00 PM |

Font: 10 pt, English (US)

| Page 13: [251] Formatted | Sun ji | 6/11/18 1:55:00 PM |

Font: 10 pt, English (US)

| Page 13: [251] Formatted | Sun ji | 6/11/18 1:55:00 PM |
|---|---|---|

Font: 10 pt, English (US)

| Page 13: [251] Formatted | Sun ji | 6/11/18 1:55:00 PM |
|---|---|---|

Font: 10 pt, English (US)

| Page 13: [251] Formatted | Sun ji | 6/11/18 1:55:00 PM |
|---|---|---|

Font: 10 pt, English (US)

| Page 13: [251] Formatted | Sun ji | 6/11/18 1:55:00 PM |
|---|---|---|

Font: 10 pt, English (US)

| Page 13: [252] Formatted | Sun ji | 6/17/18 9:26:00 PM |
|---|---|---|

Font color: Text 1

| Page 13: [252] Formatted | Sun ji | 6/17/18 9:26:00 PM |
|---|---|---|

Font color: Text 1

| Page 13: [252] Formatted | Sun ji | 6/17/18 9:26:00 PM |
|---|---|---|

Font color: Text 1

| Page 13: [252] Formatted | Sun ji | 6/17/18 9:26:00 PM |
|---|---|---|

Font color: Text 1

| Page 13: [252] Formatted | Sun ji | 6/17/18 9:26:00 PM |
|---|---|---|

Font color: Text 1

| Page 13: [252] Formatted | Sun ji | 6/17/18 9:26:00 PM |
|---|---|---|

Font color: Text 1

| Page 13: [253] Formatted | Sun ji | 6/11/18 1:55:00 PM |
|---|---|---|

Font: 10 pt, English (US)

| Page 13: [253] Formatted | Sun ji | 6/11/18 1:55:00 PM |
|---|---|---|

Font: 10 pt, English (US)

| Page 13: [254] Formatted | Sun ji | 6/11/18 1:55:00 PM |
|---|---|---|

Font: 10 pt, English (US)

| Page 13: [254] Formatted | Sun ji | 6/11/18 1:55:00 PM |
|---|---|---|

Font: 10 pt, English (US)

| Page 13: [254] Formatted | Sun ji | 6/11/18 1:55:00 PM |
|---|---|---|

Font: 10 pt, English (US)

| Page 13: [255] Formatted | Sun ji | 6/11/18 1:55:00 PM |
|---|---|---|

English (US)

| Page 13: [255] Formatted | Sun ji | 6/11/18 1:55:00 PM |
|---|---|---|

English (US)

| Page 13: [255] Formatted | Sun ji | 6/11/18 1:55:00 PM |
|---|---|---|

English (US)

| Page 13: [256] Formatted | Sun ji | 6/11/18 1:55:00 PM |
|---|---|---|

Font: 10 pt, English (US)

| Page 13: [256] Formatted | Sun ji | 6/11/18 1:55:00 PM |
|---|---|---|

Font: 10 pt, English (US)

| Page 13: [257] Formatted | Sun ji | 6/11/18 1:55:00 PM |
|---|---|---|

Font: 10 pt, English (US)

| Page 13: [257] Formatted | Sun ji | 6/11/18 1:55:00 PM |
|---|---|---|

Font: 10 pt, English (US)

| Page 13: [257] Formatted | Sun ji | 6/11/18 1:55:00 PM |
|---|---|---|

Font: 10 pt, English (US)

| Page 13: [257] Formatted | Sun ji | 6/11/18 1:55:00 PM |
|---|---|---|

Font: 10 pt, English (US)

| Page 14: [258] Formatted | Sun ji | 6/11/18 1:55:00 PM |
|---|---|---|

Font: 10 pt, English (US)

| Page 14: [259] Formatted | Sun ji | 6/11/18 1:55:00 PM |
|---|---|---|

Font: 10 pt, English (US)

| Page 14: [260] Formatted | Sun ji | 6/11/18 1:55:00 PM |
|---|---|---|

Font: 10 pt, English (US)

| Page 14: [260] Formatted | Sun ji | 6/11/18 1:55:00 PM |
|---|---|---|

Font: 10 pt, English (US)

| Page 14: [260] Formatted | Sun ji | 6/11/18 1:55:00 PM |
|---|---|---|

Font: 10 pt, English (US)

| Page 14: [261] Deleted | Sun ji | 8/2/18 2:05:00 PM |
|---|---|---|

ground

| Page 14: [261] Deleted | Sun ji | 8/2/18 2:05:00 PM |
|---|---|---|

ground

| Page 14: [262] Formatted | Sun ji | 6/11/18 1:55:00 PM |
|---|---|---|

Font: 10 pt, English (US)

| Page 14: [262] Formatted | Sun ji | 6/11/18 1:55:00 PM |
|---|---|---|

Font: 10 pt, English (US)

| Page 14: [263] Formatted | Sun ji | 6/11/18 1:55:00 PM |
|---|---|---|

English (US)

| Page 14: [263] Formatted | Sun ji | 6/11/18 1:55:00 PM |
|---|---|---|

English (US)

| Page 14: [264] Formatted | Sun ji | 6/11/18 1:55:00 PM |
|---|---|---|

English (US)

| Page 14: [264] Formatted | Sun ji | 6/11/18 1:55:00 PM |
|---|---|---|

English (US)

| Page 14: [265] Deleted | Sun ji | 5/22/18 5:29:00 PM |
|---|---|---|

is relatively small and the sample distribution

| Page 14: [265] Deleted | Sun ji | 5/22/18 5:29:00 PM |
|---|---|---|

is relatively small and the sample distribution

| Page 14: [266] Formatted | Sun ji | 6/11/18 1:55:00 PM |
|---|---|---|

English (US)

| Page 14: [266] Formatted | Sun ji | 6/11/18 1:55:00 PM |
|---|---|---|

English (US)

| Page 14: [267] Formatted | Sun ji | 6/11/18 1:55:00 PM |
|---|---|---|

Font: 10 pt, English (US)

| Page 14: [267] Formatted | Sun ji | 6/11/18 1:55:00 PM |
|---|---|---|

Font: 10 pt, English (US)

| Page 14: [267] Formatted | Sun ji | 6/11/18 1:55:00 PM |
|---|---|---|

Font: 10 pt, English (US)

| Page 14: [267] Formatted | Sun ji | 6/11/18 1:55:00 PM |
| --- | --- | --- |

Font: 10 pt, English (US)

| Page 14: [268] Formatted | Sun ji | 6/11/18 1:55:00 PM |
| --- | --- | --- |

Font: 5.5 pt

| Page 14: [269] Commented [Sj19R18] | Sun ji | 5/15/18 7:11:00 PM |
| --- | --- | --- |

It means the number of observations is limited so that the dataset is sensitive to outliers. I rephrased this sentence.

| Page 14: [270] Formatted | Sun ji | 6/11/18 1:55:00 PM |
| --- | --- | --- |

Font: 10 pt, English (US)

| Page 14: [271] Formatted | Sun ji | 6/11/18 1:55:00 PM |
| --- | --- | --- |

English (US)

| Page 14: [271] Formatted | Sun ji | 6/11/18 1:55:00 PM |
| --- | --- | --- |

English (US)

| Page 14: [271] Formatted | Sun ji | 6/11/18 1:55:00 PM |
| --- | --- | --- |

English (US)

| Page 14: [272] Formatted | Sun ji | 6/11/18 1:55:00 PM |
| --- | --- | --- |

Font: 10 pt, English (US)

| Page 14: [272] Formatted | Sun ji | 6/11/18 1:55:00 PM |
| --- | --- | --- |

Font: 10 pt, English (US)

| Page 14: [273] Commented [JV20] | JP Veefkind | 5/14/18 5:19:00 PM |
| --- | --- | --- |

We first fit, then apply the outlier, then we fit again

| Page 14: [274] Formatted | Sun ji | 6/11/18 1:55:00 PM |
| --- | --- | --- |

Font: 5.5 pt

| Page 14: [275] Commented [Sj21R20] | Sun ji | 5/16/18 11:41:00 AM |
| --- | --- | --- |

I implied that by the scatter plots in Fig.8, but I did not provide the fitting results (AAI, SSA, etc.) calculated by the whole data (including outliers).

| Page 14: [276] Formatted | Sun ji | 6/11/18 1:55:00 PM |
| --- | --- | --- |

Font: 10 pt, English (US)

| Page 14: [277] Formatted | Sun ji | 6/11/18 1:55:00 PM |
| --- | --- | --- |

Font: 10 pt, English (US)

| Page 14: [277] Formatted | Sun ji | 6/11/18 1:55:00 PM |
| --- | --- | --- |

Font: 10 pt, English (US)

| Page 14: [277] Formatted | Sun ji | 6/11/18 1:55:00 PM |
|---|---|---|

Font: 10 pt, English (US)

| Page 14: [277] Formatted | Sun ji | 6/11/18 1:55:00 PM |
|---|---|---|

Font: 10 pt, English (US)

| Page 14: [277] Formatted | Sun ji | 6/11/18 1:55:00 PM |
|---|---|---|

Font: 10 pt, English (US)

| Page 14: [277] Formatted | Sun ji | 6/11/18 1:55:00 PM |
|---|---|---|

Font: 10 pt, English (US)

| Page 14: [277] Formatted | Sun ji | 6/11/18 1:55:00 PM |
|---|---|---|

Font: 10 pt, English (US)

| Page 14: [277] Formatted | Sun ji | 6/11/18 1:55:00 PM |
|---|---|---|

Font: 10 pt, English (US)

| Page 14: [277] Formatted | Sun ji | 6/11/18 1:55:00 PM |
|---|---|---|

Font: 10 pt, English (US)

| Page 14: [277] Formatted | Sun ji | 6/11/18 1:55:00 PM |
|---|---|---|

Font: 10 pt, English (US)

| Page 14: [277] Formatted | Sun ji | 6/11/18 1:55:00 PM |
|---|---|---|

Font: 10 pt, English (US)

| Page 14: [277] Formatted | Sun ji | 6/11/18 1:55:00 PM |
|---|---|---|

Font: 10 pt, English (US)

| Page 14: [278] Formatted | Sun ji | 6/11/18 1:55:00 PM |
|---|---|---|

Font: 10 pt, English (US)

| Page 14: [279] Deleted | Sun ji | 8/10/18 3:47:00 PM |
|---|---|---|

on MetOp-A/B

| Page 14: [279] Deleted | Sun ji | 8/10/18 3:47:00 PM |
|---|---|---|

on MetOp-A/B

| Page 14: [280] Formatted | Sun ji | 6/11/18 1:55:00 PM |
|---|---|---|

English (US)

| Page 14: [280] Formatted | Sun ji | 6/11/18 1:55:00 PM |
|---|---|---|

English (US)

| Page 14: [281] Formatted | Sun ji | 6/11/18 1:55:00 PM |

Font: 10 pt, English (US)

| Page 14: [282] Formatted | Sun ji | 6/11/18 1:55:00 PM |

English (US)

| Page 14: [282] Formatted | Sun ji | 6/11/18 1:55:00 PM |

English (US)

| Page 14: [283] Formatted | Sun ji | 6/11/18 1:55:00 PM |

English (US)

| Page 14: [283] Formatted | Sun ji | 6/11/18 1:55:00 PM |

English (US)

| Page 14: [284] Formatted | Sun ji | 6/11/18 1:55:00 PM |

Font: 10 pt, English (US)

| Page 14: [284] Formatted | Sun ji | 6/11/18 1:55:00 PM |

Font: 10 pt, English (US)

| Page 14: [285] Formatted | Sun ji | 6/11/18 1:55:00 PM |

Font: 10 pt, English (US)

| Page 14: [285] Formatted | Sun ji | 6/11/18 1:55:00 PM |

Font: 10 pt, English (US)

| Page 14: [286] Formatted | Sun ji | 6/11/18 1:55:00 PM |

Font: 10 pt, English (US)

| Page 14: [287] Deleted | JP Veefkind | 5/14/18 5:31:00 PM |

from

| Page 14: [287] Deleted | JP Veefkind | 5/14/18 5:31:00 PM |

from

| Page 14: [287] Deleted | JP Veefkind | 5/14/18 5:31:00 PM |

from

| Page 14: [287] Deleted | JP Veefkind | 5/14/18 5:31:00 PM |

from

| Page 14: [287] Deleted | JP Veefkind | 5/14/18 5:31:00 PM |

from

| Page 14: [288] Formatted | Sun ji | 6/11/18 1:55:00 PM |

English (US)

| Page 14: [288] Formatted | Sun ji | 6/11/18 1:55:00 PM |

English (US)

| Page 14: [288] Formatted | Sun ji | 6/11/18 1:55:00 PM |

English (US)

| Page 14: [288] Formatted | Sun ji | 6/11/18 1:55:00 PM |

English (US)

| Page 14: [289] Formatted | Sun ji | 6/11/18 1:55:00 PM |

Font: 10 pt, English (US)

| Page 14: [290] Deleted | Sun ji | 6/12/18 8:57:00 AM |

probably did not always measure the plume feature, and may even fail to capture the elevated

| Page 14: [291] Formatted | Sun ji | 6/11/18 1:55:00 PM |

Font: 10 pt, English (US)

| Page 14: [292] Formatted | Sun ji | 6/11/18 1:55:00 PM |

Font: 10 pt, English (US)

| Page 14: [293] Formatted | Sun ji | 6/11/18 1:55:00 PM |

Font: 10 pt, English (US)

| Page 15: [294] Formatted | Sun ji | 6/11/18 1:55:00 PM |

Font: 10 pt, English (US)

| Page 15: [294] Formatted | Sun ji | 6/11/18 1:55:00 PM |

Font: 10 pt, English (US)

| Page 15: [295] Formatted | Sun ji | 6/11/18 1:55:00 PM |

Font: 10 pt, English (US)

| Page 15: [296] Formatted | Sun ji | 6/11/18 1:55:00 PM |

Font: 10 pt, English (US)

| Page 15: [297] Deleted | Sun ji | 8/3/18 11:31:00 AM |

low

| Page 15: [297] Deleted | Sun ji | 8/3/18 11:31:00 AM |

low

| Page 15: [298] Formatted | Sun ji | 6/11/18 1:55:00 PM |

Font: 10 pt, English (US)

| Page 15: [299] Formatted | Sun ji | 6/11/18 1:55:00 PM |

Font: 10 pt, English (US)

| Page 15: [300] Formatted | Sun ji | 6/11/18 1:55:00 PM |

English (US)

| Page 15: [301] Deleted | Sun ji | 5/25/18 12:25:00 PM |

, which reflects is in correspondence with the smoke ageing process (Reid et al., 2004)

| Page 15: [302] Formatted | Sun ji | 6/11/18 1:55:00 PM |

Font: 10 pt, English (US)

| Page 15: [303] Formatted | Sun ji | 6/11/18 1:55:00 PM |

Font: 10 pt, English (US)

| Page 15: [304] Formatted | Sun ji | 6/11/18 1:55:00 PM |

Font: 10 pt, English (US)

| Page 15: [304] Formatted | Sun ji | 6/11/18 1:55:00 PM |

Font: 10 pt, English (US)

| Page 15: [305] Formatted | Sun ji | 6/11/18 1:55:00 PM |

Font: 10 pt, English (US)

| Page 15: [306] Formatted | Sun ji | 6/11/18 1:55:00 PM |

Font: 10 pt, English (US)

| Page 15: [306] Formatted | Sun ji | 6/11/18 1:55:00 PM |

Font: 10 pt, English (US)

| Page 15: [307] Formatted | Sun ji | 6/17/18 5:26:00 PM |

Font color: Text 1

| Page 15: [307] Formatted | Sun ji | 6/17/18 5:26:00 PM |

Font color: Text 1

| Page 15: [307] Formatted | Sun ji | 6/17/18 5:26:00 PM |

Font color: Text 1

| Page 15: [307] Formatted | Sun ji | 6/17/18 5:26:00 PM |

Font color: Text 1

| Page 15: [307] Formatted | Sun ji | 6/17/18 5:26:00 PM |

Font color: Text 1

| Page 15: [307] Formatted | Sun ji | 6/17/18 5:26:00 PM |
|---|---|---|

Font color: Text 1

| Page 15: [307] Formatted | Sun ji | 6/17/18 5:26:00 PM |
|---|---|---|

Font color: Text 1

| Page 15: [307] Formatted | Sun ji | 6/17/18 5:26:00 PM |
|---|---|---|

Font color: Text 1

| Page 15: [307] Formatted | Sun ji | 6/17/18 5:26:00 PM |
|---|---|---|

Font color: Text 1

| Page 15: [307] Formatted | Sun ji | 6/17/18 5:26:00 PM |
|---|---|---|

Font color: Text 1

| Page 15: [307] Formatted | Sun ji | 6/17/18 5:26:00 PM |
|---|---|---|

Font color: Text 1

| Page 15: [308] Formatted | Sun ji | 6/17/18 5:26:00 PM |
|---|---|---|

Font color: Text 1

| Page 15: [308] Formatted | Sun ji | 6/17/18 5:26:00 PM |
|---|---|---|

Font color: Text 1

| Page 15: [308] Formatted | Sun ji | 6/17/18 5:26:00 PM |
|---|---|---|

Font color: Text 1

| Page 15: [308] Formatted | Sun ji | 6/17/18 5:26:00 PM |
|---|---|---|

Font color: Text 1

| Page 15: [309] Deleted | Sun ji | 6/12/18 9:06:00 AM |
|---|---|---|

. This difference might be due to the fact that the selected AERONET site is not exactly at the primary biomass burning regions as mentioned in section 3.1.3. The $\omega_0$ measured by AERONET could increase as the result of aerosol ageing. Specifically, the location of the AERONET site is in the downtown, where the more reflective urban or industrial aerosols may mix with the smoke and enhance the measured $\omega_0$. Besides

| Page 15: [310] Formatted | Sun ji | 6/11/18 1:55:00 PM |
|---|---|---|

Font: 10 pt, English (US)

| Page 15: [310] Formatted | Sun ji | 6/11/18 1:55:00 PM |
|---|---|---|

Font: 10 pt, English (US)

| Page 15: [311] Formatted | Sun ji | 6/11/18 1:55:00 PM |
|---|---|---|

English (US), Not Highlight

| Page 15: [311] Formatted | Sun ji | 6/11/18 1:55:00 PM |
|---|---|---|

English (US), Not Highlight

| Page 15: [312] Deleted | Sun ji | 6/12/18 10:09:00 AM |
|---|---|---|

micro-physics parameters retrieved from

| Page 15: [312] Deleted | Sun ji | 6/12/18 10:09:00 AM |
|---|---|---|

micro-physics parameters retrieved from

| Page 15: [313] Formatted | Sun ji | 6/11/18 1:55:00 PM |
|---|---|---|

Font: 10 pt, English (US)

| Page 15: [314] Formatted | Sun ji | 6/11/18 1:55:00 PM |
|---|---|---|

Font: 10 pt, English (US)

| Page 15: [315] Commented [JV22] | JP Veefkind | 5/14/18 5:56:00 PM |
|---|---|---|

Should these be left in? The AERONET product is much better established…

| Page 15: [316] Formatted | Sun ji | 6/11/18 1:55:00 PM |
|---|---|---|

Font: 5.5 pt

| Page 15: [317] Commented [Sj23R22] | Sun ji | 5/15/18 10:45:00 PM |
|---|---|---|

Just mention that AERONET measurements is not perfect also. The input size distribution functions may not be the major error source in our case, but the refractive index is not, which may bias the forward simulation. The SSA, as the parameter to validate, is also with uncertainty of 0.03.

| Page 15: [318] Formatted | Sun ji | 6/11/18 1:55:00 PM |
|---|---|---|

Font: 10 pt, English (US)

| Page 15: [318] Formatted | Sun ji | 6/11/18 1:55:00 PM |
|---|---|---|

Font: 10 pt, English (US)

| Page 15: [318] Formatted | Sun ji | 6/11/18 1:55:00 PM |
|---|---|---|

Font: 10 pt, English (US)

| Page 15: [319] Deleted | Sun ji | 6/11/18 7:22:00 PM |
|---|---|---|

i

| Page 15: [319] Deleted | Sun ji | 6/11/18 7:22:00 PM |
|---|---|---|

i

| Page 15: [320] Formatted | Sun ji | 6/11/18 1:55:00 PM |
|---|---|---|

Font: 10 pt, English (US)

| Page 15: [320] Formatted | Sun ji | 6/11/18 1:55:00 PM |
|---|---|---|

Font: 10 pt, English (US)

| Page 15: [320] Formatted | Sun ji | 6/11/18 1:55:00 PM |
|---|---|---|

Font: 10 pt, English (US)

| Page 15: [321] Formatted | Sun ji | 8/10/18 5:24:00 PM |
|---|---|---|

Font color: Text 1

| Page 15: [321] Formatted | Sun ji | 8/10/18 5:24:00 PM |
|---|---|---|

Font color: Text 1

| Page 15: [322] Formatted | Sun ji | 6/12/18 10:30:00 AM |
|---|---|---|

Font: 10 pt, English (US), Highlight

| Page 15: [322] Formatted | Sun ji | 6/12/18 10:30:00 AM |
|---|---|---|

Font: 10 pt, English (US), Highlight

| Page 15: [323] Formatted | Sun ji | 6/11/18 1:55:00 PM |
|---|---|---|

Font: 10 pt, English (US)

| Page 15: [323] Formatted | Sun ji | 6/11/18 1:55:00 PM |
|---|---|---|

Font: 10 pt, English (US)

| Page 15: [324] Formatted | Sun ji | 6/11/18 1:55:00 PM |
|---|---|---|

Font: 10 pt, English (US)

| Page 15: [325] Formatted | Sun ji | 6/11/18 1:55:00 PM |
|---|---|---|

English (US)

| Page 15: [326] Formatted | Sun ji | 6/11/18 1:55:00 PM |
|---|---|---|

English (US)

| Page 15: [326] Formatted | Sun ji | 6/11/18 1:55:00 PM |
|---|---|---|

English (US)

| Page 15: [327] Deleted | Sun ji | 5/25/18 12:06:00 PM |
|---|---|---|

s

| Page 15: [327] Deleted | Sun ji | 5/25/18 12:06:00 PM |
|---|---|---|

s

| Page 15: [328] Formatted | Sun ji | 6/11/18 1:55:00 PM |
|---|---|---|

Font: 10 pt, English (US)

| Page 16: [329] Formatted | Sun ji | 6/11/18 1:55:00 PM |
|---|---|---|

Font: 10 pt, English (US)

| Page 16: [330] Formatted | Sun ji | 6/11/18 1:55:00 PM |
|---|---|---|

Font: 10 pt, English (US)

| Page 16: [331] Formatted | Sun ji | 6/11/18 1:55:00 PM |
|---|---|---|

Font: 10 pt, English (US)

| Page 16: [332] Formatted | Sun ji | 6/11/18 1:55:00 PM |
|---|---|---|

Font: 10 pt, English (US)

| Page 16: [332] Formatted | Sun ji | 6/11/18 1:55:00 PM |
|---|---|---|

Font: 10 pt, English (US)

| Page 16: [333] Formatted | Sun ji | 6/11/18 1:55:00 PM |
|---|---|---|

Font: 10 pt, English (US)

| Page 16: [334] Formatted | Sun ji | 6/11/18 1:55:00 PM |
|---|---|---|

Font: 10 pt, English (US)

| Page 16: [335] Formatted | Sun ji | 6/11/18 1:55:00 PM |
|---|---|---|

English (US)

| Page 16: [335] Formatted | Sun ji | 6/11/18 1:55:00 PM |
|---|---|---|

English (US)

| Page 16: [336] Deleted | Sun ji | 5/16/18 4:43:00 PM |
|---|---|---|

for the

| Page 16: [336] Deleted | Sun ji | 5/16/18 4:43:00 PM |
|---|---|---|

for the

| Page 16: [337] Formatted | Sun ji | 6/11/18 1:55:00 PM |
|---|---|---|

English (US)

| Page 16: [338] Formatted | Sun ji | 6/11/18 1:55:00 PM |
|---|---|---|

Font: 10 pt, English (US)

| Page 16: [339] Formatted | Sun ji | 6/11/18 1:55:00 PM |
|---|---|---|

Font: 10 pt, English (US)

| Page 16: [340] Deleted | JP Veefkind | 5/14/18 6:01:00 PM |
|---|---|---|

However, even with relative reasonable retrieval of $z_{aer}$, it is noted that the $\omega_0$ retrieved on 27 January is significantly underestimated and biased from the mean level of other cases. This implies the existence of other error sources, such as the observational errors from the input $\tau$ of MODIS and the AAI of OMI to be fit.

| Page 16: [341] Formatted | Sun ji | 6/11/18 1:55:00 PM |
|---|---|---|

Font: 10 pt, English (US)

| Page 16: [342] Formatted | Sun ji | 6/11/18 1:55:00 PM |
|---|---|---|

Font: 10 pt, English (US)

| Page 16: [342] Formatted | Sun ji | 6/11/18 1:55:00 PM |
|---|---|---|

Font: 10 pt, English (US)

| Page 16: [342] Formatted | Sun ji | 6/11/18 1:55:00 PM |
|---|---|---|

Font: 10 pt, English (US)

| Page 16: [343] Deleted | Sun ji | 5/25/18 12:21:00 PM |
|---|---|---|

However

| Page 16: [343] Deleted | Sun ji | 5/25/18 12:21:00 PM |
|---|---|---|

However

| Page 16: [343] Deleted | Sun ji | 5/25/18 12:21:00 PM |
|---|---|---|

However

| Page 16: [343] Deleted | Sun ji | 5/25/18 12:21:00 PM |
|---|---|---|

However

| Page 16: [344] Formatted | Sun ji | 6/11/18 1:55:00 PM |
|---|---|---|

English (US)

| Page 16: [345] Formatted | Sun ji | 6/11/18 1:55:00 PM |
|---|---|---|

English (US)

| Page 16: [346] Deleted | Sun ji | 5/25/18 12:22:00 PM |
|---|---|---|

. E

| Page 16: [346] Deleted | Sun ji | 5/25/18 12:22:00 PM |
|---|---|---|

. E

| Page 16: [347] Formatted | Sun ji | 6/11/18 1:55:00 PM |
|---|---|---|

Font: 10 pt, English (US)

| Page 16: [347] Formatted | Sun ji | 6/11/18 1:55:00 PM |
|---|---|---|

Font: 10 pt, English (US)

| Page 16: [348] Formatted | Sun ji | 6/11/18 1:55:00 PM |

Font: 10 pt, English (US)

| Page 16: [348] Formatted | Sun ji | 6/11/18 1:55:00 PM |

Font: 10 pt, English (US)

| Page 16: [349] Formatted | Sun ji | 6/11/18 1:55:00 PM |

Font: 10 pt, English (US)

| Page 16: [350] Formatted | Sun ji | 6/11/18 1:55:00 PM |

Font: 10 pt, English (US)

| Page 16: [351] Formatted | Sun ji | 6/11/18 1:55:00 PM |

Font: 10 pt, English (US)

| Page 16: [352] Formatted | Sun ji | 6/11/18 1:55:00 PM |

Font: 10 pt, (Asian) Chinese (China), (Other) English (US)

| Page 16: [353] Formatted | Sun ji | 6/11/18 1:55:00 PM |

Font: 10 pt, English (US)

| Page 16: [354] Formatted | Sun ji | 6/11/18 1:55:00 PM |

English (US)

| Page 16: [354] Formatted | Sun ji | 6/11/18 1:55:00 PM |

English (US)

| Page 16: [355] Formatted | Sun ji | 6/11/18 1:55:00 PM |

English (US)

| Page 16: [355] Formatted | Sun ji | 6/11/18 1:55:00 PM |

English (US)

| Page 16: [356] Formatted | Sun ji | 6/11/18 1:55:00 PM |

Font: 10 pt, English (US)

| Page 16: [357] Formatted | Sun ji | 6/11/18 1:55:00 PM |

English (US)

| Page 16: [358] Formatted | Sun ji | 6/11/18 1:55:00 PM |

Font: 10 pt, English (US)

| Page 16: [358] Formatted | Sun ji | 6/11/18 1:55:00 PM |

Font: 10 pt, English (US)

| Page 16: [358] Formatted | Sun ji | 6/11/18 1:55:00 PM |

Font: 10 pt, English (US)

| Page 16: [359] Formatted | Sun ji | 6/11/18 1:55:00 PM |

English (US)

| Page 16: [359] Formatted | Sun ji | 6/11/18 1:55:00 PM |

English (US)

| Page 16: [360] Formatted | Sun ji | 6/11/18 1:55:00 PM |

Font: 10 pt, English (US)

| Page 16: [361] Formatted | Sun ji | 6/11/18 1:55:00 PM |

Font: 10 pt, English (US)

| Page 16: [362] Formatted | Sun ji | 6/11/18 1:55:00 PM |

Font: 10 pt, English (US)

| Page 16: [363] Formatted | Sun ji | 6/11/18 1:55:00 PM |

English (US)

| Page 16: [364] Formatted | Sun ji | 6/11/18 1:55:00 PM |

Font: 10 pt, English (US)

| Page 16: [365] Formatted | Sun ji | 6/11/18 1:55:00 PM |

English (US)

| Page 16: [365] Formatted | Sun ji | 6/11/18 1:55:00 PM |

English (US)

| Page 16: [366] Formatted | Sun ji | 6/11/18 1:55:00 PM |

English (US)

| Page 16: [366] Formatted | Sun ji | 6/11/18 1:55:00 PM |

English (US)

| Page 16: [366] Formatted | Sun ji | 6/11/18 1:55:00 PM |

English (US)

| Page 16: [367] Deleted | Sun ji | 6/12/18 10:52:00 AM |

 proves its usefulness and effectiveness

| Page 16: [367] Deleted | Sun ji | 6/12/18 10:52:00 AM |

 proves its usefulness and effectiveness

| Page 16: [368] Formatted | Sun ji | 6/11/18 1:55:00 PM |
|---|---|---|

Font: 10 pt, English (US)

| Page 16: [369] Formatted | Sun ji | 6/11/18 1:55:00 PM |
|---|---|---|

English (US)

| Page 16: [369] Formatted | Sun ji | 6/11/18 1:55:00 PM |
|---|---|---|

English (US)

| Page 16: [370] Deleted | Sun ji | 8/2/18 4:15:00 PM |
|---|---|---|

The retrieved $\omega_0$ is reasonable, taking into account the typical uncertainty in the $\omega_0$ retrieved from AERONET ($\pm 0.03$).

| Page 16: [371] Formatted | Sun ji | 6/11/18 1:55:00 PM |
|---|---|---|

Font: 10 pt, English (US)

| Page 16: [372] Formatted | Sun ji | 6/11/18 1:55:00 PM |
|---|---|---|

Font: 10 pt, English (US)

| Page 16: [373] Formatted | Sun ji | 6/11/18 1:55:00 PM |
|---|---|---|

Font: 10 pt, English (US)

| Page 16: [374] Deleted | Sun ji | 6/12/18 11:00:00 AM |
|---|---|---|

the assumption of homogeneous and static plume properties, which ignores the plume evolution over space and time;

| Page 16: [375] Formatted | Sun ji | 6/11/18 1:55:00 PM |
|---|---|---|

Font: 10 pt, English (US)

| Page 16: [375] Formatted | Sun ji | 6/11/18 1:55:00 PM |
|---|---|---|

Font: 10 pt, English (US)

| Page 17: [376] Deleted | Sun ji | 6/17/18 5:37:00 PM |
|---|---|---|

This study proves the potential of utilizing OMI measured AAI to quantitatively characterize aerosol optical properties like $\omega_0$. Even without direct observation of aerosol profiles, this parameter can also be retrieved with quite good confidence. However, apart from the observational uncertainties, the current study is probably somewhat limited by the necessary assumptions of homogeneous and static plume properties, whose impact on retrieved $\omega_0$ is difficult to quantify. In the future planned

| Page 17: [377] Deleted | Sun ji | 6/12/18 11:45:00 AM |
|---|---|---|

work, a chemistry transport model is needed to describe the evolution of the plume properties in space and time. Moreover, also clouds should be taken into consideration in order to use the AAI observations over clouds, thus making the maximum use of the near-UV observations.

| Page 17: [378] Formatted | Sun ji | 8/3/18 9:17:00 AM |

Font: Not Bold, English (US), Do not check spelling or grammar

| Page 17: [379] Deleted | Sun ji | 4/23/18 2:30:00 PM |

Adler, G., Flores, J. M., Abo Riziq, A., Borrmann, S. and Rudich, Y.: Chemical, physical, and optical evolution of biomass burning aerosols: A case study, Atmos. Chem. Phys., 11, 1491–1503, doi:10.5194/acp-11-1491-2011, 2011. Bäumer, D., Vogel, B., Versick, S., Rinke, R., Möhler, O. and Schnaiter, M.: Relationship of visibility, aerosol optical thicknss and aerosol size distribution in an ageing air mass over South-West Germany, Atmos. Environ., 42, 989–998, doi:10.1016/j.atmosenv.2007.10.017, 2008.

| Page 17: [380] Formatted | Sun ji | 8/7/18 10:25:00 AM |

Font: Not Bold, Font color: Auto, English (US), Do not check spelling or grammar, Not Highlight

| Page 31: [381] Formatted | Sun ji | 6/11/18 1:55:00 PM |

English (US)

| Page 31: [382] Formatted | Sun ji | 6/11/18 1:55:00 PM |

English (US)

| Page 31: [383] Formatted | Sun ji | 6/11/18 1:55:00 PM |

Font: 9 pt, English (US)

| Page 31: [384] Formatted | Sun ji | 5/15/18 10:32:00 PM |

Left

| Page 31: [385] Formatted | Sun ji | 6/11/18 1:55:00 PM |

Font: 7.5 pt, English (US)

| Page 31: [386] Formatted | Sun ji | 6/11/18 1:55:00 PM |

Font: 9 pt, English (US)

| Page 31: [386] Formatted | Sun ji | 6/11/18 1:55:00 PM |

Font: 9 pt, English (US)

| Page 31: [387] Formatted | Sun ji | 6/11/18 1:55:00 PM |

Font: 9 pt, English (US)

| Page 31: [387] Formatted | Sun ji | 6/11/18 1:55:00 PM |

Font: 9 pt, English (US)

| Page 31: [388] Formatted | Sun ji | 6/11/18 1:55:00 PM |

Font: 9 pt, English (US)

| Page 31: [388] Formatted | Sun ji | 6/11/18 1:55:00 PM |

Font: 9 pt, English (US)

| Page 31: [389] Formatted | Sun ji | 6/11/18 1:55:00 PM |
|---|---|---|

Font: 9 pt, English (US)

| Page 31: [390] Formatted | Sun ji | 5/15/18 10:32:00 PM |
|---|---|---|

Left

| Page 31: [391] Formatted | Sun ji | 6/11/18 1:55:00 PM |
|---|---|---|

Font: 7.5 pt, English (US)

| Page 31: [392] Formatted | Sun ji | 6/11/18 1:55:00 PM |
|---|---|---|

Font: 9 pt, English (US)

| Page 31: [392] Formatted | Sun ji | 6/11/18 1:55:00 PM |
|---|---|---|

Font: 9 pt, English (US)

| Page 31: [393] Formatted | Sun ji | 6/11/18 1:55:00 PM |
|---|---|---|

Font: 9 pt, English (US)

| Page 31: [393] Formatted | Sun ji | 6/11/18 1:55:00 PM |
|---|---|---|

Font: 9 pt, English (US)

| Page 31: [394] Formatted | Sun ji | 6/11/18 1:55:00 PM |
|---|---|---|

Font: 9 pt, English (US)

| Page 31: [394] Formatted | Sun ji | 6/11/18 1:55:00 PM |
|---|---|---|

Font: 9 pt, English (US)

| Page 31: [395] Formatted | Sun ji | 6/11/18 1:55:00 PM |
|---|---|---|

Font: 9 pt, English (US)

| Page 31: [396] Formatted | Sun ji | 5/15/18 10:32:00 PM |
|---|---|---|

Left

| Page 31: [397] Formatted | Sun ji | 6/11/18 1:55:00 PM |
|---|---|---|

Font: 7.5 pt, English (US)

| Page 31: [398] Formatted | Sun ji | 6/11/18 1:55:00 PM |
|---|---|---|

Font: 9 pt, English (US)

| Page 31: [398] Formatted | Sun ji | 6/11/18 1:55:00 PM |
|---|---|---|

Font: 9 pt, English (US)

| Page 31: [399] Formatted | Sun ji | 6/11/18 1:55:00 PM |
|---|---|---|

Font: 9 pt, English (US)

| Page 31: [399] Formatted | Sun ji | 6/11/18 1:55:00 PM |
|---|---|---|

Font: 9 pt, English (US)

| Page 31: [400] Formatted | Sun ji | 6/11/18 1:55:00 PM |
|---|---|---|

Font: 9 pt, English (US)

| Page 31: [400] Formatted | Sun ji | 6/11/18 1:55:00 PM |
|---|---|---|

Font: 9 pt, English (US)

| Page 31: [401] Formatted | Sun ji | 6/11/18 1:55:00 PM |
|---|---|---|

Font: 9 pt, English (US)

| Page 31: [402] Formatted | Sun ji | 5/15/18 10:32:00 PM |
|---|---|---|

Left

| Page 31: [403] Formatted | Sun ji | 6/11/18 1:55:00 PM |
|---|---|---|

Font: 7.5 pt, English (US)

| Page 31: [404] Formatted | Sun ji | 6/11/18 1:55:00 PM |
|---|---|---|

Font: 9 pt, English (US)

| Page 31: [404] Formatted | Sun ji | 6/11/18 1:55:00 PM |
|---|---|---|

Font: 9 pt, English (US)

| Page 31: [405] Formatted | Sun ji | 6/11/18 1:55:00 PM |
|---|---|---|

Font: 9 pt, English (US)

| Page 31: [405] Formatted | Sun ji | 6/11/18 1:55:00 PM |
|---|---|---|

Font: 9 pt, English (US)

| Page 31: [406] Formatted | Sun ji | 6/11/18 1:55:00 PM |
|---|---|---|

Font: 9 pt, English (US)

| Page 31: [406] Formatted | Sun ji | 6/11/18 1:55:00 PM |
|---|---|---|

Font: 9 pt, English (US)

| Page 31: [407] Formatted | Sun ji | 6/11/18 1:55:00 PM |
|---|---|---|

Font: 9 pt, English (US)

| Page 31: [408] Formatted | Sun ji | 5/15/18 10:32:00 PM |
|---|---|---|

Left

| Page 31: [409] Formatted | Sun ji | 6/11/18 1:55:00 PM |
|---|---|---|

Font: 7.5 pt, English (US)

| Page 31: [410] Formatted | Sun ji | 6/11/18 1:55:00 PM |
|---|---|---|

Font: 9 pt, English (US)

| Page 31: [410] Formatted | Sun ji | 6/11/18 1:55:00 PM |
|---|---|---|

Font: 9 pt, English (US)

| Page 31: [411] Formatted | Sun ji | 6/11/18 1:55:00 PM |
|---|---|---|

Font: 9 pt, English (US)

| Page 31: [411] Formatted | Sun ji | 6/11/18 1:55:00 PM |
|---|---|---|

Font: 9 pt, English (US)

| Page 31: [412] Formatted | Sun ji | 6/11/18 1:55:00 PM |
|---|---|---|

Font: 9 pt, English (US)

| Page 31: [412] Formatted | Sun ji | 6/11/18 1:55:00 PM |
|---|---|---|

Font: 9 pt, English (US)

| Page 31: [413] Formatted | Sun ji | 6/11/18 1:55:00 PM |
|---|---|---|

Font: 9 pt, English (US)

| Page 31: [414] Formatted | Sun ji | 5/15/18 10:32:00 PM |
|---|---|---|

Left

| Page 31: [415] Formatted | Sun ji | 6/11/18 1:55:00 PM |
|---|---|---|

Font: 9 pt, English (US)

| Page 31: [416] Formatted | Sun ji | 5/15/18 10:32:00 PM |
|---|---|---|

Left

| Page 31: [417] Formatted | Sun ji | 6/11/18 1:55:00 PM |
|---|---|---|

Font: 9 pt, English (US)

| Page 31: [418] Formatted | Sun ji | 6/11/18 1:55:00 PM |
|---|---|---|

Font: 9 pt, English (US)

| Page 31: [419] Formatted | Sun ji | 5/15/18 10:32:00 PM |
|---|---|---|

Left

| Page 31: [420] Formatted Table | ji Sun | 2/7/18 10:45:00 AM |
|---|---|---|

Formatted Table

| Page 31: [421] Formatted | Sun ji | 6/11/18 1:55:00 PM |
|---|---|---|

Font: 7.5 pt, English (US)

| Page 31: [422] Formatted | Sun ji | 6/11/18 1:55:00 PM |
|---|---|---|

Font: 9 pt, English (US)

| Page 31: [422] Formatted | Sun ji | 6/11/18 1:55:00 PM |
|---|---|---|

Font: 9 pt, English (US)

| Page 31: [423] Formatted | Sun ji | 6/11/18 1:55:00 PM |
|---|---|---|

Font: 9 pt, English (US)

| Page 31: [423] Formatted | Sun ji | 6/11/18 1:55:00 PM |
|---|---|---|

Font: 9 pt, English (US)

| Page 31: [424] Formatted | Sun ji | 6/11/18 1:55:00 PM |
|---|---|---|

Font: 9 pt, English (US)

| Page 31: [424] Formatted | Sun ji | 6/11/18 1:55:00 PM |
|---|---|---|

Font: 9 pt, English (US)

| Page 31: [425] Formatted | Sun ji | 6/11/18 1:55:00 PM |
|---|---|---|

Font: 9 pt, English (US)

| Page 31: [426] Formatted | Sun ji | 5/15/18 10:32:00 PM |
|---|---|---|

Left

| Page 31: [427] Formatted | Sun ji | 6/11/18 1:55:00 PM |
|---|---|---|

Font: 9 pt, English (US)

| Page 31: [428] Formatted | Sun ji | 5/15/18 10:32:00 PM |
|---|---|---|

Left

| Page 31: [429] Formatted | Sun ji | 6/11/18 1:55:00 PM |
|---|---|---|

Font: 9 pt, English (US)

| Page 31: [430] Formatted | Sun ji | 5/15/18 10:32:00 PM |
|---|---|---|

Left

| Page 31: [431] Formatted | Sun ji | 6/11/18 1:55:00 PM |
|---|---|---|

Font: 7.5 pt, English (US)

| Page 31: [432] Formatted | Sun ji | 6/11/18 1:55:00 PM |
|---|---|---|

Font: 9 pt, English (US)

| Page 31: [432] Formatted | Sun ji | 6/11/18 1:55:00 PM |
|---|---|---|

Font: 9 pt, English (US)

| Page 31: [433] Formatted | Sun ji | 6/11/18 1:55:00 PM |
|---|---|---|

Font: 9 pt, English (US)

| Page 31: [433] Formatted | Sun ji | 6/11/18 1:55:00 PM |
|---|---|---|

Font: 9 pt, English (US)

| Page 31: [434] Formatted | Sun ji | 6/11/18 1:55:00 PM |
|---|---|---|

Font: 9 pt, English (US)

| Page 31: [434] Formatted | Sun ji | 6/11/18 1:55:00 PM |
|---|---|---|

Font: 9 pt, English (US)

| Page 31: [435] Formatted | Sun ji | 6/11/18 1:55:00 PM |
|---|---|---|

Font: 9 pt, English (US)

| Page 31: [436] Formatted | Sun ji | 5/15/18 10:32:00 PM |
|---|---|---|

Left

| Page 31: [437] Formatted | Sun ji | 6/11/18 1:55:00 PM |
|---|---|---|

Font: 7.5 pt, English (US)

| Page 31: [438] Formatted | Sun ji | 6/11/18 1:55:00 PM |
|---|---|---|

Font: 9 pt, English (US)

| Page 31: [438] Formatted | Sun ji | 6/11/18 1:55:00 PM |
|---|---|---|

Font: 9 pt, English (US)

| Page 31: [439] Formatted | Sun ji | 6/11/18 1:55:00 PM |
|---|---|---|

Font: 9 pt, English (US)

| Page 31: [439] Formatted | Sun ji | 6/11/18 1:55:00 PM |
|---|---|---|

Font: 9 pt, English (US)

| Page 31: [440] Formatted | Sun ji | 6/11/18 1:55:00 PM |
|---|---|---|

Font: 9 pt, English (US)

| Page 31: [440] Formatted | Sun ji | 6/11/18 1:55:00 PM |
|---|---|---|

Font: 9 pt, English (US)

| Page 31: [441] Formatted | Sun ji | 6/11/18 1:55:00 PM |
|---|---|---|

Font: 9 pt, English (US)

| Page 31: [442] Formatted | Sun ji | 5/15/18 10:32:00 PM |
|---|---|---|

Left

| Page 31: [443] Formatted | Sun ji | 6/11/18 1:55:00 PM |
|---|---|---|

Font: 9 pt, English (US)

| Page 31: [443] Formatted | Sun ji | 6/11/18 1:55:00 PM |
|---|---|---|

Font: 9 pt, English (US)

| Page 31: [443] Formatted | Sun ji | 6/11/18 1:55:00 PM |
|---|---|---|

Font: 9 pt, English (US)

| Page 31: [444] Formatted | Sun ji | 6/11/18 1:55:00 PM |
|---|---|---|

Font: 7.5 pt, English (US)

| Page 31: [445] Formatted | Sun ji | 6/11/18 1:55:00 PM |
|---|---|---|

Font: 9 pt, English (US)

| Page 31: [445] Formatted | Sun ji | 6/11/18 1:55:00 PM |
|---|---|---|

Font: 9 pt, English (US)

| Page 31: [446] Formatted | Sun ji | 6/11/18 1:55:00 PM |
|---|---|---|

Font: 9 pt, English (US)

| Page 31: [447] Formatted | Sun ji | 6/11/18 1:55:00 PM |
|---|---|---|

Font: 7.5 pt, English (US)

| Page 31: [448] Formatted | Sun ji | 6/11/18 1:55:00 PM |
|---|---|---|

Font: 9 pt, English (US)

| Page 31: [448] Formatted | Sun ji | 6/11/18 1:55:00 PM |
|---|---|---|

Font: 9 pt, English (US)

| Page 31: [449] Formatted | Sun ji | 6/11/18 1:55:00 PM |
|---|---|---|

Font: 6.5 pt, English (US)

| Page 31: [450] Formatted | Sun ji | 6/11/18 1:55:00 PM |
|---|---|---|

Font: 10 pt, English (US)

| Page 31: [451] Formatted | Sun ji | 5/15/18 10:32:00 PM |
|---|---|---|

Left

| Page 31: [452] Formatted | Sun ji | 6/11/18 1:55:00 PM |
|---|---|---|

English (US)

| Page 31: [453] Formatted | Sun ji | 5/25/18 2:00:00 PM |
|---|---|---|

Normal

| Page 31: [454] Formatted | Sun ji | 6/11/18 1:55:00 PM |
|---|---|---|

Font: 6.5 pt, English (US)

| Page 32: [455] Formatted | Sun ji | 6/11/18 1:55:00 PM |
|---|---|---|

English (US)

| Page 32: [456] Formatted | Sun ji | 6/11/18 1:55:00 PM |
|---|---|---|

English (US)

| Page 32: [456] Formatted | Sun ji | 6/11/18 1:55:00 PM |
|---|---|---|

English (US)

| Page 32: [457] Formatted | Sun ji | 6/11/18 1:55:00 PM |
|---|---|---|

Font: 9 pt, English (US)

| Page 32: [457] Formatted | Sun ji | 6/11/18 1:55:00 PM |
|---|---|---|

Font: 9 pt, English (US)

| Page 32: [458] Formatted | Sun ji | 6/11/18 1:55:00 PM |
|---|---|---|

Font: 9 pt, English (US)

| Page 32: [459] Formatted | Sun ji | 5/15/18 10:32:00 PM |
|---|---|---|

Left

| Page 32: [460] Formatted | Sun ji | 6/11/18 1:55:00 PM |
|---|---|---|

Font: 7.5 pt, English (US)

| Page 32: [461] Formatted | Sun ji | 6/11/18 1:55:00 PM |
|---|---|---|

Font: 9 pt, English (US)

| Page 32: [461] Formatted | Sun ji | 6/11/18 1:55:00 PM |
|---|---|---|

Font: 9 pt, English (US)

| Page 32: [462] Formatted | Sun ji | 6/11/18 1:55:00 PM |
|---|---|---|

Font: 9 pt, English (US)

| Page 32: [462] Formatted | Sun ji | 6/11/18 1:55:00 PM |
|---|---|---|

Font: 9 pt, English (US)

| Page 32: [463] Formatted | Sun ji | 6/11/18 1:55:00 PM |
|---|---|---|

Font: 9 pt, English (US)

| Page 32: [463] Formatted | Sun ji | 6/11/18 1:55:00 PM |
|---|---|---|

Font: 9 pt, English (US)

| Page 32: [464] Formatted | Sun ji | 8/2/18 4:26:00 PM |
|---|---|---|

Font: 9 pt

| Page 32: [465] Formatted | Sun ji | 6/11/18 1:55:00 PM |
|---|---|---|

Font: 9 pt, English (US)

| Page 32: [466] Formatted | Sun ji | 5/15/18 10:32:00 PM |
|---|---|---|

Left

| Page 32: [467] Formatted | Sun ji | 6/11/18 1:55:00 PM |
|---|---|---|

Font: 7.5 pt, English (US)

| Page 32: [468] Formatted | Sun ji | 6/11/18 1:55:00 PM |
|---|---|---|

Font: 9 pt, English (US)

| Page 32: [468] Formatted | Sun ji | 6/11/18 1:55:00 PM |
|---|---|---|

Font: 9 pt, English (US)

| Page 32: [469] Formatted | Sun ji | 6/11/18 1:55:00 PM |
|---|---|---|

Font: 9 pt, English (US)

| Page 32: [469] Formatted | Sun ji | 6/11/18 1:55:00 PM |
|---|---|---|

Font: 9 pt, English (US)

| Page 32: [470] Formatted | Sun ji | 6/11/18 1:55:00 PM |
|---|---|---|

Font: 9 pt, English (US)

| Page 32: [470] Formatted | Sun ji | 6/11/18 1:55:00 PM |
|---|---|---|

Font: 9 pt, English (US)

| Page 32: [471] Formatted | Sun ji | 6/11/18 1:55:00 PM |
|---|---|---|

Font: 9 pt, English (US)

| Page 32: [471] Formatted | Sun ji | 6/11/18 1:55:00 PM |
|---|---|---|

Font: 9 pt, English (US)

| Page 32: [472] Formatted | Sun ji | 6/11/18 1:55:00 PM |
|---|---|---|

Font: 9 pt, English (US)

| Page 32: [472] Formatted | Sun ji | 6/11/18 1:55:00 PM |
|---|---|---|

Font: 9 pt, English (US)

| Page 32: [473] Formatted | Sun ji | 6/11/18 1:55:00 PM |
|---|---|---|

Font: 9 pt, English (US)

| Page 32: [473] Formatted | Sun ji | 6/11/18 1:55:00 PM |

Font: 9 pt, English (US)

| Page 32: [474] Formatted | Sun ji | 6/11/18 1:55:00 PM |

Font: 9 pt, English (US)

| Page 32: [474] Formatted | Sun ji | 6/11/18 1:55:00 PM |

Font: 9 pt, English (US)

| Page 32: [475] Formatted | Sun ji | 6/11/18 1:55:00 PM |

Font: 9 pt, English (US)

| Page 32: [475] Formatted | Sun ji | 6/11/18 1:55:00 PM |

Font: 9 pt, English (US)

| Page 32: [476] Formatted | Sun ji | 6/11/18 1:55:00 PM |

Font: 9 pt, English (US)

| Page 32: [476] Formatted | Sun ji | 6/11/18 1:55:00 PM |

Font: 9 pt, English (US)

| Page 32: [477] Formatted | Sun ji | 6/11/18 1:55:00 PM |

Font: 9 pt, English (US)

| Page 32: [477] Formatted | Sun ji | 6/11/18 1:55:00 PM |

Font: 9 pt, English (US)

| Page 32: [478] Formatted | Sun ji | 6/11/18 1:55:00 PM |

Font: 9 pt, English (US)

| Page 32: [478] Formatted | Sun ji | 6/11/18 1:55:00 PM |

Font: 9 pt, English (US)

| Page 32: [479] Formatted | Sun ji | 6/11/18 1:55:00 PM |

Font: 9 pt, English (US)

| Page 32: [479] Formatted | Sun ji | 6/11/18 1:55:00 PM |

Font: 9 pt, English (US)

| Page 32: [480] Formatted | Sun ji | 6/11/18 1:55:00 PM |

Font: 9 pt, English (US)

| Page 32: [480] Formatted | Sun ji | 6/11/18 1:55:00 PM |

Font: 9 pt, English (US)

| Page 32: [481] Formatted | Sun ji | 6/11/18 1:55:00 PM |
|---|---|---|

Font: 9 pt, English (US)

| Page 32: [481] Formatted | Sun ji | 6/11/18 1:55:00 PM |
|---|---|---|

Font: 9 pt, English (US)

| Page 32: [482] Formatted | Sun ji | 6/11/18 1:55:00 PM |
|---|---|---|

Font: 9 pt, English (US)

| Page 32: [482] Formatted | Sun ji | 6/11/18 1:55:00 PM |
|---|---|---|

Font: 9 pt, English (US)

| Page 32: [483] Formatted | Sun ji | 6/11/18 1:55:00 PM |
|---|---|---|

Font: 9 pt, English (US)

| Page 32: [483] Formatted | Sun ji | 6/11/18 1:55:00 PM |
|---|---|---|

Font: 9 pt, English (US)

| Page 32: [484] Formatted | Sun ji | 6/11/18 1:55:00 PM |
|---|---|---|

Font: 9 pt, English (US)

| Page 32: [484] Formatted | Sun ji | 6/11/18 1:55:00 PM |
|---|---|---|

Font: 9 pt, English (US)

| Page 32: [485] Formatted | Sun ji | 6/11/18 1:55:00 PM |
|---|---|---|

Font: 9 pt, English (US)

| Page 32: [485] Formatted | Sun ji | 6/11/18 1:55:00 PM |
|---|---|---|

Font: 9 pt, English (US)

| Page 32: [486] Formatted | Sun ji | 6/11/18 1:55:00 PM |
|---|---|---|

Font: 9 pt, English (US)

| Page 32: [486] Formatted | Sun ji | 6/11/18 1:55:00 PM |
|---|---|---|

Font: 9 pt, English (US)

| Page 32: [487] Formatted | Sun ji | 6/11/18 1:55:00 PM |
|---|---|---|

Font: 9 pt, English (US)

| Page 32: [487] Formatted | Sun ji | 6/11/18 1:55:00 PM |
|---|---|---|

Font: 9 pt, English (US)

| Page 32: [488] Formatted | Sun ji | 6/11/18 1:55:00 PM |
|---|---|---|

Font: 9 pt, English (US)

| Page 32: [489] Formatted | Sun ji | 5/15/18 10:32:00 PM |
|---|---|---|

Left

| Page 32: [490] Formatted | Sun ji | 6/11/18 1:55:00 PM |
|---|---|---|

Font: 7.5 pt, English (US)

| Page 32: [491] Formatted | Sun ji | 6/11/18 1:55:00 PM |
|---|---|---|

Font: 9 pt, English (US)

| Page 32: [491] Formatted | Sun ji | 6/11/18 1:55:00 PM |
|---|---|---|

Font: 9 pt, English (US)

| Page 32: [492] Formatted | Sun ji | 6/11/18 1:55:00 PM |
|---|---|---|

Font: 9 pt, English (US)

| Page 32: [492] Formatted | Sun ji | 6/11/18 1:55:00 PM |
|---|---|---|

Font: 9 pt, English (US)

| Page 32: [493] Formatted | Sun ji | 6/11/18 1:55:00 PM |
|---|---|---|

Font: 9 pt, English (US)

| Page 32: [493] Formatted | Sun ji | 6/11/18 1:55:00 PM |
|---|---|---|

Font: 9 pt, English (US)

| Page 32: [494] Formatted | Sun ji | 6/11/18 1:55:00 PM |
|---|---|---|

Font: 9 pt, English (US)

| Page 32: [494] Formatted | Sun ji | 6/11/18 1:55:00 PM |
|---|---|---|

Font: 9 pt, English (US)

| Page 32: [495] Formatted | Sun ji | 6/11/18 1:55:00 PM |
|---|---|---|

Font: 9 pt, English (US)

| Page 32: [495] Formatted | Sun ji | 6/11/18 1:55:00 PM |
|---|---|---|

Font: 9 pt, English (US)

| Page 32: [496] Formatted | Sun ji | 6/11/18 1:55:00 PM |
|---|---|---|

Font: 9 pt, English (US)

| Page 32: [496] Formatted | Sun ji | 6/11/18 1:55:00 PM |
|---|---|---|

Font: 9 pt, English (US)

| Page 32: [497] Formatted | Sun ji | 6/11/18 1:55:00 PM |
|---|---|---|

Font: 9 pt, English (US)

| Page 32: [497] Formatted | Sun ji | 6/11/18 1:55:00 PM |
|---|---|---|

Font: 9 pt, English (US)

| Page 32: [498] Formatted | Sun ji | 6/11/18 1:55:00 PM |
|---|---|---|

Font: 9 pt, English (US), Not Superscript/ Subscript

| Page 32: [499] Formatted | Sun ji | 6/11/18 1:55:00 PM |
|---|---|---|

English (US)

| Page 32: [500] Formatted | Sun ji | 6/11/18 1:55:00 PM |
|---|---|---|

English (US)

| Page 32: [501] Formatted | Sun ji | 6/11/18 1:55:00 PM |
|---|---|---|

English (US)

| Page 32: [502] Formatted | Sun ji | 6/11/18 1:55:00 PM |
|---|---|---|

English (US)

| Page 32: [503] Formatted | Sun ji | 6/11/18 1:55:00 PM |
|---|---|---|

English (US)

| Page 32: [504] Formatted | Sun ji | 6/11/18 1:55:00 PM |
|---|---|---|

English (US)

| Page 32: [505] Formatted | Sun ji | 6/11/18 1:55:00 PM |
|---|---|---|

Font: 9 pt, English (US)

| Page 32: [506] Formatted | Sun ji | 5/15/18 10:32:00 PM |
|---|---|---|

Left

| Page 32: [507] Formatted | Sun ji | 6/11/18 1:55:00 PM |
|---|---|---|

Font: 7.5 pt, English (US)

| Page 32: [508] Formatted | Sun ji | 6/11/18 1:55:00 PM |
|---|---|---|

Font: 9 pt, English (US)

| Page 32: [508] Formatted | Sun ji | 6/11/18 1:55:00 PM |
|---|---|---|

Font: 9 pt, English (US)

| Page 32: [509] Formatted | Sun ji | 6/11/18 1:55:00 PM |
|---|---|---|

Font: 9 pt, English (US)

| Page 32: [509] Formatted | Sun ji | 6/11/18 1:55:00 PM |
|---|---|---|

Font: 9 pt, English (US)

| Page 32: [510] Formatted | Sun ji | 6/11/18 1:55:00 PM |
|---|---|---|

Font: 9 pt, English (US)

| Page 32: [510] Formatted | Sun ji | 6/11/18 1:55:00 PM |
|---|---|---|

Font: 9 pt, English (US)

| Page 32: [511] Formatted | Sun ji | 6/11/18 1:55:00 PM |
|---|---|---|

Font: 9 pt, English (US)

| Page 32: [511] Formatted | Sun ji | 6/11/18 1:55:00 PM |
|---|---|---|

Font: 9 pt, English (US)

| Page 32: [512] Formatted | Sun ji | 6/11/18 1:55:00 PM |
|---|---|---|

Font: 9 pt, English (US)

| Page 32: [512] Formatted | Sun ji | 6/11/18 1:55:00 PM |
|---|---|---|

Font: 9 pt, English (US)

| Page 32: [513] Formatted | Sun ji | 6/11/18 1:55:00 PM |
|---|---|---|

Font: 9 pt, English (US)

| Page 32: [513] Formatted | Sun ji | 6/11/18 1:55:00 PM |
|---|---|---|

Font: 9 pt, English (US)

| Page 32: [514] Formatted | Sun ji | 6/11/18 1:55:00 PM |
|---|---|---|

Font: 9 pt, English (US)

| Page 32: [514] Formatted | Sun ji | 6/11/18 1:55:00 PM |
|---|---|---|

Font: 9 pt, English (US)

| Page 32: [515] Formatted | Sun ji | 6/11/18 1:55:00 PM |
|---|---|---|

Font: 9 pt, English (US)

| Page 32: [515] Formatted | Sun ji | 6/11/18 1:55:00 PM |
|---|---|---|

Font: 9 pt, English (US)

| Page 32: [516] Formatted | Sun ji | 6/11/18 1:55:00 PM |
|---|---|---|

Font: 9 pt, English (US)

| Page 32: [516] Formatted | Sun ji | 6/11/18 1:55:00 PM |
|---|---|---|

Font: 9 pt, English (US)

| Page 32: [517] Formatted | Sun ji | 6/11/18 1:55:00 PM |
|---|---|---|

Font: 9 pt, English (US)

| Page 32: [517] Formatted | Sun ji | 6/11/18 1:55:00 PM |
|---|---|---|

Font: 9 pt, English (US)

| Page 32: [518] Formatted | Sun ji | 6/11/18 1:55:00 PM |
|---|---|---|

Font: 9 pt, English (US)

| Page 32: [518] Formatted | Sun ji | 6/11/18 1:55:00 PM |
|---|---|---|

Font: 9 pt, English (US)

| Page 32: [519] Formatted | Sun ji | 6/11/18 1:55:00 PM |
|---|---|---|

Font: 9 pt, English (US)

| Page 32: [519] Formatted | Sun ji | 6/11/18 1:55:00 PM |
|---|---|---|

Font: 9 pt, English (US)

| Page 32: [520] Formatted | Sun ji | 6/11/18 1:55:00 PM |
|---|---|---|

Font: 9 pt, English (US)

| Page 32: [520] Formatted | Sun ji | 6/11/18 1:55:00 PM |
|---|---|---|

Font: 9 pt, English (US)

| Page 32: [521] Formatted | Sun ji | 6/11/18 1:55:00 PM |
|---|---|---|

Font: 9 pt, English (US)

| Page 32: [521] Formatted | Sun ji | 6/11/18 1:55:00 PM |
|---|---|---|

Font: 9 pt, English (US)

| Page 32: [522] Formatted | Sun ji | 6/11/18 1:55:00 PM |
|---|---|---|

Font: 9 pt, English (US)

| Page 32: [522] Formatted | Sun ji | 6/11/18 1:55:00 PM |
|---|---|---|

Font: 9 pt, English (US)

| Page 32: [523] Formatted | Sun ji | 6/11/18 1:55:00 PM |
|---|---|---|

English (US)

| Page 32: [524] Formatted | Sun ji | 5/15/18 10:32:00 PM |
|---|---|---|

Left

| Page 32: [525] Formatted | Sun ji | 6/11/18 1:50:00 PM |
|---|---|---|

Justified

---

## Author Response (AR3)

**This document contains the responses to the editor's comment on the manuscript version 4, followed by a version 5 manuscript (starts in page 2). The editor's comments and questions are in bold. For each comment / question, the authors' reply / answer is in black, and the corresponding modifications in the manuscript version 3 are marked in blue colour.**

**(1) Author's response to editor**

**However, an inconsistent statement found in your response, which is also cited in the revised manuscript, needs to be corrected. In response to a question raised by the Reviewer # 3 (comment # 16), it is mentioned that "This 20% spectral dependence adopted in their work is associated with findings of Hoffer et al. (2006).", which is untrue. The relative**

**spectral dependence in the imaginary index between 354 and 388 nm assumed in the OMAERUV algorithm was derived from Kirchstetter et al. (2004) results, not Hoffer et al. (2006), although the latter has found similar results (large values of AAE for organics). Please correct this part of the description in the paper and upload the revised manuscript to AMT. Also, it is advised that the manuscript undergoes a careful reading one more time to eliminate any minor language and/or grammar related issue, which might have missed during the revision. The final decision on the**

**manuscript is contingent upon the suggested corrections.**

Thank you for the correction. We have rephrased this part as following: 'This 20% spectral dependence adopted in their work is associated with findings of Kirchstetter et al. (2004). According to them, the absorbing Ångström exponent (AÅE) of urban pollution is near unit and biomass burning aerosols ranges is approximately 2 between 300 nm to 1 μm. In terms of $n_i$, a 20%

increase at 354 nm with respect to the value at 388 nm is equivalent to an AÅE value between 2.5 and 3, depending on the aerosol models of OMAERUV (Jethva and Torres, 2011). Hoffer et al. (2004) also found similar results. They conducted in situ measurements on humic-like substances (HULIS) of Amazonia biomass burning aerosols and found that around 35% - 50% light absorption occurred at 300 nm, whereas only around 15% at 400 nm. Bergstrom et al. (2007) also confirmed this conclusion from several field programs (SAFARI 2000, ACE Asia, PRIDE, TARFOX, INTEX-A). (Line 328-336)'.

We also have gone thoroughly through the manuscript to eliminate minor language and/or grammar related issues as much as possible.

[revised manuscript text omitted]